# Language Modeling by Language Models

**Junyan Cheng**[α]* **Peter Clark**[β] **Kyle Richardson**[β]
Allen Institute for AI[β] Dartmouth College[α]
jc.th@dartmouth.edu kyler@allenai.org
✚ https://github.com/allenai/genesys

## Abstract

*Can we leverage LLMs to model the process of discovering novel language model architectures?* Inspired by real research, we propose a multi-agent LLM approach that simulates the conventional stages of research, from ideation and literature search (proposal stage) to design implementation (code generation), generative pre-training, and downstream evaluation (verification). Using ideas from scaling laws, our system *Genesys* employs a *Ladder of Scales* approach; new designs are proposed, adversarially reviewed, implemented, and selectively verified at increasingly larger model scales (14M∼350M parameters) with a narrowing budget (the number of models we can train at each scale). To help make discovery efficient and factorizable, Genesys uses a novel genetic programming backbone, which we show has empirical advantages over commonly used direct prompt generation workflows (e.g., ∼86% percentage point improvement in successful design generation, a key bottleneck). We report experiments involving 1,162 newly discovered designs (1,062 fully verified through pre-training) and find the best designs to be highly competitive with known architectures (e.g., outperform GPT2, Mamba2, etc., on 6/9 common benchmarks). We couple these results with comprehensive system-level ablations and formal results, which give broader insights into the design of effective autonomous LLM-driven discovery systems.

## 1 Introduction

**Automated scientific discovery** (ASD) (Langley, 1987; Wang et al., 2023), which aims to simulate all aspects of the conventional research process — from ideation/system design to experiment execution, has the promise of changing the way that research is performed by making it more accessible, efficient, and less error-prone. However, while many new large language model (LLM)-driven ASD systems have been recently proposed, including AI Scientist (Lu et al., 2024a; Yamada et al., 2025) and others (Liu et al., 2024b; Jansen et al., 2025b; Schmidgall et al., 2025b), much of this work focuses on open-ended research with unclear goals and where discoveries are hard to verify. This motivates the development of new tasks that address foundational challenges in ASD, tasks that are broad in scope and address impactful research problems, but that have clear goals and criteria for success.

In this paper, we focus on discovery in machine learning and ask: *Can we model the process of discovering novel language model architectures that improve on the standard transformer architecture?* While transformers (Vaswani et al., 2017) remain the *de facto* standard architecture for language models, research into alternative architectures (Dao & Gu, 2024; Sun et al., 2024, 2023; Peng et al., 2024) and transformer variants (Tay et al., 2022b) remains an active and important area of research with connections to the mature field of neural architecture search (NAS) (Elsken et al., 2019). In contrast to open-ended research tasks, architecture research involves a clear goal (i.e., producing an executable architecture design) and offers many metrics for evaluating success. It also introduces

---

*Work done during an internship at the Allen Institute for AI.

39th Conference on Neural Information Processing Systems (NeurIPS 2025).

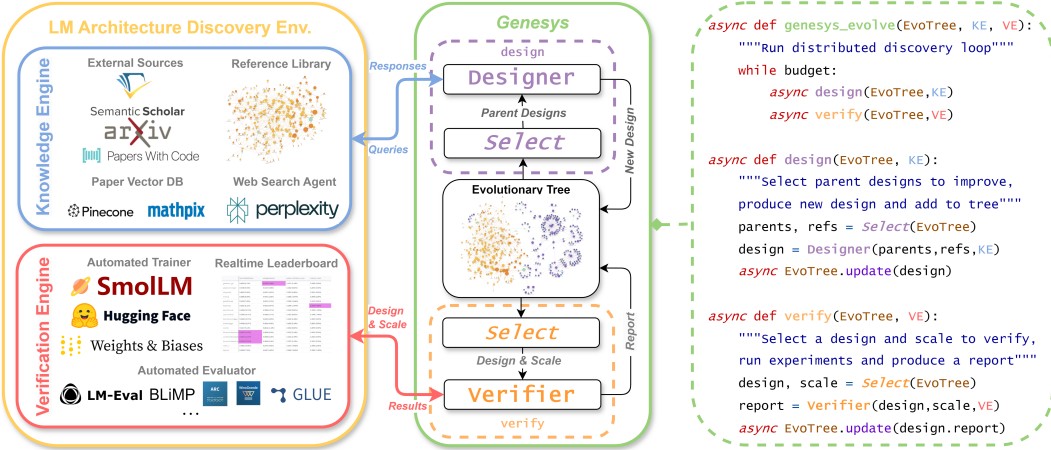

Figure 1: *Can we discover novel language model architectures?* A high-level illustration of our approach, consisting of a discovery environment (**Left**), or LMADE, that provides knowledge access (Knowledge Engine) and automated evaluation (Verification Engine). **Right**: Genesys, a LLM-driven agent system that proposes, implements, then verifies new designs using *design* and *verifier* agents (see algorithmic workflow, far right) and feedback from LMADE.

new challenges for ASD, including requiring deep literature understanding, careful management of resources (e.g., pretraining compute), and the need to write code in an unbounded design space.

Our approach is shown in Figure 1 and factors into a *discovery environment* that provides the foundational tools for ASD and a *discovery system* that produces discovery artifacts using feedback from the environment. Our Language Model Architecture Discovery Environment (**LMADE**) specifically consists of two core resources, a general-purpose **knowledge engine** that provides access to the academic literature and a **verification engine** that provides tools for performing model pre-training and evaluation. Our system **Genesys** then consists of LLM-driven **designer agents** that propose new research ideas and produce executable architecture designs, and **verifier agents** that select designs and perform on-the-fly generative pre-training. At the core of Genesys is an **evolution tree** that stores seed designs and new discovery artifacts. These artifacts are implemented using a special code construct called a generalized autoregressive block (**GAB**) (Figure 3) that is capable of expressing a wide range of neural architecture types and factorizable into discrete tree representations that allow us to employ efficient genetic programming (**GP**)-style optimization.

We performed large-scale discovery experiments that resulted in 1,062 new architecture designs fully verified through pre-training (at the 14M-350M parameter scales). To make verification feasible, we employ a **Ladder-of-Scales** approach where new designs are verified on increasingly larger model scales with a controlled budget, closely following the methodology used in research on small LMs (Lu et al., 2024b; Hu et al., 2024). To our knowledge, our work constitutes the largest ASD experiment of its kind, involving >1 billion tokens, 2.76M lines of code, and 86K agent interactions.[2] We find that our system produces highly competitive designs, e.g., ones that outperform comparable transformer and mamba2 models (Dao & Gu, 2024) in 6 / 9 common downstream tasks. These results are significant and show the feasibility of LLM-driven discovery for competitive ML research. Through systematic ablations, we also find that our system leads to more stable discovery (e.g., measurable improvements in the *fitness* of new designs over time) and effective code generation (e.g., ~86% percentage point improvement in successful design generation), which give broader insight into how to effectively build large-scale discovery systems.

## 2 Related work

**AI in Scientific Discovery**  AI approaches to ASD have recently proliferated, notably in biomedical science (Jumper et al., 2021; Cheng et al., 2023; Wong et al., 2024), material science (Park et al.,

---

[2]All code and discovery artifacts (e.g., new designs, agent interactions and dialogues) can be found at https://genesys.allen.ai (live console) and `https://github.com/allenai/genesys` (system code).

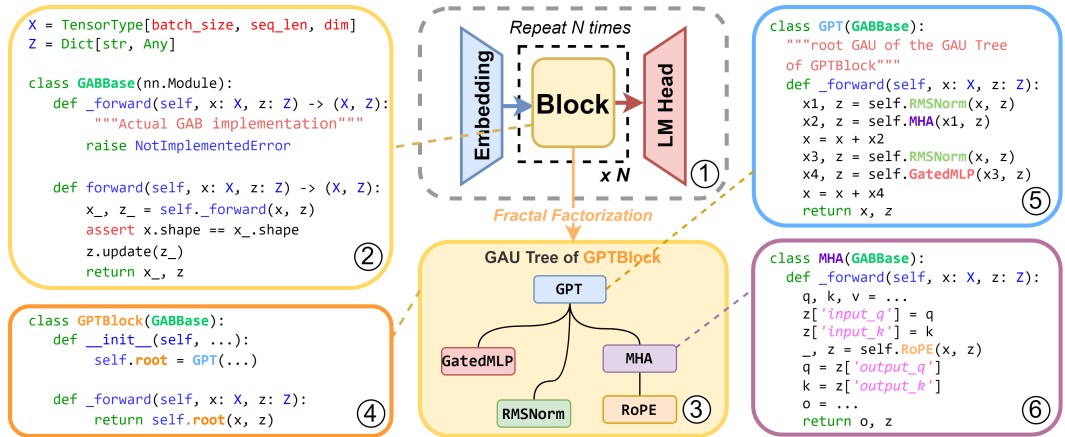

Figure 3: *What are we trying to discover?* ① visualizes standard autoregressive LMs and the **blocks** that our system aims to discover (implemented via the Pytorch modules in ② and ④ with function type $(X,Z) \rightarrow (X,Z)$). ⑤ shows an implemented block for the GPT and its factorization into a tree ③ that shows the units in that block (e.g., multi-head attention implemented in ⑥).

2024; Merchant et al., 2023), and other areas (Chen et al., 2024; Nearing et al., 2024). As discussed above, recent attempts at fully end-to-end research via LLM-driven systems, such as AI Scientist (Lu et al., 2024a; Yamada et al., 2025), AgentLab (Schmidgall et al., 2025a), CodeScientist (Jansen et al., 2025a), and AIGS (Liu et al., 2024a), have focused on open-ended research tasks with unclear goals and evaluation protocols. In contrast, we focus on the challenging task of neural architecture discovery, which offers a clear objective yet involves many new challenges for ASD.

**Language Model Architectures** Our work relates to research on efficient transformer variants (Xiao et al., 2024; Ye et al., 2024), and alternative architectures, such as state-space models (Gu et al., 2022; Dao & Gu, 2024), modern RNNs (Peng et al., 2024; Beck et al., 2024; Feng et al., 2024), and test-time training (Sun et al., 2024; Behrouz et al., 2024). Since our system aims to perform autonomous research in this area, much of this related work is modeled directly and stored in a reference library shown in Figure 2 that serves as the background work used for system ideation.

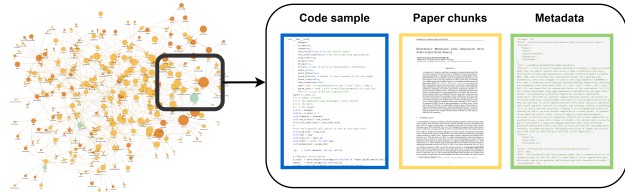

Figure 2: An illustration of our **reference library** in LMADE – a graph of papers on architecture design (nodes containing details of the original paper, code snippets, and other details) and citation links (edges) – that our system queries when performing background research.

**Neural Architecture Search (NAS)** Lastly, we take inspiration from the NAS literature (Chitty-Venkata et al., 2022; White et al., 2023; Elsken et al., 2019; Chen et al., 2023) which has the same aim of discovering improved architectures. Unlike this work, which traditionally searches fixed operation spaces (e.g., attention heads, convolution kernels), we aim for a broader space of operations and architectures and, importantly, attempt to model the broader scientific discovery process. We follow many approaches in NAS that employ genetic programming techniques (GP) (Koza, 1994) and more recent approaches that mix GP with LLMs (Hemberg et al., 2024; Romera-Paredes et al., 2024).

## 3 Language Model Architecture Discovery

As illustrated in Fig. 3 ①, standard LMs work by embedding input, then applying $N$ *layer or block transformations* over that input to produce a final representation (e.g., one that can be used for next token prediction as in autoregressive LMs). Central to any layer/block is a **block design**, concretely a piece of code $B_{LM}$, that dictates how information flows through a network. Our goal is to jointly discover novel autoregressive block designs $B_{LM}$ while also modeling the broader research

process associated with producing $B_{LM}$. In this section, we define this problem formally (§ 3.1) and introduce our Language Model Architecture Discovery Environment (**LMADE**) (§ 3.2) that provides the foundational tools used for discovery and for evaluating block designs $B_{LM}$.

## 3.1 Problem Definition and Goals

We define architecture discovery as a program search problem that involves finding an optimal program $\hat{B}_{LM}$ (in the space of valid programs $\mathcal{B}_{\mathcal{LM}}$) that maximizes some **fitness function** $: \mathcal{B}_{\mathcal{LM}} \to \mathbb{R}$. We can define this formally as:

$$\hat{B}_{LM} = \underset{B_{LM} \in \mathcal{B}_{\mathcal{LM}}}{\operatorname{argmax}} \left\{ \mathcal{F}(B_{LM}) \right\}, \quad \text{with } \mathcal{F}(B_{LM}) = \frac{1}{M \cdot K} \sum_{i=1}^{M} \sum_{j=1}^{K} \operatorname{Perf}(B_{LM}, \mathcal{D}_i, S_j) \quad (1)$$

where, following standard practice, $\mathcal{F}$ is defined as the average empirical performance Perf of $B_{LM}$ on a set of $M$ downstream tasks $\{\mathcal{D}_1, ..., \mathcal{D}_M\}$ across $K$ different model scales $\{S_1, ..., S_K\}$ (i.e., model parameter sizes). Operationally, a valid program will be any syntactically correct instantiation of the `GABBase` module in Figure 3 ② that involves (via the implementation of `_forward`) a differentiable, causal transformation of input tensor $X \in \mathbb{R}^{\text{batch\_size} \times \text{seq\_len} \times \text{emb\_dim}}$ to an output tensor of the same dimension and type, along with the other semantic constraints (see Table 6).

The role of LMADE is to provide input and feedback $\mathcal{I}$ to a separate discovery system that produces new block designs, as well as to provide all other tools needed for checking the validity of designs, verifying them through experiments, and computing $\mathcal{F}$. We consider these components next, followed by a discussion of our discovery system, Genesys, in § 4.

## 3.2 Language Model Architecture Discovery Environment (LMADE)

LMADE consists of two core utilities, a knowledge engine and a verification engine that provide signal and feedback $\mathcal{I}$ to a discovery system. The **Knowledge Engine (KE)** provides information from the academic literature that is needed to produce new research ideas. It specifically includes a manually curated *reference library* (Fig. 2) of 297 LM papers (stored in a searchable vector store) coupled with code, as well as tools for querying ArXiv, Semantic Scholar (Kinney et al., 2023), and the web via services such as Perplexity.ai. More details are provided in § B.1.3.

The **Verification Engine (VE)** then provides tools for verifying the correctness of designs and executing experiments. In the former case, the VE uses a general code construct, called a Generalized Autoregressive Block (**GAB**) (operationalized by the `GABBase` class in Figure 3, see code templates in App B.3) to represent all architecture designs $B_{LM}$ and uses the structure of this module to check the syntactic correctness of each design. Semantically, the VE also includes a **Symbolic checker** that performs static (AST-based) and runtime (PyTorch-based) code analysis to check for differentiability, causality, numerical stability, and the efficiency of a code design as detailed in Table 6 (further details in § B.1.2). Finally, VE can perform design verification by automating pretraining on a filtered SmolLM corpus (Allal et al., 2025) and evaluation on 29 selected LM-Eval benchmarks (Gao et al., 2024). Standard pretraining protocols (Biderman et al., 2023) are applied (see § B for more details).

# 4 Genesys: Genetic Discovery System

Using resources from LMADE, our system Genesys employs a genetic programming **(GP)**-style optimization to discover new designs. Importantly, this relies on a factorized representation of code designs and an **evolution tree** described in § 4.1. Genesys then includes two core sets of agents: LLM-driven **designers** (§ 4.2) that select past designs from the evolution tree, propose unit-wise modifications to those designs based on background research, then implement the proposed designs and add them to the evolution tree. **Verifiers** (§ 4.3) select designs from the evolution tree and verify them through budget-aware pre-training. We consider each component in turn and provide various technical justifications for our design decisions that we further formalize in Appendix A.

## 4.1 Evolution Tree and Design Factorization

In order to apply GP-optimization, Genesys factorizes each block program $B_{LM}$ into a discrete tree representation called a **generalized autoregressive unit** (**GAU**) tree. For example, Figure 3③ shows a GAU tree for the transformer block, where each unit, or GAU, corresponds to a portion of the executable code implementation (see example in Fig. 16). This factorization forms the basis of an **evolutionary tree** (Figure 4) that stores new designs with these

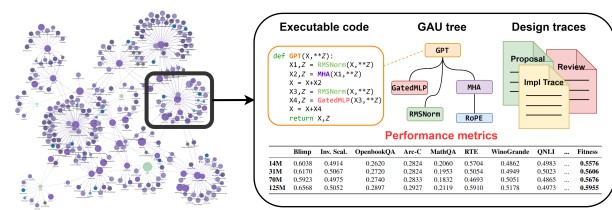

Figure 4: The evolution tree in Genesys (left), where nodes denote block designs and contain each design's **executable code**, a **GAU tree** representation of the code, **design traces**, and empirical **performance metrics**.

details and other artifacts. Importantly, this provides an interpretable representation of the discovery search space, where each block in the tree can be compared to another via their atomic units.

Based on this tree representation, standard GP operations (Koza, 1994) such as *mutation* (i.e., modifying a unit of a design) and *crossover* (i.e., merging the units of multiple designs) can be applied (see examples in Figure 14). In contrast to traditional GP, however, we use a relaxed form of GP that does not rely on a fixed inventory of mutation operators but instead uses an LLM to generate new code units, similar to Hemberg et al. (2024); Romera-Paredes et al. (2024). These units are implemented using the same GAB class construct described above, which allows for a consistent set of syntactic and semantic checks on the validity of each unit implementation.

**Design factorization: formal considerations** As we discuss later (§ 4.2), by representing each code artifact $A$ as a GAU tree, consisting of a sequence of GAUs $A = I_1, ... I_N$ (each implemented as a GABBase module in Figure 3②), we can use such a factorization to not only perform GP-style optimization and efficient validity checking, but also devise efficient algorithms for block generation. While such representations are useful for understanding the discovery space, one natural question is: *Does such a factorization adequately capture the full design space, or does it oversimplify the problem in some limiting way?* As noted in Figure 3 using torchtyping- (Kidger, 2021) and Python-style type annotations, blocks and their units are naturally expressible as compositions of functions of

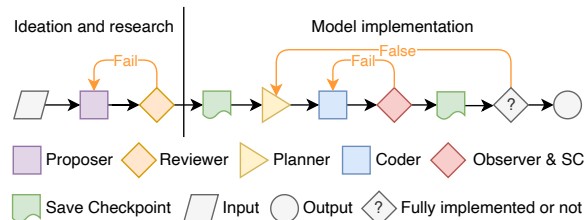

Figure 5: *How do we model the ideation and design stages in LM discovery?* A high-level illustration of the agent subsystems, including a pair of *proposer-reviewer* LLM (consisting of a **proposer** and **reviewer** agent) that draft and score research ideas (left) and a hybrid network of *planner-coder-observer* agents (right) that produce code (consisting of a **planner** agent, a **coder** agent, and **observer** agent coupled with a **symbolic checker** (SC) tool that performs static and execution-based code analysis).

type $(X,Z) \rightarrow (X,Z)$ (or more generally $\Sigma \rightarrow \Sigma$). Through further formalization of the language underlying these structures, in A.2 we show formally that any composition of $\Sigma \rightarrow \Sigma$ functions guarantees a decomposition of the resulting code into the kinds of GAU tree representations we use.

## 4.2 Model Designers

As shown in Figure 5, we break the design process into two stages, a **proposal stage** and an **implementation stage**. Further algorithmic details are provided in § B.2 with prompts in § F.

**Proposal stage** The proposal stage starts by selecting a past design or pair of designs from the evolution tree (using the strategy in § 4.3) along with background references from the reference library, which queries the knowledge engine in LMADE. Based on this input, an LLM **proposal agent** comes up with a novel research idea involving a modification of the selected design(s) and writes a research proposal with high-level details of that idea and its implementation. Modifications are limited to either mutating a particular unit in the selected design, mixing units if multiple designs are selected (crossover), or designing a block from scratch (i.e., a special case of mutating the root

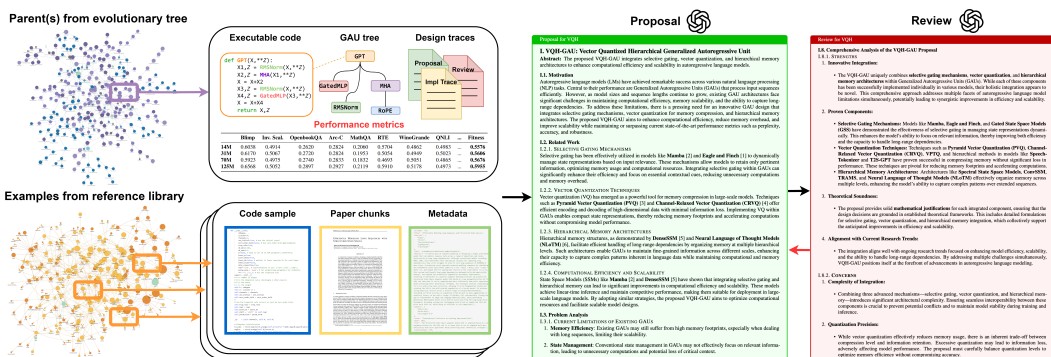

Figure 6: *How are new design ideas generated and vetted?* An illustration of our proposer-reviewer agent architecture using real example design artifacts (right). First, a proposer agent uses parent designs (GAU tree, proposal, verification reports) from the evolutionary tree and selected references (code, text chunks, metadata) from the reference library to generate a research proposal, which is then adversarially reviewed and scored by a reviewer agent before proceeding to implementation.

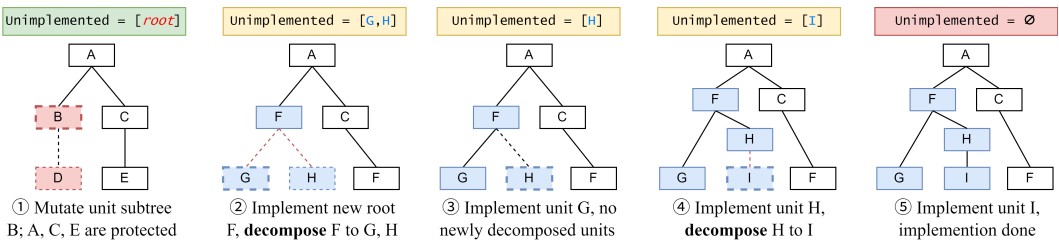

Figure 7: The abstract code generation process in Genesys, where individual units in a code's unit tree (**GAU tree**) are modified and implemented piece-by-piece. Here shows the implementation of a mutation in ① involving the marked units (B, D) (all white units are protected). A new root F ② replaces B and consists of sub-units G and H, which each get implemented in turn (i.e., transformed into executable code) ③-⑤ until the new design is fully functional.

unit in a tree). Then, a separate LLM **reviewer agent** reviews and scores this proposal in a way analogous to an adversarial peer-reviewer and compares against past proposals to ensure novelty (§ B.1.5). An illustration provided in Fig. 6. This loop continues until the proposal is accepted and the score assigned by the reviewer exceeds a certain threshold. (see full algorithm in Alg. 1)

**Implementation stage** The accepted research proposals are translated to executable designs $B_{LM}$ in this stage. Given that proposals involve unit-wise modifications to existing designs and their GAU trees, this allows for the step-by-step recursive generation shown in Fig. 7 (see also Alg. 3). This builds up a block program by incrementally constructing the GAU tree, which implicitly performs the factorization online. It maintains an `Unimplemented` list, initialized with the root of the editing subtree or a new tree. In each step: **1)** A LLM `planner` agent selects an unimplemented GAU, and provides a plan for its implementation; **2)** A LLM `coder` agent generates the Python code, potentially decomposing the GAU by declaring new children (via special statements), which will be added to `Unimplemented` with placeholder implementations; **3)** Implementation is validated by a *symbolic checker* (verifying GAU/GAB compliance for the current unit and the entire tree) and a LLM `observer` that assesses code quality, proposal adherence, and novelty against prior/sibling implementations,

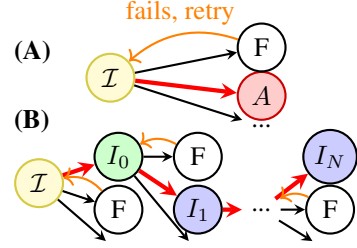

Figure 8: A visualization of a common direct prompting strategy (**A**) versus our unit-based generation strategy (**B**) and how these approaches behave in the presence of failures Ⓕ (i.e., cases when the produced artifacts do not satisfy the desired properties).

then rates it (threshold: $3/5$). If both checks pass, the GAU is accepted; otherwise, the tree state *reverts* for a retry. The implementation finishes when the `Unimplemented` is empty.

**Unit-based code generation: algorithmic advantages**   One motivation for unit-based code generation is that a direct prompting approach often fails to produce useful and valid code (i.e., code that not only improves on past designs but also satisfies the constraints in Table 6). Such a direct approach, which is familiar to many code generation systems, is illustrated in Fig. 8(**A**) and involves presenting a model with input $\mathcal{I}$ and, on failure to produce a valid/useful output $A$, retries (e.g., with a modified $\mathcal{I}$) until success. This difficulty can be understood formally: given the probability $p$ of generating a valid/useful artifact $A$, the expected number of (i.i.d) calls to the model is $\mathbb{E}[\mathbf{calls}] = \frac{1}{p}$, which will be prohibitive for most complex discovery problems with small $p$. In contrast to direct prompting, our approach (Fig. 8(**B**)) generates unit-by-unit $A = I_0, ..., I_N$, where each successful unit $I_j$ is frozen in place before the next unit is generated. This operationalizes a Viterbi-style search (Viterbi, 1967), which we show from the first principles in § A.1 exponentially reduces the expected number of model calls. This explains the results in Table 3, and highlights the importance of a factorized search space.

## 4.3   Verifiers and Efficient Evolution

**Distributed approach**   To allow for efficient exploration, Genesys runs the *designer* and *verifier* agents in parallel as in Romera-Paredes et al. (2024) (Fig. 1 Right), both of which communicate through the evolutionary tree. The evolutionary tree is initially populated with several state-of-the-art architecture designs, including the `transformer/GPT` (Biderman et al., 2023), `Mamba2` (Dao & Gu, 2024), `RetNet` (Sun et al., 2023), `RWKV6` (Peng et al., 2024), and `TTT` (Sun et al., 2024). *Designer nodes* continuously select parents (per the strategy below), query LMADE for references, and task the designer agent (§ 4.2) with generating new designs. Concurrently, *verifier nodes* select designs/scales and run verification in the LMADE Verification Engine whenever available. Further analysis of optimal worker ratios is provided in § E.

**Design selection**   To effectively allocate resources, designers and verifiers select designs from the evolutionary tree by balancing exploitation (i.e., refining promising designs) and exploration (i.e., investigating diverse options). Designs in the evolutionary tree are assessed along two dimensions: **fitness** $\mathcal{F}$ (i.e., aggregate downstream task performance) and **confidence** (i.e., number of model scales where verification was performed). Designs are then categorized into four quadrants (see Figure 15) by upper quartiles of the two dimensions (e.g., Good & Confident, Good & Unconfident). Designers primarily exploit 'Good & Confident' designs for further improvement and explore 'Poor & Confident' ones. Verifiers primarily exploit 'Good & Unconfident' designs (verifying at more scales) and explore 'Poor & Unconfident' ones. Selection within quadrants is probabilistic, favoring higher-ranked designs (by averaging fitness and confidence) but allowing occasional random picks. GP operations (mutation/crossover/scratch design) are also chosen probabilistically (0.75/0.2/0.05 in our experiments) (see Alg. 4).

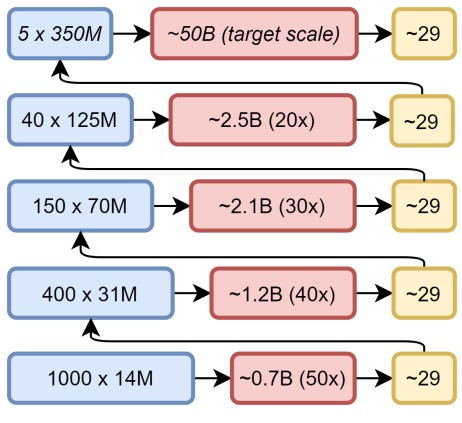

Figure 9: *How do we perform efficient verification?* Our **Ladder-of-scales** strategy involves starting small (training 1,000 14M parameter models on 0.7B tokens; bottom) and allocating progressively fewer trials for larger scales (training 5 350M parameter models on 50B tokens; top).

**Budget management**   Verifying every design at each scale is prohibitively expensive. Inspired by scaling laws (Kaplan et al., 2020; Tay et al., 2022a) – which suggest that performance correlates across scales – and the methodology commonly employed for small LMs (Hu et al., 2024), we implement the **Ladder of Scales** strategy shown in Fig. 9 where several trials are performed at small parameter/token sizes, then scaled with decreasingly fewer trials. Formally, a total verification budget $B_m = \{\beta_0, ..., \beta_{N_S}\}$ across $N_S + 1$ scales (e.g., 14M-350M parameters) is structured pyramidally: more trials $\beta_i$ at smaller scales, with $\beta_{i+1} \approx sr_i \cdot \beta_i$ ($sr_i < 1$ is the target inter-scale selection ratio). Higher-scale budgets are released gradually to ensure fairness and to prevent early depletion. A dynamic *allocatable budget* $B_a = \{\alpha_0, ..., \alpha_{N_S}\}$ is initialized with $\alpha_0 = 1$ and $\alpha_{i>0} = 0$. Budgets are replenished at the lowest scale upon use, and higher-scale budgets $\alpha_{i+1}$ are released when used lower-scale budgets $\beta_i$ exceed $1/sr_i$. The verifier node selects designs per the above strategy, verifying at the lowest unverified scale $i$ with an available budget $\alpha_i > 0$ (see Alg. 5).

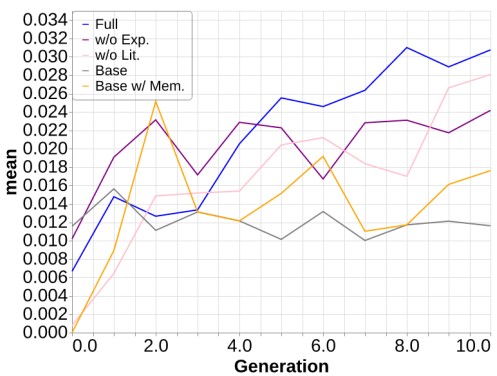 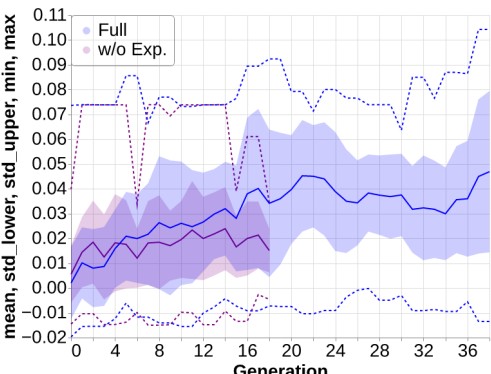

Figure 10: *Is the discovery process improving with time and benefiting from different components?* The mean fitness (i.e., empirical performance) of designs through the different **generations** of discovery, showing (**left**) the first 300 designs and (**right**) the (mean, halved min/max and std bands for better visualization) for 1000 designs. We compare our **full system** against ablated versions that remove experiment verification (**w/o Exp**), literature search (**w/o Lit.**), and direct generations (**Base**).

## 5 Experiments

We empirically test the core components and overall effectiveness of our Genesys system. We aim to demonstrate the advantages of our approach and the potential of our system to perform advanced LM research. Our investigation is structured around three key research questions: **RQ1** (§ 5.1): *Does our GP approach lead to a better and more stable optimization process, one where each system component positively impacts the evolution process?* **RQ2** (§ 5.2): *Does our unit-based code generation approach enhance design generation quality and efficiency compared to direct prompting methods, and how much do the individual model implementation agents (Fig. 5) affect performance?* **RQ3** (§ 5.3): *Does our system ultimately discover architecture designs that are competitive with standard architectures?* Detailed setups are provided in § C, with additional results in § D.

### 5.1 Evolutionary Experiments

To assess the effectiveness of our overall system and its components, we compare our **full** Genesys system against systematical ablational variants, including **w/o Lit.** that removes access to the background literature, **w/o Exp.** that removes experiment verification and fitness information from selection and **Base** that only search from the five starting designs with no evolution (**Base w/ Mem** extends this by allowing new designs to be used solely as background references similar to Romera-Paredes et al. (2024)). These variations allow us to address the following question: *How much does literature understanding, verification feedback, and access to acquired knowledge contribute to successful discovery?*

| | $\Delta \uparrow$ | $\Delta_{max} \uparrow$ | $SR \uparrow$ | $\nu \downarrow$ | $max \uparrow$ | $MDD \downarrow$ |
|---|---|---|---|---|---|---|
| *First 300 designs* | | | | | | |
| Full | **4.10** | 4.16 | **69.0** | 1.9 | **61.0** | **-0.38** |
| w/o Exp. | 2.20 | 2.33 | 26.3 | 2.6 | 60.3 | -1.10 |
| w/o Lit. | 3.37 | 3.67 | 56.7 | 1.9 | 60.3 | -0.62 |
| Base | 0.01 | 0.69 | 0.2 | **1.4** | 59.5 | -0.76 |
| w/ Mem. | 2.81 | **4.61** | 19.6 | 4.5 | 60.7 | -2.46 |
| *First 500 designs* | | | | | | |
| Full | **5.87** | **6.72** | **48.4** | **2.9** | **62.5** | **-0.79** |
| w/o Exp. | 1.69 | 3.20 | 12.2 | 3.3 | 60.8 | -1.13 |
| *First 1000 designs* | | | | | | |
| Full | 7.13 | 8.13 | 26.5 | 4.4 | 63.3 | -1.55 |

Table 1: Evolution experiments under different configurations (%). Bold/underlined denotes the best/second.

We sample 300 designs for all configurations and 500 and 1000 designs for selected setups due to computational constraints. Evolutionary progress was evaluated using the following population fitness metrics over time (population size $S_P = 50$, step size $k_s = 25$): the **End** ($\Delta$) and **peak** ($\Delta_{max}$) fitness improvement. **Volatility** ($\nu$), or the standard deviation (std) of generational differences. We also measure the **Sharpe Ratio** ($SR$), or the risk-adjusted improvement computed as the mean of generational differences divided by their std and the **Maximum Drawdown** ($MDD$) that measures the maximal fitness decrement, which indicates stability. We note that these last metrics originate from financial economics Sharpe (1994); Gu et al. (2020b) with $MDD$ and $\nu$ being the complement

| | **Pl.** | **Coder** | **Obs.** | **SC** | **UG** | Valid | Attempts | Costs | LFC |
|---|---|---|---|---|---|---|---|---|---|
| Full | ✓ | ✓ | ✓ | ✓ | ✓ | 92% | 2.6 ($\pm$1.1) | 15.0 ($\pm$18.5) | 181 ($\pm$44) |
| No UG | ✓ | ✓ | ✓ | ✓ | | 73% | 3.0 ($\pm$1.7) | 7.9 ($\pm$7.1) | 75 ($\pm$29) |
| No Pl. | | ✓ | ✓ | ✓ | ✓ | 91% | 2.6 ($\pm$1.1) | 16.0 ($\pm$20.8) | 218 ($\pm$69) |
| No Ob. | ✓ | ✓ | | ✓ | ✓ | 89% | 2.6 ($\pm$1.1) | 12.1 ($\pm$20.1) | 211 ($\pm$67) |
| No SC | ✓ | ✓ | ✓ | | ✓ | 30% | 2.4 ($\pm$1.0) | 2.9 ($\pm$4.7) | 167 ($\pm$33) |
| Direct | | ✓ | | | | 6% | 1.1 ($\pm$0.2) | 0.3 ($\pm$0.3) | 49 ($\pm$15) |

Table 3: Code quality vs. model designer variants w/wo the planner (Pl.), Coder, Observer (Ob.), Symbolic Checker (SC), and Unit-based Generation (UG), and a "Direct" prompting strategy. **Valid** reports the % of valid code generated, **Attempts** is the avg. number of generation attempts (at most 5 times), **Costs** is the average token cost, and **LFC** is the average Lines of Function-body Code.

of $SR$. Recently, $SR$ has been used in reinforcement learning to measure risk-return balance over time, which also suits our evolutionary search process and the exploration-exploitation trade-off.

The results are reported in Table 1. For the initial 300 designs, our full system performed the best by having the highest fitness improvements ($\Delta = 4.10\%$, $\Delta_{max} = 4.16\%$) and superior stability with the highest $SR = 0.69$ and lowest $MDD = -0.38\%$. w/o Lit. reduced $\Delta$ by 0.73% and $SR$ to 0.567, underscoring the importance of literature guidance for stable progress. Base showed negligible improvement, while w/ Mem. boosted $\Delta$ (to 2.81%) and $SR$ (to 0.196), confirming the value of experience. For our extended runs (500 & 1000 Designs), we see similar advantages over **w/o Exp.** with doubled $\Delta_{max}$ and quadrupled $SR$, highlighting the role of experimental feedback as selection signals. Full continued to improve up to 1000 designs, reaching a peak fitness of 0.633 and showing signs of convergence (Fig. 10 Right). Interestingly, its evolutionary tree (Fig. 35) displays "hubness", which is analogous to the long-tail distribution of paper citations (Wu et al., 2009).

Besides impacting fitness and stability, in Table 2, we show how the removal of components can result in increased errors during the verification process. For example, removing experiments led to code with a $\sim$19% higher error rate compared to our full system, showing how design evaluations can help avoid downstream errors later in discovery.

| | Full | w/o Exp. | w/o Lit. | Base | w/ Mem. |
|---|---|---|---|---|---|
| $Err$ | 8.61% | 27.31% | 7.67% | 21.09% | 23.70% |

Table 2: The error rate (%) during the design verification and evaluation stages under different system ablations.

## 5.2 Designer Agent Analysis

As mentioned earlier, a key bottleneck in our discovery system is generating valid code that satisfies the conditions in Table 6. To measure the effectiveness of our full system with its different design agents and symbolic checker (see again Fig. 5), we directly evaluated the implementation abilities of our designer agents using 100 proposals from our full evolution run as test cases. We systematically compared against variants of our system that removed the code planner (**No. Pl**), the observer agent (**No Ob.**), the unit-based generation strategy (**No UG**), and the semantic checker (**No SC**). Finally, we compared against the **Direct** prompting approach discussed in Figure 8. We measured the rate of successful implementations passing all checkers (**Valid (%)**), the average attempts (**Attempts**), and token costs (**Costs**) and Lines of Function-body Code (**LFC**) as a proxy for code complexity/quality.

Table 3 presents the results. Removing UG (No UG) significantly degraded 20.7% in the valid rate and 58.6% in LFC compared to the Full agent, highlighting the benefit of the structured, unit-by-unit approach for generating complex and valid code. Disabling the symbolic checker (No SC) drastic reduced the valid rate by 67.4%, which confirms its importance. Ablating the Planner (No Pl.) or Observer (No Ob.) results in minimal quantitative impact, yet both play a crucial qualitative role, such as guiding the implementation order and assessing novelty/quality. Moreover, the Full agent's code complexity (avg. LFC 181) was comparable to the human-written *reference library* (avg. LFC 220), suggesting more realistically complex designs. Simpler setups (Direct, No UG) often produce trivial outputs (e.g., basic ConvNets) with a significantly lower magnitude in LFC (~50).

**Few vs. Many Samples: formal considerations** One of the design principles underlying our agent system is that designers should produce few but deliberate and interpretable designs, much like in

everyday research. This is in contrast to traditional GP approaches where a vast number of simple trials are routinely performed (e.g., see Real et al. (2020); Chen et al. (2024)). This raises the natural question: *It is more effective to design more complex, and ultimately more expensive, agent systems that are more deliberate (e.g., our system with planners/observers/coders), or to rely on simpler, more cost-effective agent systems that can perform more trials?* In Appendix A.3 we provide a formal argument that attempts to justify the former and link this decision with our Viterbi-style search.

|           | Blimp  | Wnli   | RTE    | WG     | CoLA   | SST2   | WSC    | IS     | Mrpc   | Avg.   |
|-----------|--------|--------|--------|--------|--------|--------|--------|--------|--------|--------|
| *Random*  | 69.75  | 43.66  | 52.71  | 48.78  | 50.00  | 49.08  | 49.82  | 50.03  | 31.62  | 49.49  |
| GPT       | 92.70  | 60.56  | 62.80  | 52.17  | *53.24*| 54.13  | 56.76  | 55.31  | 68.38  | 61.78  |
| Mamba2    | 83.22  | 63.38  | **63.88** | 51.22 | 55.94  | 56.58  | 57.12  | 53.85  | 67.89  | 61.45  |
| RWKV7     | 88.76  | 61.97  | 60.21  | *49.80*| 54.25  | 55.32  | 54.57  | **57.00** | 68.38 | 61.14  |
| RetNet    | 85.16  | 61.97  | 61.35  | 50.51  | **56.29** | 55.43 | 56.03  | 54.95  | *56.37*| 59.78  |
| TTT       | 86.13  | 63.38  | *55.23*| 50.75  | 55.55  | 56.35  | 54.93  | 55.31  | 59.80  | 59.71  |
| VQH       | **94.37** | 59.15 | 59.91 | 50.28  | 54.25  | 53.56  | *53.83*| *49.45*| 56.62  | 59.05  |
| HMamba    | 83.74  | 64.79  | 61.35  | **53.59** | 54.69 | **57.04** | 56.40 | 54.58 | 59.31  | 60.61  |
| Geogate   | 90.95  | 59.15  | 61.35  | 52.72  | 54.25  | 55.32  | **58.96** | 54.95 | 68.63  | **61.81** |
| Hippovq   | 87.96  | *50.70*| 59.91  | 50.28  | 54.25  | 55.73  | *53.83*| 55.68  | 69.88  | 59.80  |
| SRN       | *80.83*| **65.52** | 59.55 | 50.75 | 54.45  | *52.98*| 56.03  | 54.95  | 61.03  | *59.57*|

Table 4: *How good are our discovered designs?* A comparison of our seed models (top) vs. our discovered models (350M scale, 50B training tokens) based on end-task benchmark accuracy (%). Bold/underline/italics denote the best/second/worst.

## 5.3 Discovered Model Evaluation

To measure the overall performance of our new designs, we perform a standard zero-shot end-task evaluation that compares the top 5 designs discovered using the "Full" Genesys system against the five human seed designs shown in Figure 35 (GPT2, TTT, Mamba2, RWKV, and RetNet). We specifically evaluated these designs on the scales of 125M and 350M parameters, trained on 25B and 50B tokens, respectively. Following standard protocols Groeneveld et al. (2024), tasks were selected based on their informativeness on smaller scales (see § E.3.1 and Table 16 for details).

**How good are the discovered designs?** Tables 4 and 14 (§ D.1) show the results of our evaluation. Our discovered designs outperformed/matched baselines on 7/9 (125M) and 6/9 (350M) benchmarks, with superior averages. Although no single model dominated all tasks, consistent with many other studies on small LM development (Fourrier et al., 2024)), the discovered designs consistently performed competitively with the state-of-the-art human baselines. This shows the feasibility of using LLMs to automate human-level LM research, at least at the functional level. We see scaling experiments, as well as analysis on the intelligibility of the AI designs, as promising future work.

## 6 Discussion, Limitations & Conclusion

We introduce Genesys, an autonomous system for discovering novel LM designs, featuring a novel unit-based design agent and cost-effective distributed evolution. We also present LMADE, a resource environment to support further research in this field. Current limitations include integrating efficiency-focused innovations, such as FlashAttention (Dao, 2024), hindered by complex hardware-specific evaluations, and the constraints of billion-parameter-level discovery due to limited computational resources. Future work will aim to enhance the agent's learning from feedback, possibly via reinforcement learning, as well as to develop a more adaptive design selection strategy. Our large-scale experiments yielded 1,062 novel LM architectures (14M-350M parameters), fully verified with pretraining. This is, to our knowledge, the largest automated LM discovery experiment. Genesys produced highly competitive designs; some outperformed human baselines such as the GPT and Mamba2 models in common downstream tasks. These results show the feasibility and lay the groundwork for autonomous evolutionary systems in scientifically complex and costly domains.

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

# Appendix

# A Formal Analysis of System Design and Proofs

In this section, we provide further formal analysis of the technical and algorithmic points made in § 4. In § A.1-A.1.2 we discuss the properties of our unit-by-unit Viterbi generation strategy and its advantages over a direct prompting approach. In § A.2 we formalize the structure of block programs $B_{LM}$ and use this structure to justify our GAB tree factorization. Finally, in § A.3.2 we justify our decision to optimize for few-vs-many samples in our GP approach and link this with properties of our Viterbi search strategy.

## A.1 Viterbi-style Search (VS) Proofs

To compare different prompting strategies, we analyze the expected *number of attempts* (and then token costs) of a single-shot *direct-prompting* (direct) approach versus a *Viterbi-style Search* (VS) or our unit-by-unit generation approach. The argument is based on straightforward properties of the geometric distribution, but yields an exponential advantage (Prop. 1) for VS when a design artifact must satisfy multiple constraints.

### A.1.1 Direct vs. Viterbi Approach: Basic Analysis

**Setup** Let $\mathcal{A}$ be the set of possible final artifacts (e.g. fully implemented LM architectures). We consider:

- A **direct** approach that tries to generate an artifact $A$ in *one shot*. If the artifact is invalid (e.g., doesn't pass the checker in Table 6), we discard and try again from scratch.
- A **VS** approach that factorizes the generation into $N$ sequential sub-decisions, each retried upon failure only locally (i.e. we "checkpoint" partial successes).

The following result relates to the expected number of model calls for the direct approach.

> **Lemma 1** (Single-Shot Expected Calls). *Suppose the probability of success (i.e., generating a valid output that satisfies some target constraints) in one single-shot generation is $p_{\text{valid}} \in (0, 1)$. Then the expected number of calls until success (denoted as $\mathbb{E}[\text{calls}_{\text{direct}}]$) is*
>
> $$\mathbb{E}[\text{calls}_{\text{direct}}] = \frac{1}{p_{\text{valid}}},$$
>
> *assuming each call is an i.i.d. Bernoulli($p_{\text{valid}}$) trial.*

*Proof.* This follows directly from the geometric distribution: the probability we succeed on the $k$-th attempt is $(1 - p_{\text{valid}})^{k-1} p_{\text{valid}}$, and the expected number of attempts is $1/p_{\text{valid}}$. $\square$

In many scenarios, success requires $N$ sub-components to be correct simultaneously (e.g., $N$ generated code units all being correct), each with probability $p_k$. Then

$$p_{\text{valid}} \approx \prod_{k=1}^{N} p_k \quad \implies \quad \mathbb{E}[\text{calls}_{\text{direct}}] \approx \frac{1}{\prod_{k=1}^{N} p_k}.$$

**Viterbi-style Unit-based Factorization** In the **VS** approach, we imagine the creation of an artifact as $N$ steps:
$$I_0 \to I_1 \to \cdots \to I_N = A,$$
where each step $(I_{k-1} \to I_k)$ succeeds with probability $p_k$. Failures at step $k$ *do not* discard previously completed steps; we simply revert to $I_{k-1}$ and retry. For this approach, the following holds:

**Lemma 2** (Expected Calls: VS). *If step $k$ has a probability $p_k$ of success on each attempt, the expected total number of calls to complete all $N$ steps is*

$$\mathbb{E}[\text{calls}_{\text{VS}}] = \sum_{k=1}^{N} \frac{1}{p_k}.$$

*Proof.* Step $k$ follows a geometric distribution with success probability $p_k$. Hence, its expected trials are $1/p_k$. Summing over $k = 1, \dots, N$ yields $\sum_{k=1}^{N} 1/p_k$. □

From these two facts, the following follows regarding how the VS approach requires exponentially fewer model calls over the direct approach.

**Proposition 1** (VS vs. Direct: Exponential Gain). *If $p_{\text{valid}} \approx \prod_{k=1}^{N} p_k$, then*

$$\mathbb{E}[\text{calls}_{\text{direct}}] \approx \frac{1}{\prod_{k=1}^{N} p_k}, \quad \mathbb{E}[\text{calls}_{\text{VS}}] = \sum_{k=1}^{N} \frac{1}{p_k}.$$

*In typical cases where $\prod_{k=1}^{N} p_k \ll p_j$ for each $j$, we have $\sum_{k=1}^{N} 1/p_k \ll 1/\prod_{k=1}^{N} p_k$, indicating a potential exponential improvement for VS.*

Then the following corollary follows straightforwardly.

**Corollary 2** (Identical Steps Case). *If $p_k = p$ for all $k$, then*

$$\mathbb{E}[\text{calls}_{\text{direct}}] = \frac{1}{p^N}, \quad \mathbb{E}[\text{calls}_{\text{VS}}] = \frac{N}{p}.$$

*As $N$ grows, $\frac{N/p}{1/p^N} = N\,p^{N-1} \to 0$ (exponentially), showing that the advantage of VS grows dramatically with larger $N$.*

The exponential gain of VS also explains why high-quality samples outweigh vast low-quality ones – it may take exponentially more samples to reach the same optimal point in a VS sample.

### A.1.2 Refined Analysis: Token Costs with Growing History

Given that each model call incurs token costs, an exponential improvement in the number of calls or steps $k$ means that the token costs may reduce exponentially. In this section, we quantify such token costs in terms of the prompting and history tokens that need to be processed during each try. Let:

- $H_k$: the number of "history" input tokens at step $k$.
- $\delta_k$: any additional instructions or new tokens in step $k$.
- $O_k$: the number of output tokens generated by step $k$.
- $c_i, c_o$: cost coefficients for input and output tokens, respectively.

Then the cost of a single attempt of Step $k$ is:

$$\text{Cost}_k = c_i\,(H_k + \delta_k) + c_o\,O_k.$$

Under geometric retries, the expected attempts at step $k$ are $1/p_k$ and the following holds.

**Lemma 3** (Expected Token Cost in VS). *The expected total cost to complete all $N$ steps in VS is*

$$\mathbb{E}[Cost_{VS}] = \sum_{k=1}^{N} \frac{1}{p_k} \Big[ c_i\,(H_k + \delta_k) + c_o\,O_k \Big].$$

*Proof.* At each step $k$, we expect $1/p_k$ attempts. Each attempt incurs $\text{Cost}_k$. Summing over $k$ completes the proof. □

**Comparison to Single-Shot** In a single-shot approach, the probability of success is $\prod_{k=1}^{N} p_k$, so the expected number of attempts is $1/\left(\prod p_k\right)$. Each try regenerates the entire artifact. Let its cost be $\text{Cost}_{\text{full}} = c_i(\dots) + c_o(\dots)$. Then

$$\mathbb{E}[\text{Cost}_{\text{direct}}] \;=\; \frac{1}{\prod_{k=1}^{N} p_k} \;\times\; \text{Cost}_{\text{full}}.$$

Thus, even accounting for partial history $\{H_k\}$ in VS, one can still have:

$$\sum_{k=1}^{N} \frac{\text{Cost}_k}{p_k} \;\ll\; \frac{\text{Cost}_{\text{full}}}{\prod_{k=1}^{N} p_k},$$

especially when $\prod_{k=1}^{N} p_k$ is very small. This confirms that the exponential improvement result extends to token-cost models, not just the raw count of attempts.

**Conclusion (VS)** Our Viterbi-style search can yield an exponential reduction in expected attempts (and potentially in token cost) compared to single-shot approaches, particularly when many sub-decisions need to be correct simultaneously. This underpins the efficiency of Genesys's stepwise *Planner–Coder–Observer* pipeline, where partial successes are preserved.

### A.1.3 Additional Advantage: Extended "Design Tokens" in VS

Beyond merely reducing *number of attempts* (Lemma 2), VS can also *increase* the total amount of "reasoning" or "design tokens" used in the generation of a final artifact, thereby improving its quality. We formalize this idea as follows.

**Setup and Definitions** Consider two approaches for generating a *complex* design artifact $A$:

1. **Direct Generation:** A single pass of length $M_{\text{shot}}$ produces the final design $A$.
2. **VS Generation:** Generation is factorized into $N$ sequential steps (§2), each potentially *adding or refining* partial outputs, with local checks and retries. Let $M_k$ denote the number of tokens used (or produced) at step $k$, and let $M_{\text{VS}} = \sum_{k=1}^{N} M_k$ be the total number of tokens across all steps.

We say that each token that directly contributes to the final design is a *design token*. In the VS approach, multiple partial expansions, corrections, or debugging messages can yield a *larger* corpus of design tokens than in a single-shot approach.

**Assumption 1** (Monotonicity of Quality in Token Budget). *Let $\mathcal{Q}(A)$ be the "quality" (e.g. correctness or score) of a final artifact $A$. Suppose that there is a non-decreasing function $f(\cdot)$ such that the expected quality of an output improves as the number of design tokens grows:*

$$\mathbb{E}[\mathcal{Q}(A) \mid \text{token budget} = m] \;\geq\; f(m).$$

*In particular, $f(m)$ is strictly increasing in $m > 0$ (more design tokens lead, in expectation, to higher-quality artifacts).*

This assumption echoes a widely observed phenomenon: *longer* intermediate reasoning or drafting stages (e.g., "chain-of-thought") can improve correctness for difficult tasks.

**Lemma 4** (VS Allows Strictly More Design Tokens). *Let $M_{\text{shot}}$ be the fixed budget of design tokens in a single-shot approach. In Viterbi-style Search factorization with $N$ steps,*

$$M_{\text{VS}} \;=\; \sum_{k=1}^{N} M_k, \quad M_k \geq 0,$$

*where each $M_k$ may include expansions, partial corrections, or debug logs. Provided the partial checks do not enforce a strict token cap at each step, one can typically satisfy*

$$\mathbb{E}[M_{\text{VS}}] \;>\; M_{\text{shot}},$$

*meaning the* expected *total design tokens used in VS can exceed the single-shot token budget.*

*Sketch.* In single-shot generation, the artifact is created *once*, yielding a total of $M_{\text{shot}}$ design tokens (e.g. the length of the entire output). In contrast, in VS, factorization allows partial expansions and corrections. If any of the $N$ steps fail local validation or require refinement, the retry mechanism may produce *additional* tokens: detailed debug messages, corrective instructions, or iterative expansions. Hence, the total number of generated (and possibly regenerated) design tokens $M_{\text{VS}}$ can exceed $M_{\text{shot}}$. Even with local checks, the system is not obligated to "truncate" expansions at each step, so the *expected* design token count across all steps is strictly larger on average if any fraction of steps require more than a single attempt. □

The following then holds.

**Proposition 3** (Quality Gain from Viterbi-style Search Extended Tokens). *Under Assumption 1, let $\widehat{A}_{\text{shot}}$ be the artifact returned by a single-shot approach with a fixed budget $M_{\text{shot}}$, and let $\widehat{A}_{\text{VS}}$ be the artifact returned by the VS approach with random total tokens $M_{\text{VS}}$. Then:*

$$\mathbb{E}[\mathcal{Q}(\widehat{A}_{\text{VS}})] \ \geq \ \mathbb{E}[\, f(M_{\text{VS}})\,] \ > \ f(M_{\text{shot}}),$$

*whenever $\mathbb{E}[M_{\text{VS}}] > M_{\text{shot}}$ and $f(\cdot)$ is strictly increasing.*

*Proof.* By Lemma 4, in VS, factorization can spend more design tokens in total. Since $f(\cdot)$ is strictly increasing, having a *larger* token count (on average) implies strictly higher *expected* quality. Formally,

$$\mathbb{E}[\, f(M_{\text{VS}})\,] \ \geq \ f\big(\mathbb{E}[M_{\text{VS}}]\big) \ > \ f\big(M_{\text{shot}}\big),$$

where the first inequality follows from Jensen's inequality if $f$ is convex and increasing,[3] and the second strict inequality follows by $\mathbb{E}[M_{\text{VS}}] > M_{\text{shot}}$. Hence, the expected quality under VS is greater than under a single-shot approach constrained to $M_{\text{shot}}$ tokens. □

**Interpretation** In practice, difficult designs or proofs often benefit from iterative expansions, re-checks, or "chain-of-thought" style reasoning. In VS, factorization *naturally* accommodates these partial expansions, producing a greater quantity of design tokens. If one presumes that additional design tokens correlate with more thorough reasoning—and thus higher *quality*—then Proposition 3 shows a theoretical justification for the quality advantage of multi-checkpoint VS over single-shot generation. "More tokens" here refers specifically to *effective design content*, including debug corrections or expansions that shape the final artifact, rather than random filler text. As a result, factorizing generation (rather than forcing a single, short pass) can yield both *exponential savings in attempts* (§2) *and* an *increase in solution quality* (§3), making VS highly advantageous in complex, multi-component discovery tasks.

## A.2 Unit Tree Factorization Proofs

As noted in § 4.1 and shown in Figure 3 (via the type annotations), code blocks $B_{LM}$ naturally consist of compositions of units (e.g., multiple head attention, GatedMLP) each with type $(X, Z) \to (X, Z)$ (below we generalize this to the type mapping $\Sigma \to \Sigma$). Below we use this type structure to justify our particular GAB factorization, showing specifically that any program $P : \Sigma \to \Sigma$ in the category of GAB programs can be factorized into a finite tree of sub-blocks (i.e., the kinds of unit-based representations we use).

### A.2.1 Case 1: $\Sigma \to \Sigma$ Programs

**Definition 1** (Unit Tree for $\Sigma \to \Sigma$). *Suppose $\mathcal{L}_\Sigma$ is a typed language closed under composition and identity on $\Sigma$. A* unit tree *for a program $P : \Sigma \to \Sigma$ is a finite, rooted tree $T$ such that:*

    *1. Each node is labeled by a subprogram $Q : \Sigma \to \Sigma$ in $\mathcal{L}_\Sigma$.*

---

[3]If $f$ is just non-decreasing, we have $\mathbb{E}[f(M_{\text{VS}})] \geq f(\inf M_{\text{VS}})$. In practice, partial expansions typically ensure $\inf M_{\text{VS}} \geq M_{\text{shot}}$.

2. *Leaves are* atomic *or* empty/identity. *(In practice, we do not expand or decompose a program into this level, which makes a unit tree degrade into an AST.)*

3. *An internal node labeled P factors as $P = P_1 \circ P_2 \circ \cdots \circ P_k$, where each $P_i : \Sigma \to \Sigma$.*

The following then follows and ensures we can always decompose the units in a single large GAB block into smaller $\Sigma \to \Sigma$ sub-blocks, enabling targeted GP operators.

**Theorem 4** (Unit Tree Factorization for $\Sigma \to \Sigma$)**.** *Let $\mathcal{L}_\Sigma$ be closed under composition and identity in $\Sigma$. Then every program*

$$P : \Sigma \to \Sigma, \quad P \in \mathcal{L}_\Sigma,$$

*admits a finite unit tree $T = \Phi(P)$ in the sense of Definition 1.*

*Proof.* **Base Case.** If $P$ is *atomic* or the *identity* map, we define $\Phi(P)$ to be a single-node tree labeled by $P$.

**Recursive Case.** If $P$ can be expressed as $P = P_1 \circ \cdots \circ P_k$, with each $P_i \in \mathcal{L}_\Sigma$, then by induction each $P_i$ has a tree $T_i = \Phi(P_i)$. We form $\Phi(P)$ by adding a root node (labeled $P$) with children $T_1, \ldots, T_k$. Closure under composition ensures this remains in $\mathcal{L}_\Sigma$.

**Termination.** A syntactically finite program eventually decomposes into atomic or identity forms, guaranteeing a finite tree. $\qquad\square$

### A.2.2   Case 2: Extending to General Typed Programs via Type-Lifting

In practice, many programs do not preserve the same shape or type. For example, in real block designs, the skip and residual connections involve mappings of type $Q : X \to X$, and hence do not fit in the language defined above. Below, we show how to embed (lift) $Q$ into a $\Sigma \to \Sigma$ function, then apply Theorem 4. Importantly, this shows how our factorization, as well as our broader GP search, can be extended to problems with different type structures.

**Universal Type $\Sigma$**   Assume that we have encoders and decoders:
$$\text{Enc}_X : X \to \Sigma, \quad \text{Dec}_X : \Sigma \to X, \quad \text{Enc}_Y : Y \to \Sigma, \quad \text{Dec}_Y : \Sigma \to Y,$$
such that $\text{Dec}_X(\text{Enc}_X(x)) = x$ for all $x \in X$. Then we define:

**Definition 2** (Lifted Function $\widetilde{Q}$)**.** *Given $Q : X \to Y$, its* lifted *version $\widetilde{Q} : \Sigma \to \Sigma$ is:*
$$\widetilde{Q}(s) \;=\; \text{Enc}_Y\Big[Q\big(\text{Dec}_X(s)\big)\Big].$$

The following then establishes that we can preserve the type mapping $\Sigma \to \Sigma$ using type raising.

**Proposition 5** (Unit Tree Factorization for General $Q : X \to Y$)**.** *Let $\mathcal{L}$ be closed under composition. Then any $Q : X \to Y$ can be lifted to $\widetilde{Q} : \Sigma \to \Sigma$ (per Definition 2), and by Theorem 4, $\widetilde{Q}$ admits a unit tree in $\mathcal{L}_\Sigma$. This induces a corresponding decomposition of the original $Q$.*

*Proof.* Because $\mathcal{L}$ is closed under composition, $\widetilde{Q} = \text{Enc}_Y \circ Q \circ \text{Dec}_X \in \mathcal{L}_\Sigma$. By Theorem 4, $\widetilde{Q}$ factors into a finite tree of $\Sigma \to \Sigma$ sub-blocks. Those sub-blocks, when "projected" back through $\text{Dec}_X$ and $\text{Enc}_Y$, yield a valid decomposition of $Q$. $\qquad\square$

**Designing $\Sigma$ in Practice**

- *Overhead vs. Gains*: Merging $X$ and $Y$ into a single $\Sigma$ can increase memory or prompt size. However, partial or selective factorization can mitigate overhead.
- *Atomic Black Boxes*: If certain submodules are not to be searched or mutated, we can treat them as atomic.
- *Recursion, Higher-Order Functions*: If $Q$ returns a function or is unboundedly recursive, partial unrolling or bounding is required for a finite tree.

**Conclusion**  We have shown that any $\Sigma \to \Sigma$ program can be factorized into composable sub-blocks (Theorem 4), and that one can lift a function $Q : X \to Y$ into $\widetilde{Q} : \Sigma \to \Sigma$ (Proposition 5). This generalizes the unit tree factorization approach well beyond autoregressive shapes, allowing Genesys to apply *genetic programming* (GP) operations on arbitrary programs while still benefiting from the **efficiency** of Viterbi-style Search discussed in Section A.1.

## A.3  Evolution Efficiency of Genesys: Few High-Quality Samples with VS

We now unify two key ideas behind *Genesys*'s efficiency:

1. **Few vs. Many Samples:** A smaller number of *high-quality* (and valid) samples can yield *more* improvements than a large number of low-quality trials, given a fixed cost budget.

2. **Viterbi-style Search (VS) Exponential Advantage:** Factorizing the design process into multiple sequential steps (each retried locally) exponentially reduces the expected attempts to produce a valid final artifact, compared to a single-shot (direct) approach that must get every sub-component correct in one go.

By combining these points, we show that Genesys's approach—focusing on more *careful, iterative* code generation with local checkpoints (VS)—further magnifies the benefit of "few, high-quality samples" over "vast, low-quality trials."

### A.3.1  Few High-Quality Samples Outweigh Vast Low-Quality Trials

**Setup and Yield**  Let:

- $Q \in [0, 1]$: Probability a newly generated design is a *beneficial improvement* over the current best or population.

- $E \in [0, 1]$: Probability that the design is *valid* (e.g. compiles, passes checks, etc.).

- $c > 0$: Average *cost per sample* (e.g., tokens or GPU hours per generation).

- $B > 0$: Total *budget* in the same cost units.

Hence, the maximum number of samples is $N = \frac{B}{c}$. Only a fraction $Q \times E$ of these $N$ samples will be valid *and* an improvement. Defining $r = Q \times E$, *expected yield* is:

$$Y \;=\; r \;\times\; \frac{B}{c} \;=\; \left(Q\,E\right)\frac{B}{c}.$$

If *Strategy 1* has parameters $(Q_1, E_1, c_1)$ and *Strategy 2* has $(Q_2, E_2, c_2)$, both under the same total budget $B$, then:

> **Proposition 6** (Few High-Quality Samples Outweigh Vast Low-Quality Trials)**.**
>
> $$\frac{Q_1\,E_1}{c_1} \;>\; \frac{Q_2\,E_2}{c_2} \quad\Longrightarrow\quad Y_1 \;>\; Y_2, \quad \text{where } Y_i \;=\; Q_i E_i\,\frac{B}{c_i}.$$
>
> Interpretation: *Even if Strategy 1 generates* fewer *samples (larger $c_1$), it can yield* more *total improvements, provided each sample is sufficiently more likely to be valid and beneficial.*

*Proof.*  The proof is a simple rearrangement:

$$Y_1 \;=\; (Q_1\,E_1)\,\frac{B}{c_1}, \quad Y_2 \;=\; (Q_2\,E_2)\,\frac{B}{c_2}. \quad Y_1 > Y_2 \;\Longleftrightarrow\; \frac{Q_1 E_1}{c_1} \;>\; \frac{Q_2 E_2}{c_2}.$$

$\square$

### A.3.2  Combining VS with the "Few High-Quality Samples" Argument

**VS Exponentially Increases Validity in Complex Designs**  Recall Proposition 1 in Appendix A.1: if an artifact has $N$ sub-components, each with success probability $p_k$, then a *single-shot* approach must succeed simultaneously with probability $\prod_{k=1}^{N} p_k$, which can be extremely small. By contrast,

a **Viterbi-style Search (VS)** scheme that checkpoints partial progress has an expected total number of calls only $\sum_{k=1}^{N} \frac{1}{p_k}$ rather than $1/\left(\prod_{k=1}^{N} p_k\right)$, yielding an *exponential* improvement for large $N$. Thus, under VS, the *effective validity* $E_{\text{VS}}$ (chance of eventually producing a correct artifact) can be far larger than $p_{\text{direct}} = \prod_{k=1}^{N} p_k$.

**Genesys Achieves a Higher $\frac{QE}{c}$ Term**   In Genesys, *VS* drastically increases $E$ for complex designs by preserving partial successes, while the *literature-based designer* and evolutionary selection raise $Q$ (the chance that a new design is genuinely beneficial). Even though each Genesys attempt costs somewhat more (raising $c$), the net effect can still be $\frac{QE}{c} \gg \frac{Q \prod p_k}{c_{\text{naive}}}$. Hence, by Proposition 6, Genesys can produce *more* total improvements under a fixed budget $B$.

> **Proposition 7** (Genesys's VS Increases $\frac{QE}{c}$). *If $p_{\text{direct}} = \prod_{k=1}^{N} p_k$ is the single-shot validity probability for an $N$-component artifact, and $E_{\text{VS}}$ is the probability of success via Viterbi-style Search, then typically $E_{\text{VS}} \gg p_{\text{direct}}$. As long as $c_{\text{VS}}$ (the cost per Genesys attempt) does not grow exponentially in $N$, the ratio $\frac{Q\,E_{\text{VS}}}{c_{\text{VS}}}$ can be* exponentially *greater than $\frac{Q \prod_{k=1}^{N} p_k}{c_{\text{naive}}}$, leading to higher yield $Y$.*

**Additional Quality Advantage of VS**   Beyond boosting validity $E$, the stepwise factorization in VS also *increases* the "design tokens" and iterative refinements per sample (§A.1.3). This can further raise $Q$ (the chance of a beneficial improvement) by allowing more debugging, partial expansions, or chain-of-thought. Combined, these effects further enlarge $\frac{QE}{c}$.

**Conclusion**   By merging the "few high-quality samples" principle with the *exponential* gain in validity from VS, Genesys obtains a higher $\frac{QE}{c}$ and thus a higher yield $Y = (Q\,E)\,(B/c)$. Even if Genesys attempts *fewer* samples, each has a significantly greater probability of (1) being valid (via factorized re-tries) and (2) being beneficial (via literature grounding and evolutionary selection). Empirically, this leads to more successful discoveries than approaches that generate many low-quality trials. "More tokens" also tend to enable more sophisticated reasoning, further increasing the probability that a design is *beneficial*. Thus, the synergy of **quality improvement** (raised $Q$) *and* **validity improvement** (raised $E$) explains why Genesys can be highly efficient despite producing fewer with more carefully crafted samples.

# B   Implementation Details

In this Section, we provide additional details of the LMADE components in §B.1, succinctly discuss the implementation with pseudo codes in §B.2, and conclude with the GAB, GAU, and the LM base class and templates in §B.3.

## B.1   LMADE Component Details

### B.1.1   Reference Library

We manually constructed a reference library of pivotal innovations in Transformer and alternative architectures. Besides seminal works like GPT, we manually chose papers from the last three years of leading conferences (e.g., ICLR, ICML, NeurIPS), and prominent arXiv publications based on citations or social media discourse, an increasingly prevalent means of academic dissemination. The survey papers Tay et al. (2022b); Wan et al. (2024) and the community GitHub resources cited in Table 5 served as foundational references. We exclude the work in these directions: 1) Distillation or non-standard training methods; 2) Hardware-specific optimizations, such as GPU-level optimizations or quantization; 3) Caching, or other efficiency improvements. 4) Inference-stage methods; 5) Application-specific optimizations (e.g., for finance or healthcare); 6) Audio or video processing techniques; 7) Post-training enhancements such as fine-tuning; 8) Methods based on parameter sharing. We compiled 297 reference designs. Metadata like titles, authors, and abstracts were retrieved via S2, forming *reference* nodes in the EvoTree, connected based on the citation of each other.

| Repository | Description |
|---|---|
| fla-org/flash-linear-attention | *A collection of state-of-the-art linear attention models.* |
| LAION-AI/lucidrains-projects | *Projects created by lucidrains about transformers.* |
| Xnhyacinth/Awesome-LLM-Long-Context-Modeling | *Must-read papers and blogs on LLM-based Long Context Modeling.* |
| Event-AHU/Mamba_State_Space_Model_Paper_List | *Paper list for State-Space-Model and its Applications.* |
| yyyujintang/Awesome-Mamba-Papers | *This repository compiles a list of papers related to Mamba and SSM.* |
| XiudingCai/Awesome-Mamba-Collection | *A curated collection of resources related to Mamba.* |

Table 5: Github repos we referred to when building the reference library.

| Component | Description of test | static | execution |
|---|---|:---:|:---:|
| parser | Checks that block is syntactically valid (AST-based). | ✓ | |
| formatter | Checks that block follows GAB protocol (Fig. 3). | ✓ | |
| initialization | Checks that the PyTorch module can be initialized. | | ✓ |
| forward | Checks that forward pass can be performed. | | ✓ |
| backward | Checks that backward pass can be performed. | | ✓ |
| causality | Checks that block employs causal masking. | | ✓ |
| differentiability | Checks that module is differentiable and doesn't involve unused parameters. | | ✓ |
| effectiveness | Checks for correct training behavior on a small corpus, e.g., stable gradients, loss convergence, reasonable flops. | | ✓ |

Table 6: *How do we check if a block design $B_{LM}$ is valid?* A description of our **Symbolic checker** in LMADE that performs **static**- and **execution**-based code analysis to determine code validity.

We manually find their implementations, 185 out of them have released available code base, we select 5 most typical designs as seed designs to initialize the EvoTree where each of them represents a popular or novel architectural idea: **GPT** Brown et al. (2020) is the most popular Transformer-based architecture; **Mamba2** Dao & Gu (2024) represents the State Space Machines and Linear Attention models; **RWKV6** Peng et al. (2024) represents the latest progress on modern RNNs; **RetNet** Sun et al. (2023) explores the balance among Transformers, RNNs, and Linear Attention models; **TTT** Sun et al. (2024) represents a novel idea of test-time training. For the other 180 designs, we manually extract the LM block implementation or core implementations of their proposed method from the released code base and store them in the node data. When a reference is selected, the metadata, as well as the code, if any, will be provided as part of the prompt.

### B.1.2 Symbolic Checkers

We develop a symbolic checker to check the validity of a design without performing the costly actual verification process. The components are listed in Table 6. It can be roughly divided into the **static format checks** based on Abstract Syntax Tree (AST) traversal which mainly checks if the code follows the format of GAU and GAB, and fix some simple errors like not passing the dtype and devices, not using required arguments; and the **Runtime functional checks**, which tries to initialize the corresponding PyTorch model, then check its forward and backward pass, differentiability of parameters, as well as its causality by examing: given a sequence $X$ with length $L$, whether $Y = f(X[1 : t])$ changes by changing $X[t + 1 : L]$ for $t$ from 1 to $L - 1$. It also launches a quick training with 10 gradient steps on the Wikitext-2 dataset, then checks if the gradient norm exploding, if the loss is decreasing, and if the training time and FLOPs are 5 times higher than a GPT model trained in the current machine, whose training statistics is automatically tested and stored in a benchmarking report to compare with. § E.2.2 analyzes the distribution of errors detected by the symbolic checkers.

***Early Termination of Implementation:*** A unit is accepted and the implementation state advances to $T^{t+1}$ only if it passes both the checker and the observer. Otherwise, rollback to $T^t$ and retry this step. After $K_{fails}$ failures, the agent ceases effort and will re-implement this proposal at a later time. A proposal may be abandoned up to $K_{attempts}$ times before it is deemed "unimplementable".

### B.1.3 Knowledge Engine

As shown in Fig. 1, the Knowledge Engine contains three modules, the *External Sources* for the literature search, the general *Web Search*, and the *Paper Vector DB* for the internal Reference Library as discussed above in § B.1.1. When querying the Knowledge Engine, the agent needs to fill in three fields: the keywords, a description of the intended content, and the instructions for the web search agents. The keywords were used to query the external sources, while the description was applied to locate relevant excerpts from the paper vector DB. Missing keywords or descriptions will lead to the skipping of the external sources search and the paper vector DB search, respectively.

**External Sources**   We search for papers after 2015 from the top ML or NLP conferences, including NeurIPS, ICML, ICLR, ACL, EMNLP and NAACL from S2. In arXiv, we filter the results by domains: Machine Learning (`cs.LG`) and Computation and Language (`cs.CL`). For PapersWithCode, we do not set a filter, and we request both paper and repo.

**Paper Vector DB**   We downloaded the paper PDFs for the papers in the reference library, and converted them into text by MathPix, which can accurately convert mathematical content into plain text, then split them into chunks with the `SemanticChunker` from LangChain [4] which breaks down the text into semantically different chunks by analyzing the gradients of distances of chunks computed with the OpenAI `text-embedding-3-large` embedding model. We embed each chunk with the same embedding model in vectors and then store them in the Pinecone vector store [5]. The vector db will be available to use by the Knowledge Engine. When retrieving, we apply a Cohere [6] `rerank-english-v3.0` reranker to filter the top 20% most relevant results.

**Web Search**   We use Perplexity.ai for the web search, which is an LLM-based search engine. It accepts natural language queries as input and returns a response containing a summary of search results with references from the websites. We select `llama-3.1-sonar-large-128k-online` as the base model with a maximal number of completion tokens set to 4000. We apply the following system prompt:

> **System Prompt for Peplexity.ai**
>
> You are an AI research assistant who helps a language model researcher gather information for discovering the best novel autoregressive LM block that can defeat the existing state-of-the-art models.
>
> ## Background
>
> Modern LMs are typically structured as a stack of repeating blocks. The goal is to design a novel LM block that outperforms current state-of-the-art models, aiming for:
> - Low perplexity on corpora,
> - High accuracy on downstream tasks,
> - Robustness to varied inputs,
> - Efficiency in both training and inference,
> - Excellent scalability with more data and larger models.
>
> You will be provided with the researcher's thoughts, analysis, and descriptions, and your task is to understand the intent of the researcher and search for the information that can best help the intent.

---

[4] `https://python.langchain.com/docs/how_to/semantic-chunker/`,
[5] `https://www.pinecone.io/`
[6] `https://cohere.com/`

We use the following prompt to pass a *query* to the model:



**Prompt for Peplexity.ai Query**

Here is the information from the researcher:
{query}
Understand the goal, idea, and intent of the researcher. Find the most useful information that can best help the researcher to achieve the goal.



**Interface** When querying the Knowledge Engine, the agent needs to fill in three fields: the keywords, a description of the intended content, and the instructions for the web search agents. The keywords were used to query the external sources, while the description was applied to locate relevant excerpts from the paper vector DB. Missing keywords or descriptions will lead to the skipping of the external sources search and the paper vector DB search, respectively. The instruction is fed to Perplexity.ai for web search. If the instruction is missing, the other non-empty fields will be provided to the agent as the query. The composed results from all sources are returned

### B.1.4 Verification Engine

| component | description |
|---|---|
| check # parameters + tuning | *Checking model has appropriate # parameters, and tunes parameters to fit model scale.* |
| gradient accumulation steps | *Tune gradient accumulation steps to avoid OOM.* |

Table 7: Auto-Tuner components.

**Auto-Tuner** At the beginning of the verification process, as presented in Table 7, we use an auto-tuner to guarantee the model size fits the scale and decide on gradient accumulation steps that do not trigger an Out-Of-Memory (OOM) error in the current machine. A block loader automatically fetches the GAB and composes the LM, then uses this Auto-Tuner to do pre-verification checks and tuning.

***Tuning model size:*** We tune the model size by adjusting the two standard arguments of GAB, which are detailed in § B.3, $num\_block$ and $embed\_dim$. We apply a simple depth-first strategy as per Tang et al. (2024), which claims that depth ($num\_block$) provides more performance gain than width ($embed\_dim$). For each scale $s$, we take the non-embedding parameter number of GPT $P_s$ as a reference, tuning the parameter until the model size $M$ falls into the region $0.8P_s < M < 1.2P_s$. It first tunes the $num\_block$, starting with 1 and gradually increasing until 1. The size fits the region, 2. the size exceeds, or 3. tries for more than 1000 times. If exceeding, the tuner will try to tune the $embed\_dim$ by gradually reducing, every time reducing 16, the $embed\_dim$ may cause an error as some operation may depend on the $embed\_dim$ (e.g., attention heads), thus we will check the forward pass in every attempt, we tune it until the size 1. fits the region, 2. the size smaller than the lower bound. If smaller, the tuner gives up and reports an error.

***Finding gradient accumulation steps:*** We tune the gradient accumulation steps as it theoretically does not influence the training process compared to batch size, which can also overcome the OOM issue. We tune it with a fast, test training of 10 gradient steps on `wikitext-2`, we start from 1 and iteratively double it until no OOM error is triggered.

Once the tuning is completed, the tuned model and parameters are passed to the trainer for the next steps.

**Trainer and Evaluator** We use a Huggingface trainer to train the model. Once a model passes checks and tunes from the Auto-Tuner, the trainer launches the training and reports progress to the Weight & Biases [7]. We use the LM-Eval framework [8] to evaluate the downstream performance of trained models. Once training is complete, the trainer passes the model to the LM-Eval, which then automatically runs the evaluations and returns the report.

---

[7]https://wandb.ai
[8]https://github.com/EleutherAI/lm-evaluation-harness

| component | description |
|---|---|
| Grad norm monitor | *If the grad norm is too high ($> 1e4$).* |
| Loss monitor | *The the loss is exploding ($> 1e4$) or vanishing ($\leq 0$)* |
| Step time monitor | *If the step time is too high (around 10 times) compared to a reference GPT model with the same scale.* |
| Exception handler | *Monitoring if there is any errors occur throughout the verification process.* |

Table 8: Auto-Tuner components.

**Runtime Checker** As presented in 8, we apply a runtime checker to monitor the entire verification process, specifically, we monitor the gradient norm, loss, and step time for every training step, besides, we catch any error that occurs during the whole process, if any of these problems are caught, the verification process will be terminated and the design will be recorded and marked as erroneous in this V-Node and will not be selected in this node for verification. As some errors happen due to environmental settings or unexpected situations in the node (e.g., being preempted, connection lost), only if a design is marked by more than three V-Nodes as erroneous, it will be recognized as an erroneous design. Table 2 reports the runtime error rate of different evolution setups.

### B.1.5 Additional Details of Designer Agent

**Self unit tests and debugging assistance** We force the agent to generate at least one unit test for each unit implementation, the unit tests are decorated with `@gau_test` for being able to be detected by the checker. The checker will run the unit tests and catch any output and results, then bring them back as part of the check report. We also encourage the agent to write assertions and assistive prints to help it debug the code; all outputs will be caught and returned to the agent.

**Hybrid foundational models** Instead of choosing a fixed foundation model for each agent (i.e., Proposer, Reviewer, Planner, Coder, and Observer), we decide on distributions of models for agents (e.g., 0.7 for GPT-4o and 0.3 for Claude-3.5 Sonnet); a different agent may have different distributions. Before a design task, the models for the agents are randomly sampled based on these distributions.

**Internal Unit and Proposal Search** As discussed in §4.2, we allow the reviewer and observer to search from the previous proposals and units to check for self-replication. We store all the proposals and unit codes along with the documentation in a library that can be queried by comparing the cosine distance between the embedding of the query proposal/unit code and the items in the library; the ones with the shortest distances would be returned. In addition, we also return the sibling proposals that are based on the exact same parents, with the query, if any, we randomly select at most two siblings.

## B.2 Pseudo Code

In this section, we provide the extended algorithmic details of our designers and different components of Genesys.

**Algorithm 1** The Design Process (DESIGNMODEL)

**Input:** $EvoTree$ (the current evolutionary tree of designs), $Library$ (reference library)
**Output:** $proposal$ (high-level design proposal), $implementation$ (the final LM code)
**Function** $Propose(EvoTree, Library)$:
 0: **for** $k \leftarrow 1$ **to** $K_{attempts}$ **do**
 0:   $\pi \leftarrow$ None    // no proposal yet
 0:   **for** $i \leftarrow 1$ **to** $MAX\_ROUNDS$ **do**
 0:     $\pi \leftarrow$ PROPOSER.SEARCHANDREFINE$(EvoTree, Library, \pi)$
 0:   **end for**
 0:   $\rho \leftarrow$ None    // no review yet
 0:   **for** $i \leftarrow 1$ **to** $MAX\_ROUNDS$ **do**
 0:     $\rho \leftarrow$ REVIEWER.SEARCHANDREFINE$(EvoTree, Library, \pi, \rho)$
 0:   **end for**
 0:   **if** $\rho.rating \geq THRESHOLD$ **then**
 0:     **return** $(\pi, \rho)$    // accept proposal
 0:   **end if**
 0: **end for**
 0: **raise** "Failed to propose a valid design"
**Function** $DesignModel(EvoTree, Library)$:
 0: $proposal \leftarrow Propose(EvoTree, Library)$
 0: $implementation \leftarrow$ IMPLEMENT$(EvoTree, proposal)$
 0: **return** $(proposal, implementation)$

---

**Algorithm 2** Compose GAU Tree To GAB Code

**Input:** $GAUTree$ (the hierarchical GAU structure)
**Output:** $gabCode$ (a string or code object representing the final composed GAB)
**Function** $ComposeToCode(GAUTree)$:
 0: **let** $gabCode \leftarrow$ INITIALIZEROOTGAB$(GAUTree.\text{root})$ //Start with a minimal GAB that calls the root unit
 0: **let** $units \leftarrow$ TopologicalOrder$(GAUTree)$ // or any valid traversal for sub-units
 0: **for each** $unit$ in $units$ **do**
 0:   **if** $unit.\text{isImplemented} =$ True **then**
 0:     $gabCode +=$ codeOf$(unit)$
 0:   **else**
 0:     $gabCode +=$ placeholderCode$(unit)$
 0:   **end if**
 0: **end for**
 0: **return** $gabCode$

**Algorithm 3** Viterbi-style Search Implementation with GP

---

**Input:** *Parents* (one or more parent designs, or empty), *Proposal* (the high-level design), *EvoTree* (for references / re-use)
**Output:** A valid GAB implementation (GAU tree + code)
**Function** *Implement*($EvoTree$, $Proposal$):
 0: **if** $|Parents| = 1$ **then**
 0:   $GAUTree \leftarrow$ CopyOf($Parents[0]$)
 0:   $unimplemented \leftarrow [Proposal.\text{SelectedSubtree}]$ // for MUTATION
 0: **else if** $|Parents| > 1$ **then**
 0:   $GAUTree \leftarrow$ NEWGAUTREE(empty root)
 0:   $unimplemented \leftarrow [GAUTree.\text{root}]$ // for CROSSOVER
 0: **else**
 0:   $GAUTree \leftarrow$ NEWGAUTREE(empty root) // from DESIGN FROM SCRATCH
 0:   $unimplemented \leftarrow [GAUTree.\text{root}]$
 0: **end if**
 0: **while** $unimplemented \neq []$ **do**
 0:   $(plan, unit) \leftarrow$ PLANNER($GAUTree'$, $unit$, $Proposal$)
 0:   **for** $tries \leftarrow 1$ **to** $K_{fails}$ **do**
 0:     $GAUTree' \leftarrow$ CopyOf($GAUTree$)
 0:     **if** $|Parents| > 1$ **then**
 0:       $reusePool \leftarrow$ ALLGAUUNITS($Parents$)
 0:       $forceReuse \leftarrow$ True
 0:     **else**
 0:       $reusePool \leftarrow$ RECOMMENDREUSEUNITS($EvoTree$, $unit.\text{description}$)
 0:       $forceReuse \leftarrow$ False
 0:     **end if**
 0:     $(code, children) \leftarrow$ CODER($GAUTree'$, $unit$, $plan$, $Proposal$, $reusePool$, $forceReuse$)
 0:     **if** $\neg$FORMATCHECKER($code$) **then**
 0:       **continue** // retry if formatting failed
 0:     **end if**
 0:     $GAUTree'.\text{update}(unit, code, children)$
 0:     $gabCode \leftarrow$ COMPOSETOCODE($GAUTree'$)
 0:     **if** FUNCTIONALITYCHECKER($gabCode$) $\wedge$ OBSERVER($code$, $GAUTree'$, $Proposal$) **then**
 0:       $GAUTree \leftarrow GAUTree'$ // accept changes
 0:       $unimplemented.\text{extend}(children)$
 0:       **break** // proceed to the next unit
 0:     **end if**
 0:   **end for**
 0: **end while**
 0: **return** $GAUTree$

---

**Algorithm 4** Quadrant-Based Selection with Scheduler

---

**Input:** *EvoTree* (set of designs, each with fitness & confidence), *Mode* (e.g., *Design* or *Verify*), $pRandom(t)$ (scheduler probability of picking from initial seeds), $pExplore$ (probability of choosing exploration quadrant), $K$ (size for top-$K$ selection), $t$ (current iteration or time step)

**Output:** A selected design from *EvoTree*

**Function** *QuadrantSelect(EvoTree, Mode, pRandom(t), pExplore, K)*:

 0: **comment** // 1. Possibly pick from initial seeds (random)
 0: **sample** $r \leftarrow$ Uniform$(0, 1)$
 0: **if** $r \leq pRandom(t)$ **then**
 0:    **return** RandomlyPickSeed$(EvoTree)$
 0: **end if**
 0: **comment** // 2. Compute medians for fitness & confidence
 0: $F_{\mathrm{med}} \leftarrow$ Median$\big\{d.\text{fitness} \mid d \in EvoTree\big\}$
 0: $C_{\mathrm{med}} \leftarrow$ Median$\big\{d.\text{confidence} \mid d \in EvoTree\big\}$
 0: **comment** // 3. Partition into quadrants
 0: $Q_1, Q_2, Q_3, Q_4 \leftarrow \emptyset$
 0: **for each** design $d$ in *EvoTree* **do**
 0:    **if** $(d.\text{fitness} \geq F_{\mathrm{med}})$ **and** $(d.\text{confidence} \geq C_{\mathrm{med}})$:
 0:       $Q_1.\text{add}(d)$   // good & confident
 0:    **else if** $(d.\text{fitness} \geq F_{\mathrm{med}})$ **and** $(d.\text{confidence} < C_{\mathrm{med}})$:
 0:       $Q_2.\text{add}(d)$   // good & not confident
 0:    **else if** $(d.\text{fitness} < F_{\mathrm{med}})$ **and** $(d.\text{confidence} \geq C_{\mathrm{med}})$:
 0:       $Q_3.\text{add}(d)$   // poor & confident
 0:    **else**
 0:       $Q_4.\text{add}(d)$   // poor & not confident
 0: **end for**
 0: **comment** // 4. Choose quadrant(s) based on the Mode
 0: **sample** $r \leftarrow$ Uniform$(0, 1)$
 0: **if** $r \leq pExplore$:
 0:    $Q \leftarrow Q_3$ **if** $Mode =$ "Design" **else** $Q_4$
 0: **else**
 0:    $Q \leftarrow Q_1$ **if** $Mode =$ "Design" **else** $Q_2$
 0: **end if**
 0: **comment** // 5. Top-$K$ style selection with small-prob picking from outside top-$K$
 0: $Q.\text{sortDescendingByFitness}()$ // or by other priority
 0: $chosen \leftarrow \emptyset$ // we may pick multiple or just 1
 0: **for** $i \leftarrow 1$ **to** $K$ **do**
 0:    **sample** $p \leftarrow$ Uniform$(0, 1)$
 0:    **if** $p < \alpha$ // small $\alpha$, e.g. 0.1
 0:       // pick from outside top-$K$ in $Q$
 0:       $d \leftarrow$ RandomFrom$(Q[K{:}])$
 0:    **else**
 0:       // pick next from top-$K$ portion
 0:       $d \leftarrow Q[i]$
 0:    **end if**
 0:    $chosen.\text{add}(d)$
 0: **end for**
 0: **return** $chosen$

---

**Algorithm 5** Ladder of Scales Budget Control and Scale Selection

---

**Input:** *usedBudgets* (dict: scale ↦ used verifications), *verifyBudgets* (dict: scale ↦ total budget), *EvoTree* (for usage info), *design* (the current design to verify)

**Output:** A chosen scale, or None if none is available

**Function** *AssignLoSBudgets*( *usedBudgets*, *verifyBudgets* ):
- 0:     **comment** // 1. Identify the lowest scale, e.g., "14M"
- 0:     *lowestScale* ← FINDLOWESTSCALE(*verifyBudgets*)
- 0:     *lowestScaleUsed* ← *usedBudgets*[*lowestScale*]
- 0:     **comment** // 2. Initialize local data structures
- 0:     *loSBudgets* ← {}    // Desired usage at each scale
- 0:     *availableBudget* ← {}
- 0:     *loSBudgets*[*lowestScale*] ← *lowestScaleUsed*
- 0:     **if** *loSBudgets*[*lowestScale*] > *verifyBudgets*[*lowestScale*]
- 0:         *availableBudget*[*lowestScale*] ← 0
- 0:     **else**
- 0:         *availableBudget*[*lowestScale*] ← 1
- 0:     **end if**
- 0:     **comment** // 3. Sort scales ascending (e.g., 14M < 31M < 70M)
- 0:     *scales* ← SORTEDSCALES(*verifyBudgets*)
- 0:     **comment** // 4. Build usage for higher scales with a selection ratio
- 0:     **for** *i* **from** 0 **to** (len(*scales*) − 2) **do**
- 0:         *currScale* ← *scales*[*i*]
- 0:         *nextScale* ← *scales*[*i* + 1]
- 0:         $selectRatio \leftarrow \dfrac{verifyBudgets[nextScale]}{verifyBudgets[currScale]}$
- 0:         *loSBudgets*[*nextScale*] ← int(*loSBudgets*[*currScale*] × *selectRatio*)
- 0:         **if** *loSBudgets*[*nextScale*] > *verifyBudgets*[*nextScale*]
- 0:             *availableBudget*[*nextScale*] ← 0
- 0:         **else**
- 0:             *availableBudget*[*nextScale*] ← *loSBudgets*[*nextScale*] − *usedBudgets*[*nextScale*]
- 0:         **end if**
- 0:     **end for**
- 0:     **return** *availableBudget*

**Function** *SelectScale*( *design*, *EvoTree*, *verifyBudgets* ):
- 0:     **comment** // 1. Gather usage info
- 0:     *usedBudgets* ← *EvoTree*.usedBudget
- 0:     *availableBudget* ← *AssignLoSBudgets*(*usedBudgets*, *verifyBudgets*)
- 0:     **comment** // 2. Find which scales the design has *not* verified
- 0:     *verifiedScales* ← *EvoTree*.getVerifiedScales(*design*)
- 0:     *unverifiedScales* ← ALLSCALES(*verifyBudgets*) \ *verifiedScales*
- 0:     **comment** // 3. Among the available, pick the lowest unverified
- 0:     *candidateScales* ← { *s* | *s* ∈ *availableBudget*, *availableBudget*[*s*] > 0 } ∩ *unverifiedScales*
- 0:     **if** len(*candidateScales*) = 0
- 0:         **return** None
- 0:     **else**
- 0:         **return** FINDLOWESTSCALE(*candidateScales*)
- 0:     **end if**

---

**Algorithm 6** Genesys Evolutionary Loop

---

**Input:** *EvoTree* (population of designs), *Library* (reference library), *Budgets* (per-scale verification budgets)

**Output:** *EvoTree* updated with new designs and verification results

**Function** *Design*(*EvoTree*, *Library*):
0: (*seed*, *refs*) ← SELECTSEEDPARENTS(*EvoTree*, *Library*)
0: (*proposal*, *impl*) ← DESIGNMODEL(*seed*, *refs*)
0: *EvoTree*.addDesign(*proposal*, *impl*)

**Function** *Verify*(*EvoTree*, *Budgets*):
0: (*design*, *scale*) ← EXPERIMENTER.SELECTFOREVAL(*EvoTree*, *Budgets*)
0: **if** *scale* = None **then**
0:     **return** // no budget left
0: **end if**
0: *report* ← RUNTRAINEVAL(*design*, *scale*)
0: *EvoTree*.updateFitness(*design*, *report*)
0: *Budgets*[*scale*] ← *Budgets*[*scale*] − 1

**Function** *Evolve*(*EvoTree*, *Library*, *Budgets*):
0: **while** $\forall s$ with *Budgets*[*s*] > 0 **do**
0:     **async** *Design*(*EvoTree*, *Library*)
0:     **async** *Verify*(*EvoTree*, *Budgets*)
0: **end while**
0: **return** *EvoTree*

---

```python
1   import torch
2   import torch.nn as nn
3   from model_discovery.model.utils.modules import GAUBase, gau_test, UnitDecl
4
5   # YOU CAN IMPORT MORE MODULES HERE #
6
7   # YOU CAN DEFINE MORE CLASSES OR FUNCTIONS HERE #
8
9   class UnitName(GAUBase):
10      """ FILL IN THE DOCSTRING HERE """
11      def __init__(self, embed_dim: int, block_loc: tuple, kwarg_all: dict,
12          device=None, dtype=None,**kwargs): # YOU CAN ADD MORE ARGS#
13          self.factory_kwargs = {"device": device, "dtype": dtype}
14          super().__init__(embed_dim, block_loc, kwarg_all) # DO NOT CHANGE #
15          # COMPLETING THE CODE HERE #
16          raise NotImplementedError
17
18      # YOU CAN ADD MORE FUNCTIONS HERE #
19
20      def _forward(self, X, **Z):
21          # THIS CODE MUST BE COMPLETED #
22          raise NotImplementedError
23
24  # WRITE YOUR UNIT TEST FUNCTIONS HERE #
25  @gau_test # DO NOT CHANGE THIS DECORATOR #
26  def unit_test_name(device=None, dtype=None)->None: # KEEP THE ARGS #
27      # WRITE ASSERTIONS TO PERFORM THE TEST, USE PRINT TO DEBUG #
28      raise NotImplementedError # YOU MUST IMPLEMENT THIS FUNCTION #
29
30  # DECLARE ALL CHILDREN GAUs HERE (EITHER EXISTING OR NEW) #
31  CHILDREN_DECLARATIONS = [ # DO NOT REMOVE THIS LINE #
32      # UnitDecl(
33      #   unitname="", # Name of the child GAU
34      #   requirements="", # Requirements of the child GAU
35      #   inputs=[], # List of argument names
36      #   outputs=[] # List of argument names
37      # ),
38      # ... ADD MORE CHILDREN GAU DECLARATIONS HERE ... #
39  ]
```

Figure 11: Code template for creating new block units.

## B.3 Program Templates and Base classes

In Figure 3, we simplify the presentation to involve just a single Pytorch module `GABBase`. In our system implementation, we have separate classes for units (Fig. 12), called `GAUBase`, in addition to `GABBase`, which is specific to full blocks. We use two classes since units and full blocks have slightly different initialization and constraints. Importantly, however, both modules share the same type structure. Below we show the code template for each, as well as the code template for full autoregressive models called `GAMBase` (standing for **Generalized Autoregressive Model Base**, Fig. 13). In Fig. 11, we show the template that our system fills in when designing new units.

```python
1   class GABBase(nn.Module):
2       """ Base class for Generalized Autoregressive Blocks """
3       def __init__(self,embed_dim: int, block_loc: tuple):
4           super().__init__()
5           self.embed_dim = embed_dim
6           self.block_loc = block_loc # (layer_idx, n_block)
7
8       def _forward(self, X, **Z):
9           raise NotImplementedError
10
11      def forward(self, X, **Z):
12          """Forward pass of the model"""
13          assert len(X.shape) == 3, "Input shape must be (B,L,D)"
14          assert X.shape[-1] == self.embed_dim, "Input shape must be (B,L,D)"
15          Y = self._forward(X, **Z)
16          if isinstance(Y, tuple):
17              Y, Z_ = Y
18          else:
19              Z_ = {}
20          assert Y.shape == X.shape, "Output shape must be (B,L,D)"
21          assert isinstance(Z, dict), "Z must be a dict"
22          Z.update(Z_)
23          return Y, Z
24
25  class GAUBase(nn.Module):
26      """ Base class for Generalized Autoregressive Units """
27      def __init__(self, embed_dim: int, block_loc: tuple, kwarg_all: dict):
28          super().__init__()
29          self.embed_dim = embed_dim
30          self.block_loc = block_loc # (layer_idx, n_block)
31          self.kwarg_all = kwarg_all # kwargs of all units in a GAB
32
33      def _forward(self, X, **Z):
34          raise NotImplementedError
35
36      def forward(self, X, **Z):
37          assert len(X.shape) == 3, "Input shape must be (B,L,D)"
38          assert X.shape[-1] == self.embed_dim, "Input shape must be (B,L,D)"
39          _params = inspect.signature(self._forward).parameters
40          X=X.to(**self.factory_kwargs)
41          _Z = {k: v for k, v in Z.items() if k in _params}
42          Y = self._forward(X, **_Z)
43          if isinstance(Y, tuple):
44              Y, Z_ = Y
45          else:
46              Z_ = {}
47          assert Y.shape == X.shape, "Output shape must be (B,L,D)"
48          assert isinstance(Z_, dict), "Z must be a dict"
49          Z.update(Z_)
50          return Y, Z
```

Figure 12: Base classes and Pytorch modules for model units (`GAUBase`) and block designs (`GABBase`). While both are functionally similar, the `GABBase`.

```python
class GAM(nn.Module):
    ''' Generalized Autoregressive Models'''
    def __init__(self, d_model: int, n_block: int, vocab_size: int = 50277,
            norm_epsilon: float = 1e-5, device = None, dtype = None):
        self.factory_kwargs = {"device": device, "dtype": dtype}
        super().__init__()
        self.d_model = d_model
        self.embedding = nn.Embedding(vocab_size, d_model,
            **self.factory_kwargs)

        block_config = gab_config()
        self.blocks = nn.ModuleList([
            GAB(embed_dim=d_model, block_loc=(layer_idx,n_block),
                device=device, dtype=dtype, **block_config
            ) for layer_idx in range(n_block)
        ])
        self.norm_out = nn.LayerNorm(d_model, eps=norm_epsilon,
            **self.factory_kwargs)

    def forward(self, input_ids):
        hidden_states = self.embedding(input_ids)
        intermediate_vars = {}
        for block in self.blocks:
            hidden_states, intermediate_vars = block(
                hidden_states, **intermediate_vars)
        hidden_states = self.norm_out(hidden_states)
        return hidden_states

class GAB(GABBase):
    def __init__(self,embed_dim: int, block_loc: tuple,
            device=None,dtype=None,**kwargs):
        factory_kwargs = {{"device": device, "dtype": dtype}}
        super().__init__(embed_dim, block_loc)
        self.root = ROOT_GAU(embed_dim=embed_dim, block_loc=block_loc,
            kwarg_all=kwargs, **factory_kwargs, **kwargs)

    def _forward(self, X, **Z):
        X, Z = self.root(X, **Z)
        return X, Z
```

Figure 13: The code template for full autoregressive models.

## B.4 Mutation and Crossover

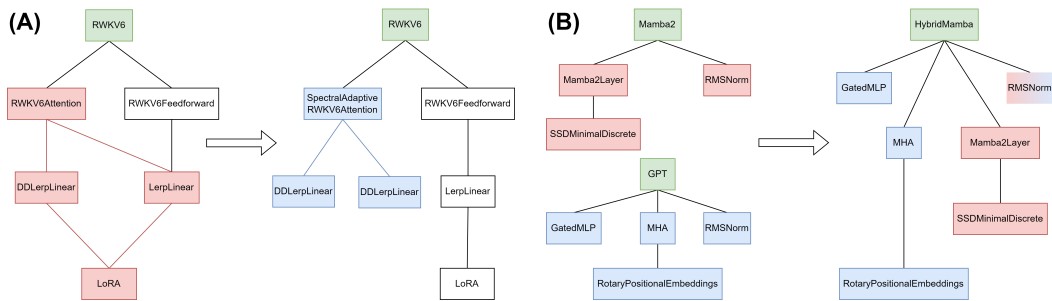

Figure 14: Examples of the **mutation** (A) and **cross-over** (B) operations used in Genesys over GAU trees to create new designs. In A, a variant of the `RWKV6` block is created by replacing the `KWKV6Feedforward` unit (red) with a new unit `SpectralAdaptiveRWKV6Attention` (blue). In B, a new block is created via a novel combination of units in the `Mamba2` and `GPT` blocks.

## B.5 Design Selection

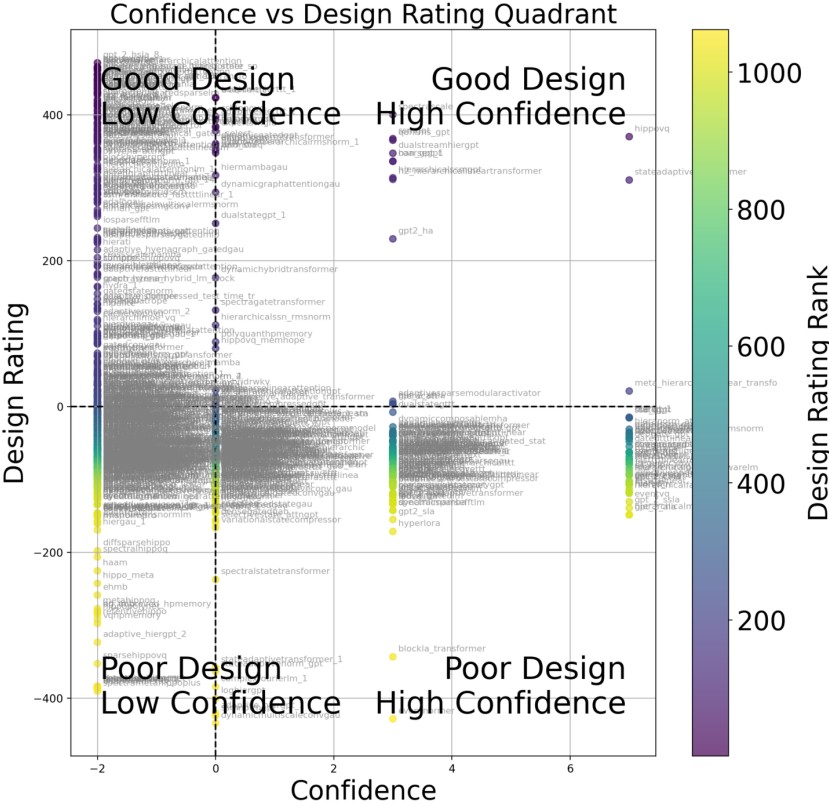

Figure 15: *How do we select input designs for evolution?* A visualization of our quadrant system used for design selection where designs in the evolution tree (points) are scored and ranked according to their fitness or aggregate empirical performance (**Design Rating**) and **confidence** (i.e., number of model scales at which design verification has been performed).

### B.6 Discovery Console

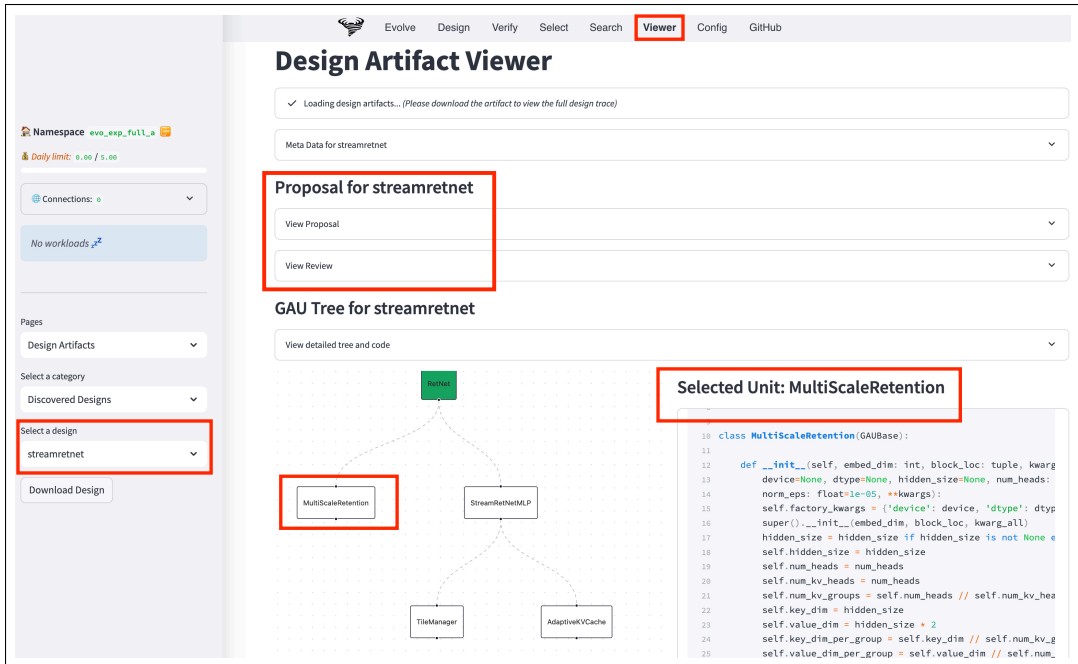

Figure 16: A screenshot of our discovery console available at `https://genesys.allen.ai/` that can be used for running our full system and viewing discovery artifacts. Here we show details of the `StreamRetNetMLP` design (**SRN** in Table 4) using the *Viewer* tab (top), which shows the *original proposal* and *review* (drop-downs in middle) and the agent-authored GAU code design with concrete unit-by-unit implementation details (tree and code on bottom).

## C Experiment Details

### C.1 Experiment setups

#### C.1.1 Training Corpus

Studies in small-scale LMs Eldan & Li (2023); Abdin et al. (2024); Allal et al. (2024) and high-quality datasets Penedo et al. (2024) demonstrate that LMs converge faster in more *educational* datasets with fewer training tokens compared to the lower-quality ones, and the smaller LMs can outperform larger-scale ones with high-quality data. This motivates us to create a small, high-quality educational corpus that allows us to train the models with fewer tokens required, thus improving the verification efficiency. We build our corpus upon SmolLM Allal et al. (2024), a high-quality dataset for training high-performance small LMs. It is a hybrid of 6 subsets, and we filter samples from each of them.

| Subset | Ratio | Tokens | Train | Test | Eval |
|---|---|---|---|---|---|
| FineWeb-Edu | 70.0% | 24.25B | 23.67B | 0.29B | 0.29B |
| Cosmopedia-v2 | 15.0% | 5.20B | 5.08B | 60M | 60M |
| Python-Edu | 8.0% | 2.83B | 2.75B | 40M | 40M |
| OpenWebMath | 5.5% | 1.90B | 1.86B | 20M | 20M |
| StackOverflow | 1.0% | 0.4B | 388M | 6M | 6M |
| DeepMindMath | 0.5% | 0.2B | 194M | 3M | 3M |
| Total | 100% | 34.78B | 33.94B | 0.42B | 0.42B |

Table 9: The statistics of our SmolLM-1/8-Corpus. The ratio of each subset in the mixture and the number of tokens.

Firstly, we filtered samples with $score >= 4$ from FineWeb-edu Penedo et al. (2024), which rates the educational level of each sample from 1 to 5. Then, we filter other subsets while keeping the same mixture as SmolLM. Python-Edu also provides a rating of samples, and we set a cutoff of 3.65 to keep the mixture. All other subsets (Cosmopedia-v2, OpenWebMath Paster et al. (2024), DeepMindMath Saxton et al. (2019), and StackOverflow [9]) do not provide the sample rating; thus, we randomly select from them. This results in an around 1/8 high-quality subset of SmolLM Corpus, which we call **SmolLM-1/8-Corpus**. Statistics presented in Table 9. Following Gao et al. (2020), we randomly sampled 1GB of data from the original SmolLM for each of the test and eval sets, respectively, then removed any verbatim from the training set.

**Customed LM-Eval**    We customized the LM-Eval framework to allow it to accept the GAB models and add the hooks to the verification-time checkers. Moreover, we optimized its evaluation speed for our tasks. We recognize an overhead in LM-Eval during results processing to the stage where some results are not cached and are recomputed. This may be due to some tasks having their internal processing, which makes such caching not generalizable. We manually add this result caching to our benchmarks, which improves the speed of the verification process.

### C.1.2    Hardware Environment

Our experiments are carried out mainly in a set of 10 machines from our internal cluster. There are 8 machines used as the V-Nodes, including three machines with 8 Nvidia A6000 48GB vRAM GPUS with 124 Cloud CPUs and 512G RAM, and five machines with 8 Nvidia L40S 48GB vRAM GPUS with 256 Cloud CPUs and 1TB RAM. Two machines with 3 Nvidia A6000 48GB vRAM GPUS with 34 Cloud CPUs and 254.3 GB RAM are configured as D-Nodes. All machines are running on Ubuntu 20.04. The D-Nodes may execute multiple design threads, while V-Nodes always occupy all resources for one verification thread. We dynamically maintain a ratio between the Design thread and Verification thread to be around 2 to 1 based on our analysis on § E.4.2. In addition, we implement our backend based on Firebase [10].

### C.1.3    Model Experiment Settings

We train all models with single 8 Nvidia L40S machines introduced above for the discovered model evaluations in §5.3. While the discovered models are trained with the same settings from the evolution experiments, we train baseline models with their official implementation or implementation from FLA[11], a community framework for Transformer-alternative architectures. Specifically, we use the Mamba2 repo (`https://github.com/state-spaces/mamba`) for training both Mamba2 and GPT models, the official RWKV repo (`https://github.com/BlinkDL/RWKV-LM`) for the RWKV7 model, and the official TTT implementation (`https://github.com/test-time-training/ttt-lm-jax`) for the TTT model; For RetNet, we apply its FLA implementation. We apply the hyperparameters from their model card in Huggingface with the same scale and directly find them from their official GitHub repositories or papers.

---

[9]`https://huggingface.co/datasets/bigcode/stackoverflow-clean`
[10]`https://firebase.google.com/`
[11]`https://github.com/fla-org/flash-linear-attention`

## C.2 Evolution Parameters

## C.3 Verification

| Knowledge Engine | | | | | |
|---|---|---|---|---|---|
| **Result limits** | | | | **Embeding models** | |
| *arXiv* | *PwC* | *S2* | *RefLib* | *Paper DB* | openai-text-embedding-3-large |
| 3 | 3 | 5 | 5 | *Proposal* | cohere-embed-english-v3.0 |
| **Perplexity settings** | | | | *Unit code* | cohere-embed-english-v3.0 |
| *Model size* | | *Max tokens* | | **Embedding distances** | |
| Large | | 4000 | | *Proposal* | Cosine |
| **Unit search** | | | | *Unit code* | Cosine |
| *Cut-off* | | *Top-K* | | **Paper DB Rerank ratio** | |
| 0.5 | | 4 | | 0.2 | |
| **Proposal search** | | | | | |
| *Cut-off* | | *Top-K* | | *Siblings* | |
| 0.5 | | 4 | | 2 | |

Table 10: Detailed settings for knowledge engine.

| Designer Agent | | | | |
|---|---|---|---|---|
| **Model Dist** | *G4O* | *C35* | *O1P* | *O1M* |
| ***Proposer*** | 0.15 | 0.35 | 0.3 | 0.2 |
| ***Reviewer*** | 0.2 | 0.3 | 0.25 | 0.25 |
| ***Planner*** | 0.06 | 0.32 | 0.32 | 0.3 |
| ***Coder*** | 0.0 | 0.25 | 0.5 | 0.25 |
| ***Observer*** | 0.15 | 0.2 | 0.1 | 0.55 |
| **Rating Thresholds** | | | | |
| *Reviewer* | | *Observer* | | |
| 4.0 | | 3.0 | | |
| **Max Retries** | | | | |
| *Max Search* | *Proposal* | | *Debug* | *Retry* |
| 4 | 5 | | 5 | 5 |

Table 11: Detailed settings for Model Designer Agent.

| Verification Engine | |
|---|---|
| *Context Len.* | 2045 |
| *Optimizer* | AdamW |
| *Tokenizer* | Llama-2-7b-hf |
| *LR Sheduler* | Cosine with min lr |
| *Min lr rate* | 0.1 |
| *Warmup ratio* | 0.02 |
| *Batch size* | 0.5M tokens |
| **Learning rate** | |
| *14M* | 1e-3 |
| *31M* | 1e-3 |
| *70M* | 1e-3 |
| *125M* | 6e-4 |

Table 12: Detailed settings for verification engine.

**Verification Engine and Knowledge Engine** Table 10 and 12 showcase the VE and SE configurations, with specific training settings for varying scales detailed under VE. In SE, "Results limits" specify the count of items retrieved from external resources and the paper vector database ("RefLib"). For unit and proposal search, elaborated in § B.1.5, a "Cut-off" is set to exclude items beyond a certain cosine distance, followed by a selection of "Top-K" samples; in proposals, additional siblings with the same parent as the query design are included. We report the embedding models and distance measures used across various modules. In the Paper Vector DB, Cohere reranker is employed: initially retrieving $K/r$ items (with $0 < r < 1$ as the rerank ratio), it reranks them and selects the top-$K$ items.

**Designer Agent and Selector & Experimenter** The detailed settings for the quadrant selection in Selector and Experimenter (components for design selection in designer and verifier, respectively) are presented in Table 13 and 11. "Quardtile cutoffs" means the position to divide fitness and confidence ranks, 0.25 means the upper 25% designs in the rank are regarded as good/confident. Crossover and design-from-scratch operations can only be selected after their "Warmup rounds". The number of parents decides which GP operation to perform, and this number is sampled from the distribution in the table. We sample references from different types, the items in the reference library with and without reference code, and the previous designs. For different types, we randomly sample with a predefined number.

| | | Selector & Experimenter | | |
|---|---|---|---|---|
| **Quartile Cutoffs** | | | **Seed Distribution** | |
| *Fitness* | *Confidence* | | GPT2 | 0.3 |
| 0.25 | 0.25 | | RetNet | 0.15 |
| **Warmup Rounds** | | | TTT | 0.15 |
| *Crossover* | *Design from Scratch* | | RWKV6 | 0.15 |
| 20 | 30 | | Mamba2 | 0.25 |
| **Num. Parents Distribution** | | | **Restart Scheduler** | |
| *0 (Scratch)* | *1 (Mutation)* | *2 (Cross.)* | *Restart Prob* | *Anneal.* |
| 0.05 | 0.75 | 0.2 | 0.05 | 10 |
| **Num. Reference by Type** | | | **Exploration** | |
| *Reference* | *Ref. w/ Code* | *Prev. Design* | *Top-K Noise* | *Quad.* |
| 2 | 2 | 2 | 0.05 | 0.15 |

Table 13: Detailed settings for Selector & Experimenter.

Besides the Quadrant selection introduced in 4.3, a *random start mechanism* with a probability $p_{rs}$, gradually annealing by a linear schedule that decreases from 1 to $p_{rs}$ after $K_{rs}$ rounds, randomly revisits the initial five designs to foster diverse directions and prevent convergence on narrow paths. The restart is scheduled by a scheduler from 1 to the final restart probability. The "Seed Distribution" determines the probability of the seed designs to be selected when sampling from the seed tree in the restart. "Top-K Noise" decides the change to sample from non-top choices in the Quadrant Selection. "Quad." stands for the probability of sampling from the exploration pool other than the exploitation pool. The foundation model of each designer agent component is randomly selected by a distribution "Model Dist." of different kinds of models. In "Max Rounds", the maximum times to query the knowledge engine ("Max Search"), fail by Reviewer ("Proposal") or observer and symbolic checkers ("Debug"), and retry to implement a proposal ("Retry") are provided.

# D  Additional Results

## D.1  Experiment Results on 125M

| | **Blimp** | **Wnli** | **RTE** | **WG** | **CoLA** | **SST2** | **WSC** | **IS** | **Mrpc** | **avg.** |
|---|---|---|---|---|---|---|---|---|---|---|
| *Random* | 69.75 | 43.66 | 52.71 | 48.78 | 50.00 | 49.08 | 49.82 | 50.03 | 31.62 | 49.49 |
| GPT | 83.22 | 57.75 | 56.32 | 51.93 | 50.05 | 50.44 | 52.75 | 51.79 | 59.07 | 57.04 |
| Mamba2 | 86.13 | 59.15 | 58.11 | 50.28 | 50.69 | 51.83 | **55.68** | 50.43 | 62.99 | 58.37 |
| RWKV7 | 79.16 | 56.34 | 52.17 | 49.09 | 50.06 | 51.26 | 52.01 | 52.47 | 68.38 | 56.77 |
| RetNet | 79.16 | 56.34 | 54.15 | 49.64 | **52.78** | 50.69 | 52.38 | 51.92 | 56.37 | 55.94 |
| TTT | 81.76 | 54.93 | *48.01* | 50.04 | 51.14 | 51.61 | 50.92 | 52.24 | 60.29 | *55.66* |
| VQH | 90.27 | 57.75 | 53.79 | 49.96 | 50.85 | 49.08 | 50.92 | *48.57* | 54.66 | 56.21 |
| HMamba | *77.05* | 60.56 | **59.74** | **52.09** | 50.50 | **53.33** | 52.75 | 51.12 | 68.38 | **58.39** |
| Geogate | 85.96 | 54.93 | 54.15 | 50.36 | 50.21 | 50.92 | 52.01 | 51.79 | **68.63** | 57.66 |
| Hippovq | **92.09** | *46.89* | 53.43 | 50.12 | 50.07 | 50.34 | 49.08 | **53.17** | 68.38 | 57.06 |
| SRN | 79.05 | **61.97** | 53.43 | 49.88 | 50.21 | 49.89 | 52.01 | 50.63 | 54.41 | 55.72 |

Table 14: Performance of human designs and Discovered Models on Various Benchmarks (125M Parameters, 25B Tokens). Metrics indicate accuracy percentages, with bold highlighting the best performance and underlined indicating the second-best. Italics denote outlier performances.

Table 14 presents the experiment results for 125M models of discovered designs and human designs, all models are trained with 25B tokens with the same setting of 350M experiments in §5.3. The discovered model group achieved the best results in 7 out of the 9 benchmarks while also obtaining the highest average score among all designs.

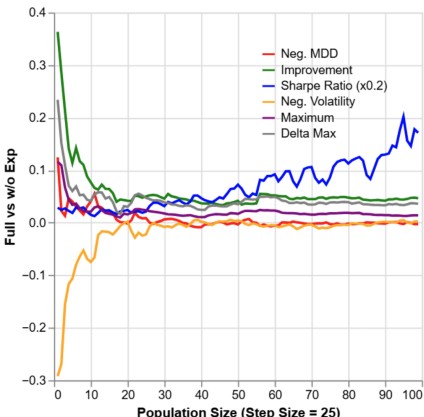 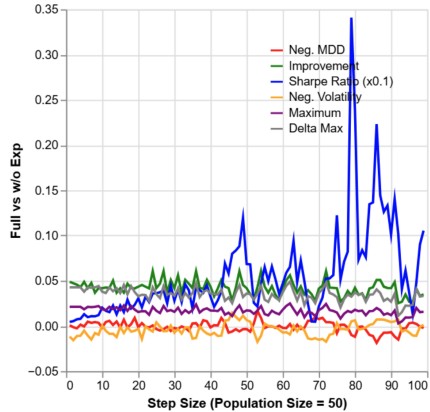

Figure 17: **Left:** How the selection of population size affects metrics, fix step size of 25. **Right:** The impact of step size on metrics, fixing population size to be 50. We present the difference of metrics between "Full" and "w/o Exp." for the first 500 designs.

We study how the selection of population size $S_P$ and step size $k_s$ impact the metrics of the evolution experiments presented in §5.1. We fix the $S_P = 50$ and $k_s = 25$, the settings in our experiments, respectively. Then change the other one from 1 to 100 and compute how the metrics are affected. The results are presented in Fig. 17, where we present negative MDD and volatility, respectively, to make them the higher the better, like other metrics, and scaled SR for better readability. For the population size, besides the initial unstable region when about $S_P < 20$, which is smaller than the step size, the "Full" system consistently presents an overall advantage compared to the "w/o Exp.". For the step size, despite high variances, the advantages stably persist as well.

# E    Analysis of Genesys

To obtain better insights into how to perform optimal evolution with Genesys and how to build more efficient evolutionary systems and discovery agents in general, we provide a detailed analysis of our full evolution experiment here.

## E.1    Analysis of Evolution

We analyze the model design sessions that occurred during the evolution process. Then we obtain more insights by analyzing the entire EvoTree.

### E.1.1 Analysis of Design Sessions

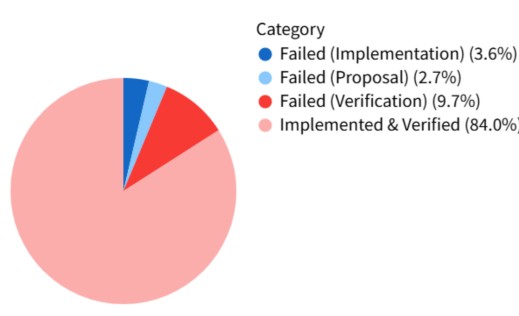

Figure 18: Distribution of the end states of design sessions. "Implemented & Verified" marks a successfully valid design, while brackets for "Failed" mark the stages when a design fails.

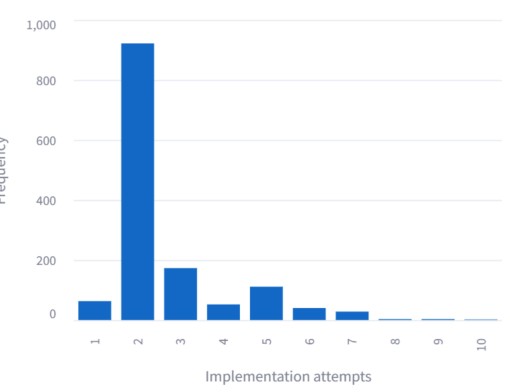

Figure 19: Distribution of number of attempts for implementing design proposals. Mean: 2.65, Median: 2.00, Std: 1.36.

**Statistics of End States** Fig. 18 shows the distribution of the end reasons of design sessions, which shows a full-process valid rate of 84.0%. Despite the high-quality model design agent reducing the design-stage error rate as low as 6.3%, there are still around 9.7% designs that end up not being able to be successfully verified. Resulting in 16.0% invalid designs, which is an unignorable waste, while verify-stage errors can be varied and unpredictable (e.g., some errors like divergence happen at a late stage of training or verification).

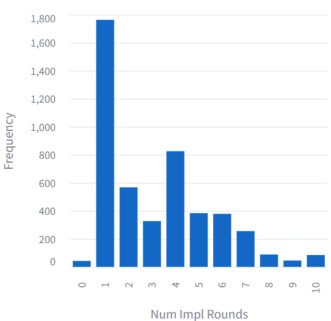

Figure 20: Distribution of the number of rounds that implement units during each implementation attempt. Mean: 3.17, Std: 2.32, Median: 3.00

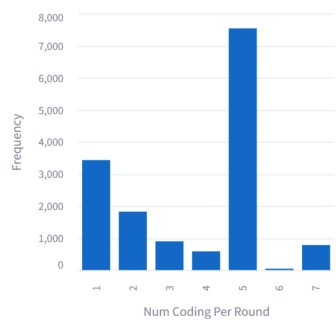

Figure 21: Distribution of the number of coding steps during each implementation round. Mean: 3.68, Std: 1.86, Median: 5.00

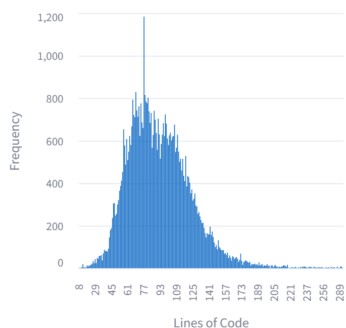

Figure 22: Distribution of lines of generated code in each coding step. Mean: 92.74, Std: 33.99, Median: 88.00.

**Statistics for Implementation Stage** Fig. 19 shows the distribution of attempts made to implement a design proposal. Most proposals can be implemented with 2 attempts. Fig. 20 presents the distribution of the number of rounds in each implementation attempt. Most attempt needs only one round. This is because we allow the agent to implement multiple units under a subtree at a time, while in mutation mode, which is the most frequent operation in our experiment, only one subtree needs to be implemented. Fig. 21 shows the coding steps in each implementation round, including the initial code generation and the later debugging steps with symbolic checker and observer feedback. It takes a majority of 5 steps to implement a unit, showing the difficulty of generating a single valid unit. It can be exponentially more difficult to generate the whole block with the same complexity, and extremely hard to generate in a single shot without VS. Fig. 22 visualizes the number of lines of code produced in each coding step. With VS, the agent only needs to focus on the correctness of an average of around 93 lines of code every time, which largely reduces the complexity.

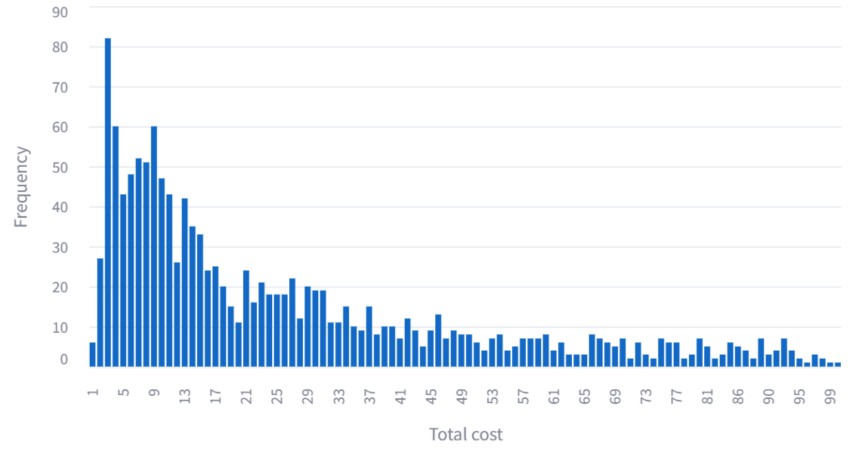

Figure 23: Total design cost distribution. Mean: 28.63, Median: 16.90, Std: 29.49.

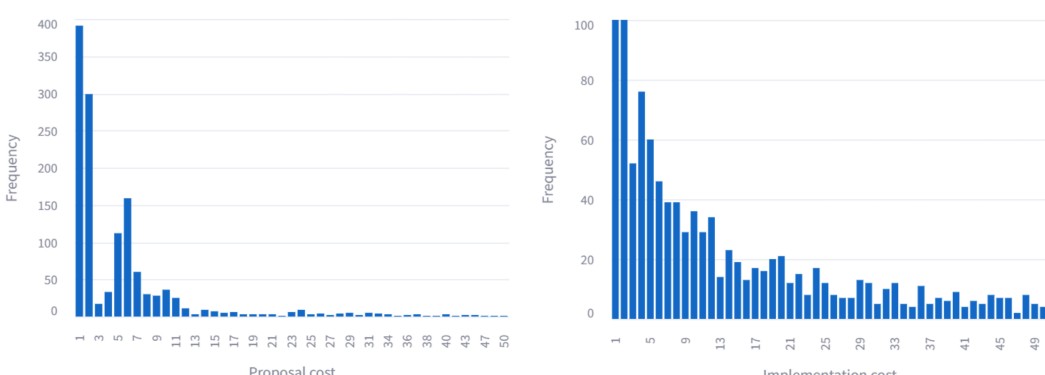

Figure 24: Proposal stage cost distribution. Mean: 6.09, Std: 7.66, Median: 2.68.

Figure 25: Implementation stage cost distribution. Mean: 22.54, Std: 28.80, Median: 10.48.

**Design Costs** The statistics of the total cost are shown in Fig. 23. It takes 28.63 dollars on average for the designs in our experiment. The proposal stage takes a relatively low portion of the cost as presented in Fig. 24, while the implementation stage incurs a high cost as shown in Fig. 25.

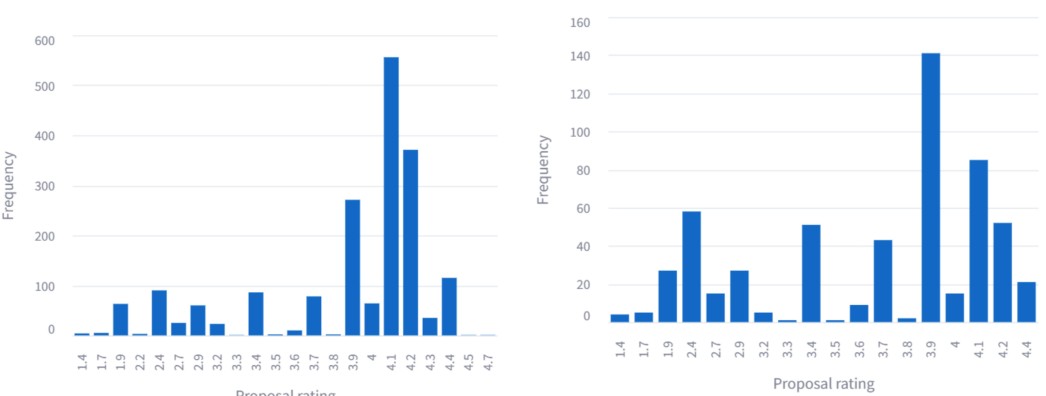

Figure 26: Proposal ratings from all reviewer agents. Mean: 3.91, Median: 4.20, Std: 0.62.

Figure 27: Proposal ratings from O1 preview reviewer agent (o1 preview 2024-09-12). Mean: 3.64, Median: 4.00, Std: 0.72.

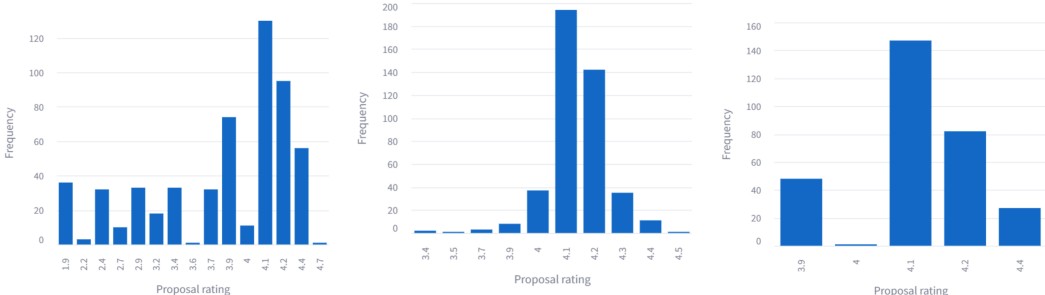

Figure 28: Proposal ratings from O1 mini reviewer agent (`o1 mini 2024-09-12`). Mean: 3.78, Median: 4.10, Std: 0.72.

Figure 29: Proposal ratings from Claude 3.5 Sonnet reviewer agent (`claude-3-5 sonnet 20241022`). Mean: 4.23, Median: 4.20, Std: 0.11.

Figure 30: Proposal ratings from GPT-4o reviewer agent (`gpt-4o 2024-08-06`). Mean: 4.22, Median: 4.20, Std: 0.13.

**Proposal Ratings**  Fig. 26 shows the distribution of all the ratings given by the reviewer agent. LLM agents have a tendency to produce encouraging outputs regardless of prompting that promotes strict and harsh outputs. As a result, we take 4 as the passing border in our experiment. The rating also differs by model type. The OpenAI O1 produces more diverse ratings as shown in Fig. 27 for `O1-preview` and Fig. 28 for `O1-mini` respectively, compared to `GPT-4o` in Fig. 30 and `Claude 3.5 Sonnet` in Fig. 29, which concentrates their rating in a small band of a few numbers. Especially, all ratings given by `Claude 3.5 Sonnet` are above 3.0, while `GPT-4o` produces only 5 different ratings.

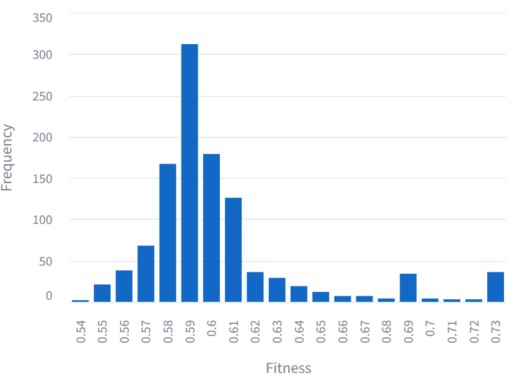

Figure 31: Design fitness Distribution. Mean: 0.61, Median: 0.60, Std: 0.04.

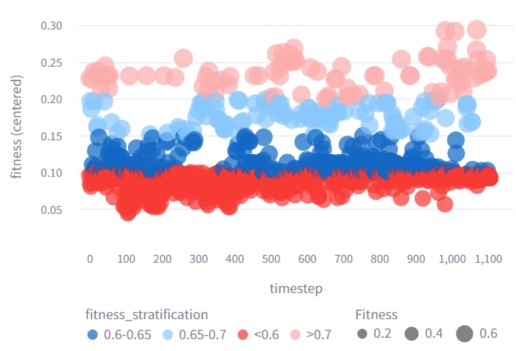

Figure 32: Proposal rating-Timestep Correlation. Correlation coefficient: 0.14

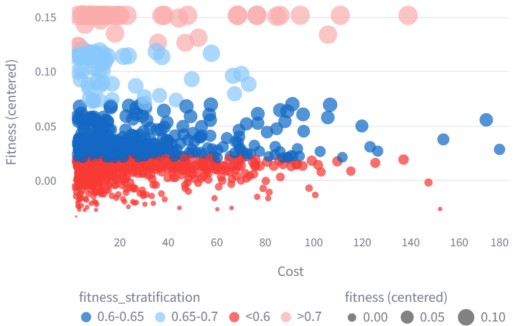

Figure 33: Fitness-Cost Correlation. Correlation coefficient: 0.04

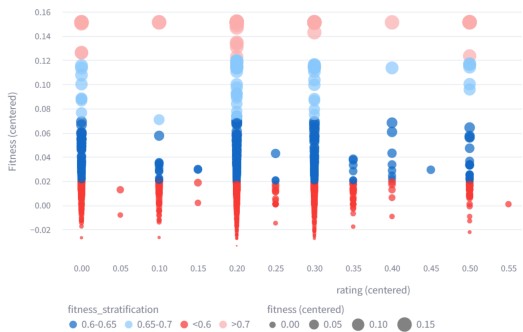

Figure 34: Proposal rating-Fitness Correlation. Correlation coefficient: 0.04

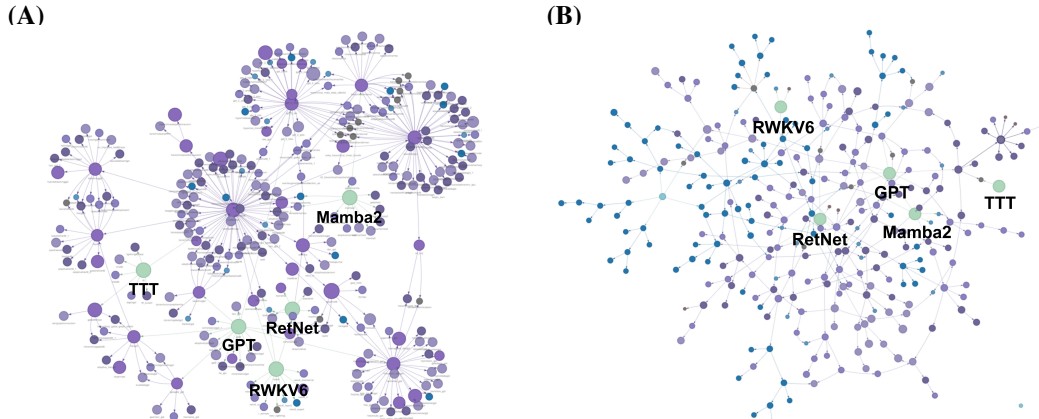

**(A)**          **(B)**

Figure 35: The evolutionary tree (first 300 designs) for our full system (**A**) and our system without experiment verification (**B**) (i.e., designs are selected randomly without fitness) shown with the five starting seed designs that drive discovery (`GPT, TTT, Mamba2, RWKV, RetNet`).

**Fitness**    Fig. 31 shows the fitness of designs, which is nearly normally distributed. Fig. 32 shows the fitness by time steps, where we sort the designs by their timestamp of sampling and take the ranking as the step. The fitness is slowly improving over time, with a positive correlation coefficient of 0.14. We further analyze the correlation between fitness and design cost in Fig. 33, the correlation coefficient is near zero, though positive, showing that putting more design cost does not saliently improve the design quality. Fig. 34 presents the correlation between proposal rating and fitness, similar to design cost, it has a low but negative correlation coefficient, which shows the agent rating is not precise and that the rating can not effectively show the quality of a passing design. Despite a low correlation, the design cost allows us to apply our VS-based implementation process, which already provides some guarantees on the quality of design, and the reviewer serves as a filter for the novelty and quality, which cannot be directly reflected in a simple benchmark-based fitness; moreover, the low-quality designs with a low rating that are already excluded from this analysis may have lower fitness.

### E.1.2    Analysis of Evolutionary Tree

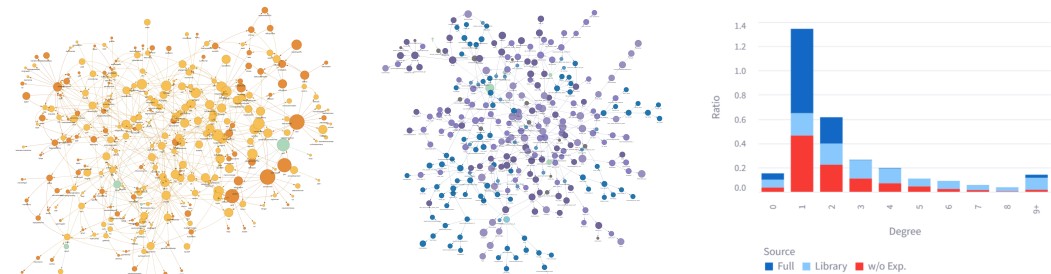

Figure 36: Evolutionary tree of human designs from the reference library.

Figure 37: Evolutionary tree of "w/o Exp." evolution configuration. Showing the first 300 nodes.

Figure 38: Distribution of node degrees in the different configurations.

**EvoTree Visualization**    We visualize the evolutionary tree of human designs from the reference library, whose edges are citation relations in Fig. 36, AI-discovered designs from the "w/o Exp." setup in Fig. 37, and the "full" evolution in Fig. 10 Right. The connection patterns of the "full" evolution and a "w/o Exp." one that does not use fitness-based selection are largely different, where "w/o Exp." is more randomly connected and "full" shows a hubness pattern. It can also be shown from the distribution of node degree in Fig. 38, that the reference library shows an even distribution

of degrees, the degrees of "w/o Exp." are more concentrated in lower degrees, however still more even than the "full" evolution which has few ratios of degrees besides 0, 1, and 2.

| | # node | # edge | Dens. | Deg. mean | Deg. std | $\alpha$ | # Com. | Max com. | Com. size |
|---|---|---|---|---|---|---|---|---|---|
| Library | 297 | 582 | 6.62 | 3.92 | 3.5 | 5.04 | 29 | 55 | 10.24 |
| Full | 1454 | 1670 | 0.79 | 2.30 | 10.9 | 3.42 | 93 | 269 | 15.63 |
| w/o Exp. | 848 | 913 | 1.27 | 2.15 | 1.8 | 5.65 | 61 | 51 | 13.9 |

Table 15: Evolutionary tree statistics. "Deg." represents degree. "Density" measures the connectivity of the network, the more edges, the higher density; "Alpha" is the fitting parameter of a power-law distribution $p(x) \propto x^{-\alpha}$ with degrees. "Com." means Louvain communities that detect node clusters with similar features .

**EvoTree Statistics** We analyzed the statistics of the EvoTrees of the three settings respectively in Table 15. The reference shows a higher density than the discovered EvoTrees; however, notice that the edges in the discovered tree are only the parent nodes, while the random references are not included. "Full" shows a high standard deviation of degrees, which is due to the hubness. A lower alpha in "Full" also shows that it is closer to a long-tail distribution, which also results in its uneven community size distribution. The human EvoTree is closer to the random "w/o Exp." one, which leans more toward explorations, while the "Full" shows a centralized pattern that focuses on improving based on the known good designs.

## E.2 Analysis of Agents

### E.2.1 Analysis of Foundation Models

We analyze how the selection of foundation models may impact fitness and the overall cost, and explore the most cost-effective selections. We use abbreviations of models: C35 represents `Claude 3.5 Sonnet 20241022`, G4O represents `GPT-4o 2024-08-06`, O1P and O1M stand for `O1 preview 2024-09-12` and `O1 mini 2024-09-12` respectively.

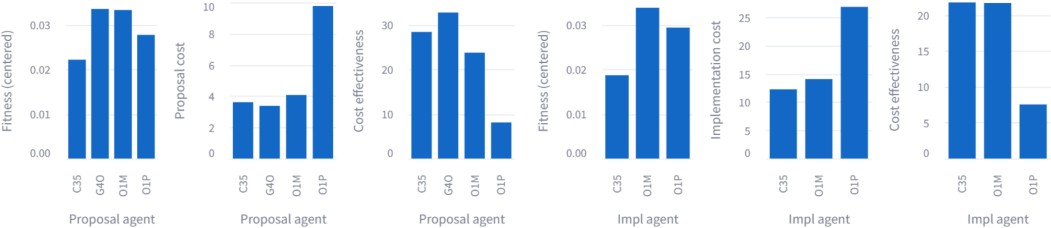

Figure 39: Cost-Effectiveness (fitness by cost) of proposal and implementation stage agents (Proposer and Coder). We plot the proposal or implementation stage costs, the fitness of the produced design, as well as the cost-effectiveness of different proposer and coder models, respectively.

**Cost-Effectiveness of Proposer and Coder** We analyze the cost-effectiveness of foundation models in different stages in Fig. 39. Proposals generated from GPT-4o and O1-mini show a slightly higher fitness of the final design, while the low cost of GPT-4o makes it the highest cost-effective choice for the proposer agent. However, empirically, we observe that GPT-4o sometimes generates preservative designs that directly apply existing known structures, such as simple variants of Transformers, which may guarantee better fitness with a loss of novelty. However, for the course of scientific discovery, sometimes we prefer to perform more explorations on highly inspiring designs with a cost of lower fitness in the current setting. Thus, a mix of multiple agents for better exploration of design ideas is still necessary, and a manual check of proposal quality is important for practical deployment. For the implementation stage, we do not include a GPT-4o agent as it sometimes uses a simplified implementation that deviates from the original proposal to pass the checkers. The O1 mini and Claude 3.5 Sonnet show great cost-effectiveness, reflecting their promises on coding ability, especially, the O1 mini presents an advantage in fitness for its implemented designs.

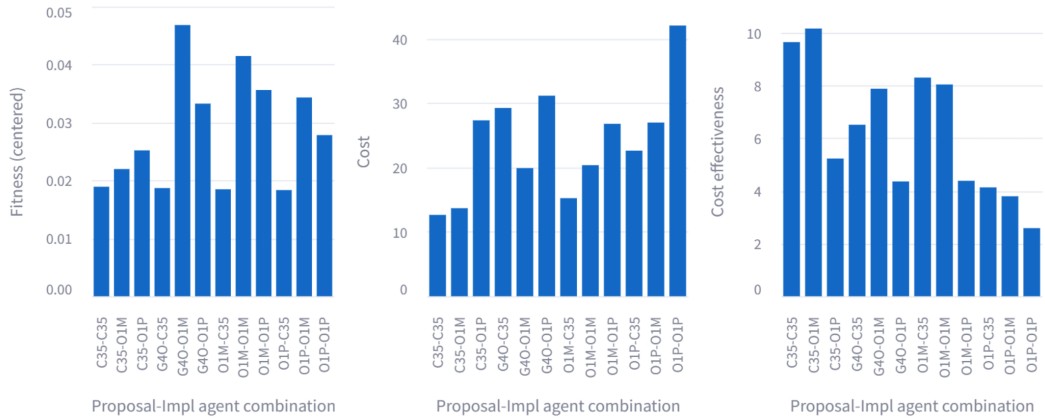

Figure 40: End design fitness, design costs, and Cost-effectiveness analysis for the Proposal-implementation stage agents (proposer and coder) combinations.

**Proposer-Coder Combinations**     We then analyze the proposer and coder for integrity and explore the effectiveness of different combinations in Fig. 40. The combination of GPT-4o and O1-mini shows a fitness advantage despite our observation of preservative designs as discussed above, while the O1-mini-O1-mini combination also shows an above 0.04 centered fitness. The most cost-effective combinations are Claude 3.5 Sonnet with Claude 3.5 Sonnet or O1-mini, while the most expensive O1-preview does not show an advantage in our tasks.

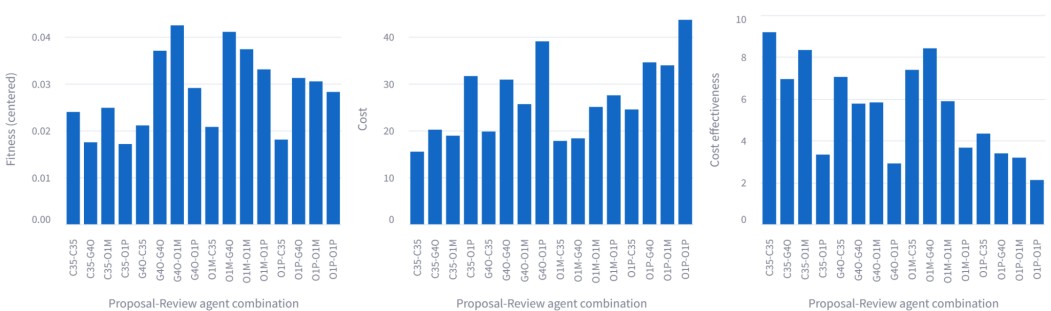

Figure 41: End design fitness, proposal cost, and cost effectiveness for proposal and reviewer agent combinations.

**Proposal-stage Analysis**     We analyze the combination of proposer and reviewer agents in Fig. 41. The combinations of the Claude-3.5 Sonnet and Claude-3.5 Sonnet or O1-mini still show good cost-effectiveness. GPT-4o reviewer with an O1-mini performance also performs well. The O1-preview combinations show the lowest cost-effectiveness with a middle fitness score.

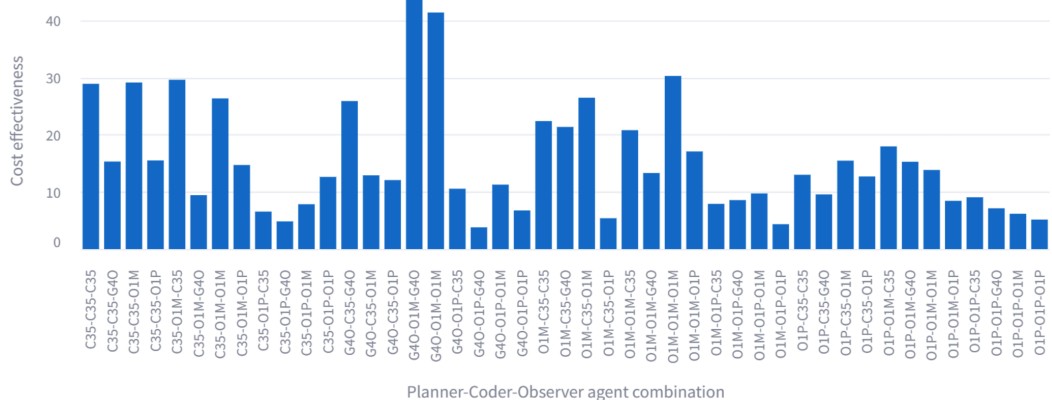

Figure 42: Cost-effectiveness of Planner-Coder-Observer combinations.

**Implementation-stage Analysis**    We study the cost-effectiveness of combinations of planner, coder, and observer in Fig. 42. The GPT-4o planner with O1-mini coder and GPT-4o or O1-mini observers shows the best cost-effectiveness that surpasses all other combinations by at least 30%, showing a great coding ability of O1-mini. Claude-3.5 Sonnet also shows a good coding power that all combinations with cost-effectiveness above 20 have either an O1-mini or Claude-3.5 Sonnet as the coder.

**Conclusion**    Our analysis suggests that the Claude-3.5 Sonnet suits the proposer. GPT-4o, O1-mini, and Claude-3.5 Sonnet are good choices for reviewers. The O1-mini or Claude-3.5 Sonnet are good coders that can be combined with a GPT-4o or Claude-3.5 Sonnet planner and observer.

### E.2.2    Analysis of Implementation Errors

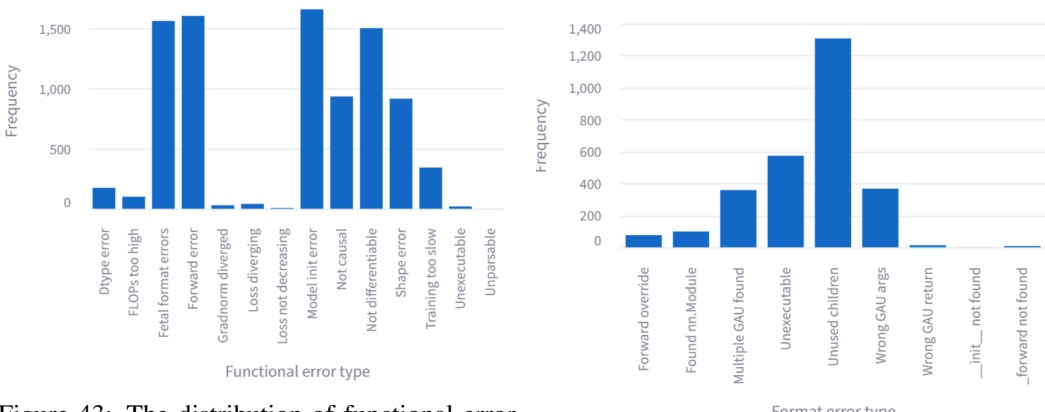

Figure 43: The distribution of functional error types. The designs with fetal format errors may skip the functional checks.

Figure 44: The distribution of format error causes.

We analyze the causes of the design's failure in symbolic checkers. Fig. 43 shows the distributions of functional error types while Fig. 44 presents the format error types. 91.0% erroneous designs have functional errors while 17.3% have format errors. Besides the fetal formats errors such as not found GAU implementations, a large portion of functional errors are incurred in the forward pass, model init, and incorrect tensor shapes in operations, causality and differentiability are also challenging while one common cause of differentiability from our observation is due to the redundant parameters that are not used in the backward passes. A frequent format error is declaring children in the declarations while not instantiating them in the unit body. Unexecutable code is also common, while including multiple GAUs in a single file, which we require to make them separate, and passing wrong arguments to the GAU, such as missing templated args, are frequent.

### E.3 Analysis of Discovered Models

### E.3.1 Analysis of Verification Process

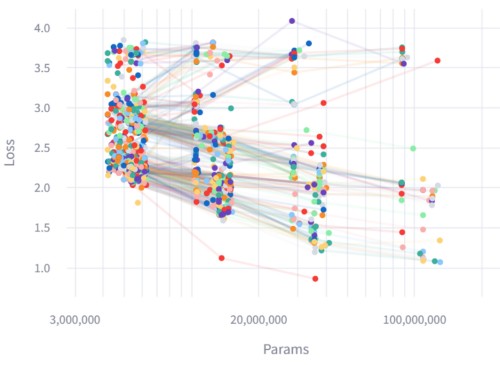

Figure 45: The training loss by the number of parameters of models. Different colors mark different designs.

Figure 46: The training loss by number of training tokens. Each point marks a trained model.

**Training Loss** We show how the training loss changes under different scales in Fig. 45. We can observe an overall trend of the training loss decreasing by scales, which shows that most designs follow scaling laws Kaplan et al. (2020). Fig. 46 shows another view by comparing training losses and training tokens. The training losses present a decreasing trend as the training token increases.

|  | std | mean | min | max | rand | # | max/rand>0.05 | #>200 |
|---|---|---|---|---|---|---|---|---|
| mrpc | 0.1465 | 0.4929 | 0.3113 | 0.6863 | 0.3162 | 408 | True | True |
| qqp | 0.1112 | 0.5011 | 0.3675 | 0.6329 | 0.6318 | 40430 | False | True |
| wnli | 0.0593 | 0.5017 | 0.3239 | 0.6479 | 0.4366 | 71 | True | False |
| blimp | 0.0566 | 0.7044 | 0.5777 | 1.0 | 0.6975 | 65093 | True | True |
| rte | 0.0288 | 0.5064 | 0.4224 | 0.5884 | 0.5271 | 277 | True | True |
| tinyTruthfulQA | 0.0185 | 0.5233 | 0.3967 | 0.5469 | 0.4866 | 100 | True | False |
| wsc273 | 0.0179 | 0.5005 | 0.4322 | 0.6117 | 0.4982 | 273 | True | True |
| inverse_scaling | 0.0141 | 0.4992 | 0.3868 | 0.5462 | 0.5003 | 29548 | True | True |
| cola | 0.0138 | 0.4979 | 0.4512 | 0.5452 | 0.5 | 1043 | True | True |
| sst2 | 0.0138 | 0.4997 | 0.4495 | 0.5573 | 0.4908 | 872 | True | True |
| qa4mre_2011 | 0.012 | 0.1549 | 0.1167 | 0.2583 | 0.15 | 120 | True | False |
| winogrande | 0.0103 | 0.495 | 0.4538 | 0.5359 | 0.4878 | 1267 | True | True |
| mnli | 0.01 | 0.3432 | 0.3256 | 0.3564 | 0.3545 | 9815 | False | True |
| sciq | 0.0088 | 0.1998 | 0.0 | 0.219 | 0.2 | 1000 | True | True |
| mnli_mismatch | 0.0084 | 0.3428 | 0.3255 | 0.3568 | 0.3522 | 9832 | False | True |
| mathqa | 0.0078 | 0.1984 | 0.1792 | 0.2352 | 0.202 | 2985 | True | True |
| qnli | 0.0073 | 0.4995 | 0.4746 | 0.534 | 0.5054 | 5463 | True | True |
| qa4mre_2012 | 0.0073 | 0.2578 | 0.1938 | 0.2812 | 0.2562 | 160 | True | False |
| qa4mre_2013 | 0.0069 | 0.1699 | 0.1479 | 0.1972 | 0.1655 | 284 | True | True |
| openbookqa | 0.0066 | 0.2664 | 0.244 | 0.286 | 0.258 | 500 | True | True |
| arc_challenge | 0.0054 | 0.2861 | 0.227 | 0.3012 | 0.2841 | 1172 | True | True |
| tinyGSM8k | 0.004 | 0.006 | 0.0055 | 0.078 | 0.0055 | 100 | True | False |
| piqa | 0.0036 | 0.5005 | 0.4908 | 0.5125 | 0.5022 | 1838 | False | True |
| arc_easy | 0.0033 | 0.2627 | 0.2508 | 0.2727 | 0.2618 | 2376 | False | True |
| hellaswag | 0.0011 | 0.2611 | 0.2504 | 0.2653 | 0.2603 | 10042 | False | True |
| squad_completion | 0.0011 | 0.0012 | 0.0 | 0.0047 | 0.0 | 2984 | True | True |
| swag | 0.0008 | 0.2539 | 0.2466 | 0.2566 | 0.254 | 20006 | False | True |
| lambada_openai | 0.0004 | 0.0 | 0.0 | 0.0128 | 0.0 | 5153 | True | True |
| triviaqa | 0.0001 | 0.0 | 0.0 | 0.0002 | 0.0002 | 17944 | False | True |
| tinyGSM8k | 0.0 | 0.0055 | 0.0055 | 0.0055 | 0.0055 | 100 | False | False |

Table 16: Statistics of evaluation benchmarks results from all verifications performed in the full evolution experiment. Sorted by the standard deviations.

**Benchmarks analysis**  Table 16 presents statistics of evaluation results from the full evolution of each benchmark as well as the standard that we select the benchmarks in the discovered model evaluation experiments in §5.3. We first focus on the standard deviations, which show a "learnability" of a task in our setting. If all models perform similarly, then it means a task cannot be effectively learned. We consider only the ones with a $std > 0.01$. We then analyze the ratio between the best result and the random result. If the best improvement is less than 5%, we skip the task as it is too challenging. Finally, we consider the number of examples with 200 as a cut-off, which we regard more sufficient to provide a comprehensive evaluation. The only exception is WNLI; we preserve it as firstly it has a high standard deviation, and secondly, it is a widely used benchmark that shows a good evaluation quality.

**Benchmark Correlations**  Fig. 47 further analyzes the correlation of the selected tasks. Most tasks show a low correlation with each other, with few pairs, inverse-scaling and blimp, rte, and wnli showing a relatively higher negative correlation with coefficients $< -0.5$. For the positively correlated tasks, WSC and cola show a correlation coefficient of $0.35$ while all other tasks are $\leq 0.25$. It shows a low overall correlation between tasks, which implies our task selections have a low overlap in the tasking topics and can evaluate different parts of the underlying abilities and potential of the design.

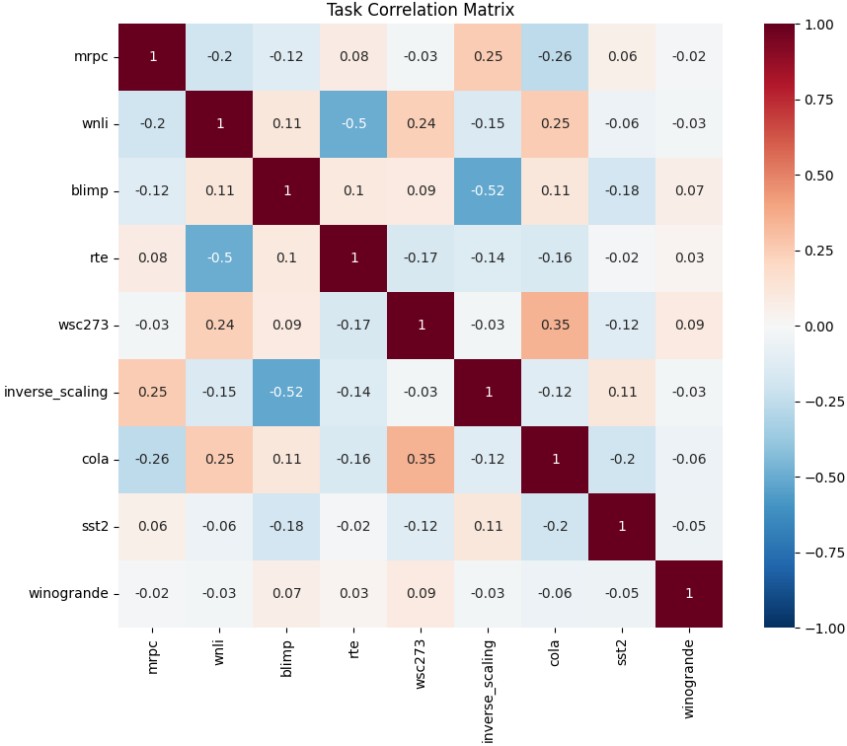

Figure 47: The correlations between the nine selected tasks based on the evaluation results.

### E.3.2 Analysis of Model Units

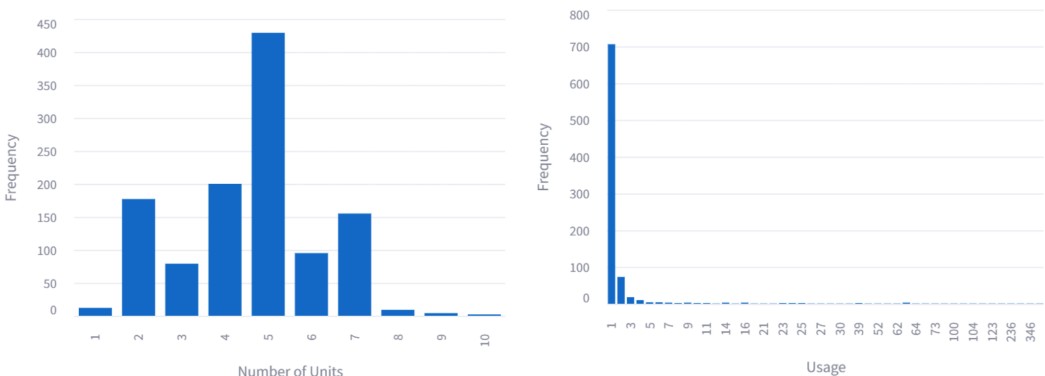

Figure 48: Number of units per design. Mean: 4.59, Std: 1.61, Median: 5.00.

Figure 49: Number of usages for the units. Mean: 5.47, Std: 34.48, Median: 1.00.

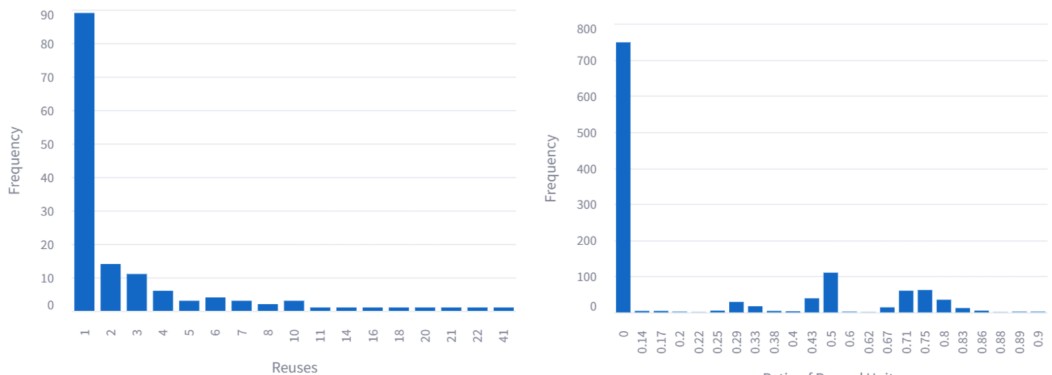

Figure 50: Number of times being *adapted* to reuse of units for the units that have been reused. Mean: 3.10, Std: 5.06, Median: 1.00.

Figure 51: Ratio of reused units in designs. Mean: 0.21, Std: 0.30, Median: 0.00.

**Unit Usage and Reuses**   Fig. 48 presents the number of units contained in designs, and Fig. 49 shows the statistics for the number of times a unit gets used. Some common units are frequently used, such as the RMSNorm and RotaryPositionalEmbedding. With those existing building blocks, it makes it easier to build relatively complex solutions with around 5 or more blocks, which is the median number of units in a design. Fig. 50 considers the case of unit *reuse*, which specifically refers to the case of *adapting* an existing unit to a new scenario with code modifications as compared to the direct *usage* above, which does not need to change the code. Fig. 51 further presents the ratio of unit reuses. Model reuse provides a flexible way to reduce the complexity of implementation and the accumulation of experience. The dictionary of units from previous designs is provided to the agent for access with embedding-based search analogous to deep learning frameworks that simplify novel model constructions by improving code reusability.

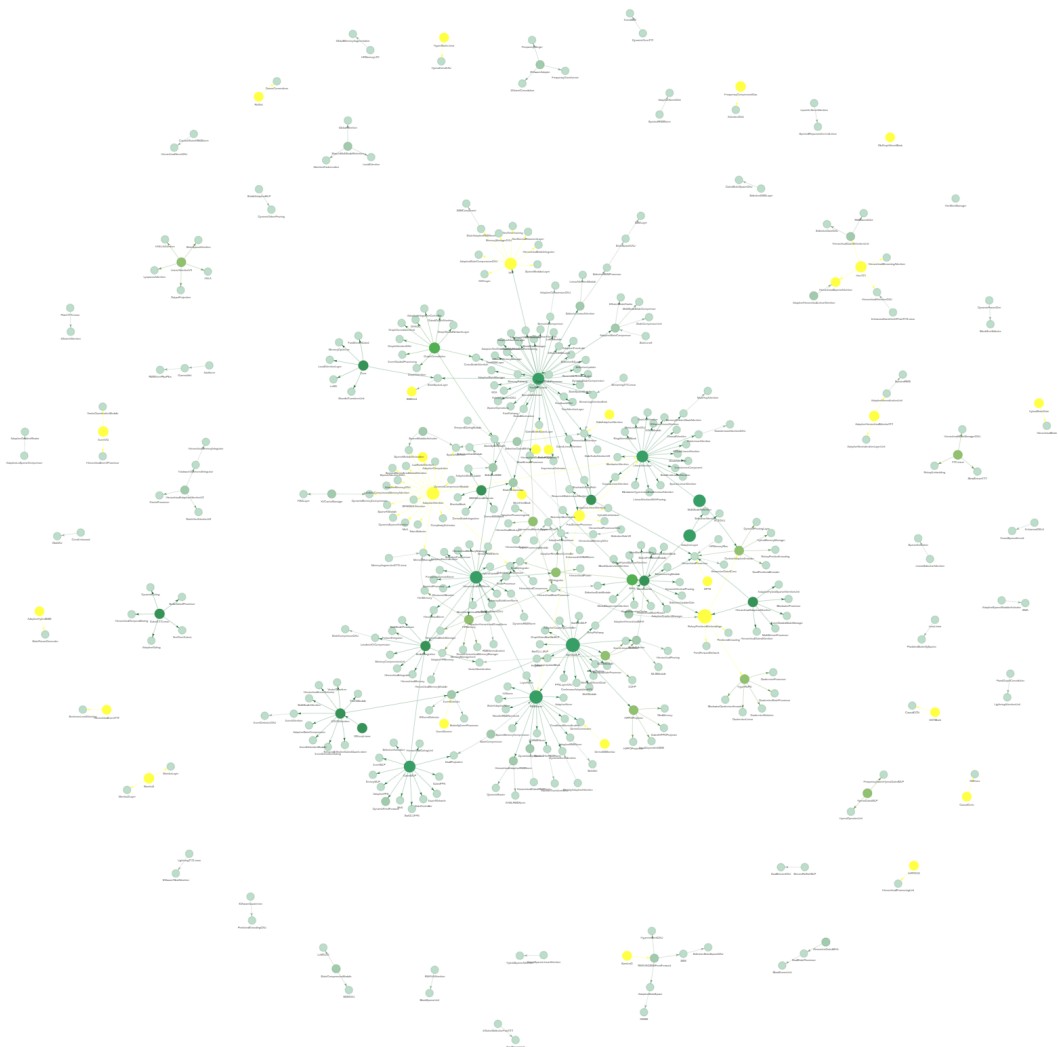

Figure 52: Unit network. The edges represent reuse relations between units.

### E.3.3 Unit-Performance Relation

We explore the predicativity of model or task performance by unit. By using a Bag-of-Words (BoW) as a simple unit-based descriptor of model architecture, which shows underlying constitutions of a design (for example, GPT can be represented as `GPT MHA GatedMLP RMSNorm RotaryPositionalEmbedding`), we study whether it can be used to predict the model performance.

| | 5-Fold | | LOOCV | | Support | | | |
|---|---|---|---|---|---|---|---|---|
| | Training | Test | Training | Test | S1 | S2 | S3 | S4 |
| blimp | 0.817 (± 0.004) | 0.720 (± 0.032) | 0.734 (± 0.002) | 0.633 | 684 | 291 | 51 | 34 |
| inverse_scaling | 0.768 (± 0.011) | 0.671 (± 0.034) | 0.743 (± 0.001) | 0.655 | - | 36 | 714 | 310 |
| openbookqa | 0.621 (± 0.007) | 0.481 (± 0.019) | 0.618 (± 0.001) | 0.413 | 157 | 466 | 338 | 99 |
| arc_challenge | 0.805 (± 0.010) | 0.787 (± 0.020) | 0.831 (± 0.001) | 0.79 | - | - | 207 | 853 |
| lambada_openai | 0.998 (± 0.001) | 0.999 (± 0.002) | 0.997 (± 0.000) | 0.999 | 1059 | - | - | 1 |
| mathqa | 0.684 (± 0.017) | 0.524 (± 0.025) | 0.675 (± 0.001) | 0.52 | 79 | 551 | 389 | 41 |
| rte | 0.704 (± 0.004) | 0.548 (± 0.036) | 0.648 (± 0.001) | 0.535 | 192 | 253 | 571 | 44 |
| mnli | 0.672 (± 0.003) | 0.532 (± 0.030) | 0.572 (± 0.001) | 0.351 | 172 | 154 | 333 | 401 |
| winogrande | 0.753 (± 0.005) | 0.605 (± 0.015) | 0.709 (± 0.001) | 0.515 | 10 | 460 | 569 | 21 |
| piqa | 0.661 (± 0.011) | 0.524 (± 0.050) | 0.677 (± 0.001) | 0.542 | 87 | 593 | 365 | 15 |
| qa4mre_2012 | 0.826 (± 0.004) | 0.763 (± 0.016) | 0.798 (± 0.000) | 0.746 | - | - | 264 | 796 |
| mnli_mismatch | 0.655 (± 0.032) | 0.475 (± 0.054) | 0.547 (± 0.001) | 0.331 | 168 | 224 | 325 | 343 |
| qnli | 0.777 (± 0.007) | 0.658 (± 0.042) | 0.759 (± 0.000) | 0.656 | 28 | 714 | 312 | 6 |
| hellaswag | 0.837 (± 0.002) | 0.780 (± 0.023) | 0.832 (± 0.001) | 0.775 | - | - | 229 | 831 |
| sciq | 1.000 (± 0.000) | 1.000 (± 0.000) | 1.000 (± 0.000) | 1.0 | - | - | - | 1060 |
| qqp | 0.629 (± 0.009) | 0.443 (± 0.035) | 0.576 (± 0.001) | 0.377 | 403 | 239 | 125 | 293 |
| arc_easy | 0.749 (± 0.010) | 0.614 (± 0.031) | 0.704 (± 0.001) | 0.573 | 25 | 341 | 641 | 53 |
| qa4mre_2013 | 0.654 (± 0.006) | 0.510 (± 0.030) | 0.602 (± 0.001) | 0.436 | 178 | 475 | 341 | 66 |
| sst2 | 0.710 (± 0.011) | 0.543 (± 0.033) | 0.699 (± 0.001) | 0.485 | 44 | 538 | 469 | 9 |
| mrpc | 0.631 (± 0.010) | 0.431 (± 0.028) | 0.561 (± 0.002) | 0.288 | 278 | 190 | 245 | 347 |
| swag | 0.775 (± 0.008) | 0.700 (± 0.048) | 0.781 (± 0.001) | 0.68 | - | - | 330 | 730 |
| wnli | 0.654 (± 0.004) | 0.497 (± 0.049) | 0.609 (± 0.001) | 0.405 | 297 | 311 | 425 | 27 |
| qa4mre_2011 | 0.830 (± 0.005) | 0.762 (± 0.016) | 0.814 (± 0.001) | 0.755 | 817 | 243 | - | - |
| wsc273 | 0.679 (± 0.008) | 0.557 (± 0.036) | 0.691 (± 0.001) | 0.538 | 82 | 592 | 354 | 32 |
| tinyGSM8k | 0.986 (± 0.002) | 0.990 (± 0.004) | 0.983 (± 0.000) | 0.99 | 1049 | 7 | 2 | 2 |
| squad_completion | 0.801 (± 0.009) | 0.695 (± 0.018) | 0.707 (± 0.000) | 0.621 | 679 | 168 | 188 | 25 |
| cola | 0.733 (± 0.011) | 0.658 (± 0.028) | 0.710 (± 0.000) | 0.643 | 183 | 691 | 168 | 18 |

Table 18: Naive Bayes on BoW-tasks, F1 scores are presented. Support is the total number of each segment; the values are divided by the 4 evenly spaced segments between min and max. Using even segments instead of quartiles to avoid repeated values. Excluded the triviaqa and tinyGSM8k, which have min and max are too close.

| Fitness | Precision | Recall | F1 | Support | Precision | Recall | F1 | Support |
|---|---|---|---|---|---|---|---|---|
| | *Training set (80%)* | | | | *Test set (20%)* | | | |
| 0.6-0.65 | 0.817 | 0.597 | 0.690 | 298 | 0.528 | 0.264 | 0.352 | 72 |
| 0.65-0.7 | 0.647 | 0.733 | 0.688 | 45 | 0.706 | 0.750 | 0.727 | 16 |
| <0.6 | 0.778 | 0.930 | 0.847 | 471 | 0.635 | 0.871 | 0.735 | 116 |
| >0.7 | 0.938 | 0.441 | 0.600 | 34 | 0.000 | 0.000 | 0.000 | 8 |
| Accuracy | | | 0.783 | 848 | | | 0.623 | 212 |
| Macro avg. | 0.795 | 0.675 | 0.706 | 848 | 0.467 | 0.471 | 0.453 | 212 |
| Weighted avg. | 0.791 | 0.783 | 0.774 | 848 | 0.580 | 0.623 | 0.576 | 212 |
| *5-fold CV F1* | *0.785 (± 0.004)* | | | | *0.658 (± 0.024)* | | | |
| *LOOCV F1* | *0.786 (± 0.002)* | | | | *0.658* | | | |

Table 17: Use BoW of units to predict the fitness with Naive Bayes.

**BoW to Fitness**  We divide fitness into four levels, then train a Naive Bayes (NB) class over the BoW representations to predict the fitness level; the results are presented in 17. In the test set, it achieves an F1 in a Leave-One-Out Cross Validation (LOOCV) of 0.658, which shows some predictivity. While a BoW model and an NB classifier are simple and naive setups, we show that it is possible to predict a design performance with its architectural components, which can be used to guide the model design process.

**BoW to Task Performance**  We then use the same manner to predict each downstream task performance in Table 18. 19 tasks show a test LOOCV F1 higher than 0.5, which indicates the predictability from unit to certain downstream abilities. This shows the possibility of providing fine-grain unit-level guidance for designing models with certain targeted downstream applications (i.e., what kind of unit combination may help to improve the performance in certain tasks).

### E.4 Analysis of System and Performance

#### E.4.1 Training Time Estimation

| | 14M | 31M | 70M | 125M | 14M | 31M | 70M | 125M | 14M | 31M | 70M | 125M |
|---|---|---|---|---|---|---|---|---|---|---|---|---|
| | | *Optimistic* | | | | *Median* | | | | *Pessimistic* | | |
| Single run (s) | 46 | 123 | 498 | 3258 | 174 | 437 | 1338 | 8778 | 302 | 751 | 2178 | 14298 |
| Token mult. | 50 | 40 | 30 | 20 | 50 | 40 | 30 | 20 | 50 | 40 | 30 | 20 |
| Num. runs | 1000 | 400 | 150 | 40 | 1000 | 400 | 150 | 40 | 1000 | 400 | 150 | 40 |
| Total GPU hrs | 255.6 | 218.7 | 249 | 289.6 | 966.7 | 776.9 | 669 | 780.3 | 1677.8 | 1335.1 | 1089 | 1270.9 |
| Est. hours | 31.9 | 27.3 | 31.1 | 36.2 | 120.8 | 97.1 | 83.6 | 97.5 | 209.7 | 166.9 | 136.1 | 158.9 |
| **Est. total** | | | | **5.3 days** | | | | **16.6 days** | | | | **28.0 days** |

Table 19: Estimation of total Training time in a single $8\times$A6000 machine. The single run is the training time under training tokens to be 20 times of parameters, while the estimated times are based on the training tokens in our setting.

We present the estimated training time in a single $8\times$A6000 machine in Table 19. We train a fully optimized GPT model equipped with hardware optimizations like FlashAttention in different scales with training tokens to be 20 times the parameters within our setting and record the running time as the *Optimistic* case $t_O$, then a *Median* case $t_M$ with an unoptimized, vanilla implementation of GPT, the *Pessimistic* case is a simple linear extrapolation $t_P = 2 \times t_M - t_O$. We then scale it to our training tokens and the number of runs by simply scaling up linearly. It provides us with a simple way to estimate of running time and decide the budget.

#### E.4.2 Optimal Pipeline Throughput and the V-D Ratio

Consider a distributed pipeline with two types of nodes:

- **D-nodes**: Responsible for *design* tasks. There are $N_D$ such nodes. In practice, each D-Node can launch multiple design threads in parallel. We assume single threads here for simplicity. Each design task takes an average time of $T_D$ to complete.

- **V-nodes**: Responsible for *verification* tasks. There are $N_V$ such nodes. Each verification task takes an average time of $T_V$ to complete.

A design cannot be considered "finished" until it has been verified. Hence, the overall throughput of *verified* designs (i.e., finished products) depends on the capacity of both types of nodes in the pipeline.

**Throughput formulation** Assume each node works in parallel (each occupying a "lane"), and that work can be pipelined optimally so different nodes start at possibly shifted time points to maximize steady-state throughput. The effective throughput $\Theta$ (the number of verified designs produced per unit time) is determined by the slower stage of the pipeline. In steady state, each design task must be followed by a verification task, so the throughput is

$$\Theta \;=\; \min\!\Big(\frac{N_D}{T_D}, \frac{N_V}{T_V}\Big). \tag{2}$$

This expression states that the system cannot output verified designs at a rate faster than the rate of design ($N_D/T_D$) or the rate of verification ($N_V/T_V$), whichever is smaller.

**Optimal ratio of V-nodes to D-nodes** Let $r = \frac{N_V}{N_D}$ be the ratio of verification nodes to design nodes. Our goal is to choose $r$ to maximize the throughput (2). Observe that if

$$\frac{N_D}{T_D} \;>\; \frac{N_V}{T_V},$$

then verification is the bottleneck, and adding more D-nodes beyond a certain point does not improve the final throughput of *verified* designs. Conversely, if

$$\frac{N_D}{T_D} \;<\; \frac{N_V}{T_V},$$

then design is the bottleneck, and extra V-nodes do not improve the throughput.

Hence, the throughput is maximized when

$$\frac{N_D}{T_D} = \frac{N_V}{T_V}.$$

Solving for $r = \frac{N_V}{N_D}$ gives

$$r^* = \frac{T_V}{T_D}. \tag{3}$$

That is, the *optimal ratio* of V-nodes to D-nodes matches the ratio of the average times required: $\frac{T_V}{T_D}$. In this balanced scenario, both design and verification stages have the same steady-state throughput, and no additional overprovisioning of either D-nodes or V-nodes will further increase $\Theta$.

**Maximal throughput**    When $r = r^*$, the stages are balanced, and the maximal throughput is

$$\Theta_{\max} = \frac{N_D}{T_D} = \frac{N_V}{T_V} \quad \text{where} \quad N_V = r^* N_D.$$

This completes the derivation of the optimal pipeline configuration for maximizing the rate of fully verified (and thus valid) designs.

**Optimal V-D ratio for Genesys**    The expected time of verification jobs for a design can be computed as

$$E(T_V) = \sum_s^{\mathbf{S}} t_s P_{verify}(s)$$

where $t_s$ is the expected running time for a verification on scale $s$, $\mathbf{S}$ is the set of all scales, $P_{verify}(s)$ is the probability of verify a design on scale $s$, which is computed as the accumulated selection ratio $\prod_{i=\hat{s}}^{s} sr_i$ from the lowest scale $\hat{s}$.

Based on Table 2, the expected verification time $T_V$ with a single 8×A6000 node for the optimistic, median, and pessimistic cases is 189s, 566s, and 944s. Notice that it only considers the training time, which underestimates the actual time, we take an approximation of 30% of training time here. The expected design time can be computed as $\frac{\bar{T}_D}{1-Err_{vt}}$ where $\bar{T}_D$ is the average agent design time which we take 20 minutes here empirically, and $Err_{vt}$ is the verify time error rate, which is 8.61% as presented in Table 2. It results in 21.9 minutes of $T_D$. Thus, the optimal V-D ratio for our system with a median expectation of verification time is about 0.56. Notice that each D-Node can execute multiple design threads at a time, thus *each V-Node can be saturated with around 2 design threads*. It helps us to decide the total design threads in our evolution setting as discussed in S C.1.

In practice, we take a V-D ratio lower than this number, as additional V-Nodes do not impact the evolutionary progress by exhausting the verification budget, which is guarded by the LoS, while an inadequate V node may lead to the D-Node selecting parents without sufficient confidence. Due to the risk of connection instability, preemption, or any unexpected down of the machine, the availability of a V-Node is not steady, a lower V-D ratio increases the robustness of the system.

# F Prompts

## F.1 Proposer

### F.1.1 System and GP Background

---

**System Prompt for Proposer**

You are a language modeling researcher, your role is to propose a novel autoregressive language model (LM) block design.

## Background

Modern LMs are typically structured as a stack of repeating blocks. Each block processes:

1. **Input**: A sequence of embeddings X of shape (B, L, D), where:
- B is the batch size.
- L is the sequence length.
- D is the embedding dimension.
2. **Intermediate Variables**: Z (e.g., memory, states, caches) passed as keyword arguments.

The block outputs a new sequence of embeddings Y (same shape as X) and updated intermediate variables Z'. The overall architecture can be represented as follows:

```python
tokens = Tokenizer(sentence)
X = Embeddings(tokens)
Z = {} # Initialized as an empty dictionary, updated by each block.
for block in Blocks:
    X, Z = block(X, **Z)
output = Logits(X)
```

Your goal is to design a proposal for a novel LM block that outperforms current state-of-the-art models, aiming for:
- Low perplexity on corpora,
- High accuracy on downstream tasks,
- Robustness to varied inputs,
- Efficiency in both training and inference,
- Excellent scalability with more data and larger models.

### Generalized Autoregressive Units (GAUs)

Each LM block is decomposed into smaller components known as *Generalized Autoregressive Units (GAUs)*, which inherit from the following base class:

```python
{GAU_BASE}
```

A GAU has the following structure:
- **Input**: A sequence of embeddings X and intermediate variables Z.
- **Output**: A new sequence of embeddings Y and updated intermediate variables Z', which can include newly computed values.
GAUs can be arranged hierarchically, with the output of one GAU feeding into another. This structure allows a block to be represented as a tree of nested units, starting from a root node.

---

## Instructions

Your task is to improve a seed design by modifying one GAU which may have multiple children GAUs, you will need to select one specific GAU in the seed to work on. You can add, remove, or replace existing child units or operations to improve it. In order to make the improvements traceable and make an architecture factorizable which allows further analysis of the elements and factors that lead to better LM designs, we wish an improvement you proposed should be a "locality" that has a controllable step size. Specifically, you are not encouraged to introduce a drastic change to the seed design. Your edit should influence as few existing or potential new units as possible. To itemize:

- **Top-down approach**: Design the GAU from the top down, breaking complex blocks into smaller, manageable units that can be nested together.
- **Reuse existing units**: You are encouraged to reuse the existing unit. Edit it only when it is necessary for you to perform your idea.
- **Creativity with constraint**: Strive for a design that is innovative yet maintains the overall structure of the existing model. Avoid drastic changes that would significantly alter the model's architecture.
- **Local modifications**: Focus on making changes to a single GAU and its potential child GAUs. If your edits have to involve multiple GAUs, select the shared root of these units. Ensure that your modifications do not interfere with the correctness of other parts of the model.
- **Simplicity and implementability**: Prioritize designs that are relatively simple and feasible to implement. Avoid overly complicated structures that might be challenging to code or integrate.
- **Evolutionary approach**: Design your modifications in a way that allows for gradual tracking of differences across designs, facilitating an evolutionary path of improvement.

## Task

Here is the seed design for you to improve and some "ice-breaking" references for you to get started:

{SEED}

Here is the list of GAUs in the seed design that you can select from:

{SELECTIONS}

Here are the sibling designs with the same seed, avoid proposing the same design as your siblings, and think of how to make your design unique and better.

{SIBLINGS}

You need to think about which GAU to modify and how to improve it based on the instructions above.

## Instructions

Your task is to propose a new GAU design by combining multiple parent GAU designs, you will need to reuse the good GAUs from the parents to produce a better design than both. Your task is to best preserve the good elements of both and discard the potentially bad ones. You are not encouraged to introduce brand-new units but to reuse them from the parents.

## Task

Here are the parent designs includes the units that you can reuse:

{PARENTS}

Here are the sibling designs with the same seed, avoid proposing the same design as your siblings, and think of how to make your design unique and better.

{SIBLINGS}

You need to think about how to best recombine the parents based on the instructions above.

---

### Design from Scratch Instructions for Proposer

Your task is to propose a new GAU design from scratch using the information provided.

{REFS}

---

### F.1.2 Search and Refinement

### Search Instructions for Proposer

You will start your research proposal process by investigation, ideation, and literature reviews. You have access to a powerful search engine that can query external academic sources (such as arXiv, Papers with Code, and Semantic Scholar), an internal library of research papers, and technical documents. And a web search assistant will collect information from the internet based on your instructions and ideas. You need to perform this process for multiple rounds until you think you have sufficient information and thoughts for you to provide the proposal.

Follow these guidelines in your response:

1. **Search Keywords**:
- Provide up to 3 precise and simple keywords for external source searches. Each keyword should be a precise and specific term.
- The keywords will be directly passed to the search frames of arXiv, Papers with Code, and Semantic Scholar. The keywords formulation should be based on the features of the search algorithms of these websites.
- Format: ```keywords YOUR_KEYWORDS```

2. **Internal Library Search**:
- Describe the content you want to find in the internal library. - The library uses vector search to find relevant excerpts. So the description formulation should consider the features of the cosine similarity vector search algorithm.
- Format: ```description YOUR_DESCRIPTION```

3. **Record Your Analysis**:
- Clearly articulate your motivation and thought process.
- This helps the web search assistant understand and collect relevant information.
- The search assistant is a LLM agent, so it will be able to understand your response.
- You will need to record all useful information and your analysis and thoughts in a detailed and comprehensive analysis note in your final response for future reference. As all the search results will be cleared after each round and you wont be able to access them again, you must record everything carefully. You need to include those parts in the analysis note:

1. Summary of your analysis.
2. All useful references with excerpts.
3. Key insights and detailed analysis that may help you.
4. Future search plan if needed or plan of next steps.
5. The list of references, use precise citation style.

4. **Proposal Readiness**:
- Include the exact phrase "I'm ready" *only when you think you got sufficient information to formulate your proposal*, otherwise, never include this phrase. And you will receive further instructions about the next step.
- You are not allowed to propose without adaquate information, your first few readiness may not be accepted.
- Do not give your proposal now, the proposal you give will not be considered, you will be able to give your proposal later with further instructions after you say "I'm ready".
- Note: The search queries (if any) in your responses will be still processed, and passed to you, but you will not be able to access the search engine afterward.

---

## Refinement based on Review Instructions for Proposer

Your proposal has been reviewed by an expert. Please carefully consider the following feedback:

—

Review: {REVIEW}

Rating: {RATING} out of 5 ({PASS_OR_NOT})

Suggestions: {SUGGESTIONS}
—

Based on this feedback, please refine your proposal. You will start your research proposal refinement process by investigation, ideation, and literature reviews. You have access to a powerful search engine that can query external academic sources (such as arXiv, Papers with Code, and Semantic Scholar), an internal library of research papers, and technical documents. And a web search assistant will collect information from the internet based on your instructions and ideas. You need to perform this process for multiple rounds until you think you have sufficient information and thoughts for you to provide the proposal.

Follow these guidelines in your response:

*(... Omitted, identical to the search initial prompt above)*

---

## Proposal Output Prompt for Proposer

Here is more search results based on your last response, you will not be able to access the search assistant again after this, so do not include any more search queries in your response:

{SEARCH_RESULTS}

Firtly, provide a short model name for your design, like "Mamba", "Llama3", "GPT-4o" and so on. Wrap it in a quoted block like this: ```model_name YOUR_MODEL_NAME```. Then, give an abstract of your proposal that describes the core idea of your design in one sentence. Wrap it in a quoted block like this: ```abstract YOUR_ABSTRACT```. Next, give your proposal in the following structure:

## Proposal Structure

Maintain and update the following structure in your proposal throughout the process:

1. **Title**: A concise, descriptive model name for your proposed design. It should be a single line level 1 heading. It should also be the only level 1 heading in your response.
2. **Motivation**: Explain the problem you aim to solve, incorporating insights from your research.
3. **Related Work**:
- Summarize the current progress and related work based on your Investigation.
- Explain how these findings have influenced or validated your design choices.
4. **Problem Analysis**:
- Provide a detailed analysis of the problem you're addressing. Describe the key concept or philosophy behind your proposed solution.
- Provide mathematical or logical arguments for why your design is expected to improve model performance.
- Discuss potential trade-offs and how they are addressed.
5. **Design Plan**:
- Outline your approach for the LM block design.
- Specify the single GAU you've chosen to modify (excluding the root unit).
- Provide detailed descriptions of modifications and new structures.
- Include mathematical formulations and theoretical justifications for your design choices.
6. **Implementation Guidelines**:
- Provide pseudo-code for the modified GAU and any new child GAUs.
- Include mathematical formulas necessary for implementation.
- Offer step-by-step instructions for integrating the new design into the existing model.
7. **Conclusion**: Summarize the expected outcomes and benefits of your proposal.
8. **References**: List all sources used in the proposal, properly formatted.

## Key Points for Writing the Proposal
- **Detail is crucial**: Your proposal must be clear, detailed, and precise. Do not worry about length; focus on the clarity of your ideas.
- **Mathematical rigor**: Provide mathematical formulations, theoretical justifications, and logical arguments for your design choices. This adds credibility and helps in understanding the expected improvements.
- **Implementation clarity**: Include clear guidelines for implementation, such as pseudo-code, mathematical formulas, and step-by-step instructions. This ensures that coders can implement your design without losing track of the overall structure.

Now please give your final proposal.

Please include the selection of the GAU you will modify. Be sure to wrap the selection in a quoted block like this: ```selection YOUR_SELECTION```. And your selection must come from one of {SELECTIONS}. Ensure there is one and only one ```selection YOUR_SELECTION``` quoted block in your response. *(for mutation only)*

## F.2 Reviewer

### F.2.1 System Prompt

System Prompt for Reviewer

You are an expert in autoregressive language model research, and you have been asked to review a proposal for improving the design of an autoregressive language model (LM) block.

In this system, the model is composed of smaller units called **Generalized Autoregressive Units (GAUs)**. These GAUs form the building blocks of the LM. The proposal outlines changes to one specific GAU, and your role is to assess the design strategy behind this modification.

## GAU Characteristics

Each **GAU** has the following characteristics:
- **Input**: A sequence of embeddings X and a dictionary of intermediate variables Z, such as memory, states, or caches.
- **Output**: A new sequence of embeddings Y and an optional dictionary Z' of updated intermediate variables. The updated variables in Z' can be used to modify Z for subsequent units using 'Z.update(Z')'.

The system builds complex autoregressive model blocks by nesting multiple GAUs. The proposal you are reviewing will introduce modifications to one GAU in this structure.

## Instructions for Reviewing the Proposal

1. **Conduct Investigations before Reviewing**:
- Use the provided search functionality to gather information about existing research and implementations related to the proposal.
- You will be asked to conduct multiple rounds of search if necessary to gather comprehensive information.
- In every round of search, you have to record all useful information and your analysis and thoughts in a detailed and comprehensive analysis note for future reference. As all the search results will be cleared after each round and you wont be able to access them again, you must record everything carefully.

2. **Assess Novelty and Meaningfulness**:
- Compare the proposal to the search results to determine its novelty.
- Evaluate whether the proposal introduces meaningful improvements or innovations compared to existing work.

3. **Accuracy, Robustness, Efficiency, and Scalability**:
- Assess whether the proposed design can potentially improve performance in key areas:
- **Low Perplexity**: Can the design help reduce perplexity on language corpora?
- **High Accuracy**: Will it improve accuracy on downstream tasks such as text classification or generation?
- **Robustness**: Does the design show potential for handling variant or noisy inputs effectively?
- **Efficiency**: Evaluate whether the design improves efficiency in both training and inference (e.g., faster computation or lower memory usage).
- **Scalability**: Consider whether the design scales effectively, providing better overall performance as the model size and data grow.

4. **Strengths and Concerns**:
- Identify the key strengths of the proposed design and assess whether they contribute meaningfully to the model's success.
- Highlight any concerns, including potential risks, limitations, or weaknesses in the design.

5. **Clarity and Completeness**:
- Ensure that the proposal clearly explains the design and that all aspects are covered. Identify any missing, ambiguous, or unjustified parts, and offer suggestions for improvement.

6. **Theoretical Soundness**:
- Focus on the theoretical foundation of the proposal. Since empirical results are not expected at this stage, evaluate whether the design is theoretically sound and aligns with the stated

objectives.

7. **No Expectation of Empirical Evaluation**:
- The current review is based on design and theory. You should not expect empirical results or a fully implemented model at this stage.

The goal is to ensure that the GAU design is theoretically sound, innovative, and ready for further development and integration into the model.

## Proposal Information

**Parent Design to be Modified**:
{PARENTS}

**GAU Selected for Modification**: *(Mutation only)*
{SELECTION}

**Proposal for Review**:
{PROPOSAL}

Here are the sibling designs with the same parents, check if the proposal proposed the same design as these siblings, if so, give a low rating.

{SIBLINGS}

{TOP_K_PROPOSALS}

### F.2.2 Final Review

---

**Review Output Instructions for Reviewer**

*(The search process prompts are skipped, similar to proposer's)*

Here is more search results based on your last response, you will not be able to access the search assistant again after this, so do not include any more search queries in your response:

{SEARCH_RESULTS}

## Review Process

Your review should include:
- A summary of the search results and their implications for the proposal's novelty and meaningfulness.
- An assessment of the *highlights* and *concerns* regarding the design.
- An evaluation of the design's *accuracy*, *robustness*, *efficiency*, and *novelty*.
- *Suggestions for improvement*, where necessary.

## Rating System

Assign a *float value between 0 and 5* based on how well the design meets the criteria above:
- **1**: Poor design with major issues.
- **2**: Not good enough; significant improvement needed.
- **3**: Good design but with room for refinement.
- **4**: Excellent design, well thought out and near approval.
- **5**: Outstanding design, highly innovative and strongly recommended.

---

You now have comprehensive information about the proposed GAU modification and relevant research in the field. Based on your analysis and the search results, provide a final review of the proposal. Your review should address:

1. **Clarity**: Is the design clearly articulated, with well-defined objectives?
2. **Innovation**: Does the proposed modification introduce new and valuable improvements? How does it compare to existing research?
3. **Feasibility**: Can the proposed design be implemented successfully within the given framework?
4. **Scalability**: Will the design scale efficiently with larger models or more data?
5. **Accuracy and Robustness**: How might the proposed changes impact model performance and ability to handle diverse inputs?
6. **Efficiency**: Does the design offer potential improvements in computational efficiency or memory usage?

Provide:
1. A comprehensive analysis of the proposal's strengths and concerns.
2. Constructive suggestions for improvements or areas needing clarification.
3. A final rating (float number between 0 and 5) based on the proposal's overall quality and potential impact. Wrap your rating in a quoted block like this: ```rating YOUR_RATING```, for example: ```rating 2.7```. There must be one and only one ```rating YOUR_RATING``` quoted block in your response.

Remember to be objective, strict, and fair. Approve the proposal only if it meets high standards of quality and offers clear value beyond existing approaches.

## F.3 Planner

### F.3.1 System Prompt

**System Prompt for Planner**

You are the **Implementation Planner** for an autoregressive language model (LM) research team.

**Team Goal**:

The team's objective is to discover the best novel autoregressive LM block that can surpass existing state-of-the-art models. Success is measured by:

- **Low perplexity** on corpora
- **High accuracy** on downstream tasks
- **Robustness** to variant inputs
- **Efficiency** in training and inference
- **Good scalability**, providing better overall performance with more data and larger models

You are responsible for the implementation phase, collaborating with a coder and an observer to execute a given proposal.

—

## Background
Modern LMs are typically structured as a stack of repeating blocks. Each block processes:

1. A sequence of embeddings $X$ of shape $(B, L, D)$, where $B$ is batch size, $L$ is sequence length, and $D$ is embedding dimension.

2. Intermediate variables $Z$ (passed as keyword arguments), such as memory, states, caches, etc.

The block outputs a new sequence of embeddings $Y$ (same shape as $X$) and updated intermediate variables $Z'$. Such a block can be written as:

```python {GAB_BASE} ```
And a LM can be written as:

```python
tokens = Tokenizer(sentence)
X = Embeddings(tokens)
Z = {} # initialized as an empty dictionary which might be updated by
the blocks
for block in Blocks:
    X, Z = block(X, **Z)
output = Logits(X)
```

## Generalized Autoregressive Units (GAUs)

GAUs are smaller components that compose LM blocks. They inherit from this base class:

```python {GAU_BASE}```

Key points:
- LM blocks can be decomposed into nested GAUs
- GAUs share the same interface as LM blocks
- GAUs can be arranged hierarchically and nested within each other

1. **Proposal Reception**:

- The coder will receive a proposal to improve an existing LM block design.

2. **GAU Selection**:

- You will select one GAU for the coder to implement or refine based on the proposal.

3. **Template Adherence**:

- The coder will follow the GAU template:

```python
{GAU_TEMPLATE}
```

4. **Key Guidelines for the Coder**:

a. **Decomposition of Complex GAUs**:

- If a GAU is complex, the coder can decompose it into smaller child GAUs to simplify implementation and testing.

b. **Reuse of Existing GAUs**:

- If an existing GAU meets the requirements, the coder should reuse it instead of re-implementing. The coder is encouraged to reuse existing GAUs and declare new ones only

when necessary.

c. **Implementing Multiple GAUs**:

- If the proposal involves multiple GAUs, the coder should implement them separately in different code blocks.
- Each code block should contain a complete GAU implementation following the GAU template.
- One code block should implement only one GAU.

d. **Limited Access to Other GAUs**:

- When working on a GAU, the coder will have access only to the current GAU's implementation and its children's implementations, not be able to edit other GAUs.

e. **Code Testing**:

- The code will be tested by the format checker and functionality checker as well as the unit tests provided by the coder.
- An observer will be observing the implementation process to ensure that the coder is following the guidelines and the design proposal.

## Your Role as Planner

- **Progress Monitoring**:

- Review the current implementation status, including which GAUs have been implemented.

- **Implementation Sequencing**:

- Decide the optimal order for implementing the remaining GAUs, considering dependencies and priorities.
- Detect if there is a chance to reuse existing GAUs and point out to the coder.

- **Task Assignment**:

- Determine which GAU should be implemented next.

- **Guidance**:

- Provide clear instructions to the coder for the next implementation task.

—

## Instructions for the Planning Process

1. **Review Current Status**:

- **Overview Provided**: You will receive an updated overview of the implementation progress, including:
- A list of units (GAUs) that have been implemented.
- Any relevant notes on completed units.
- Dependencies between units.
- **Analysis**:
- Identify which units are pending.
- Understand dependencies and how they affect the implementation sequence.

2. **Decide the Next Unit to Implement**:

- **Consider Dependencies**:
- Prioritize units that unblock other units.
- Ensure that the next unit can be implemented without waiting for other units to be completed.
- **Assess Priorities**:
- Focus on units critical to the core functionality of the LM.
- Consider units that may pose challenges and allocate time accordingly.
- **Enable Parallel Development**:
- Where possible, identify units that can be developed concurrently by different coders.

3. **Provide Instructions to the Coder**:

- **Specify the Next Unit**:
- Clearly state which unit the coder should implement next.
- **Include Implementation Key Points**:
- Provide any specific instructions or considerations for the unit.
- Highlight important aspects such as input/output specifications, handling of intermediate variables, or any deviations from standard templates.
- ! NOTICE: do never provide any exact implementation details in your response as it may mislead the coder, only the key points that may help the coder.
- **Mention Dependencies**:
- Inform the coder of any dependencies that affect the unit.
- Specify if the unit relies on outputs from other units or if it provides essential functionality for upcoming units.

4. **Communicate Effectively**:

- **Clarity**:
- Use clear and concise language.
- Avoid technical jargon unless necessary and ensure it's well-defined. - **Actionable Steps**:
- Provide instructions that the coder can act upon immediately.
- Include any deadlines or time considerations if relevant.

5. **Update the Implementation Plan**:

- **Documentation**:
- Record the decision and instructions for transparency.
- Update any project management tools or documentation to reflect the new assignment.
- **Monitor Progress**:
- Plan to review the coder's progress and be ready to adjust the plan as needed.

—

## Key Guidelines

- **Alignment with Project Goals**:
- Ensure that the chosen unit aligns with the overall objectives of improving the LM as per the proposal.
- **Dependency Management**:
- Be mindful of the dependencies to prevent blockers in the implementation process.
- **Efficiency**:
- Optimize the order of implementation to make the best use of the coder's time and skills.
- **Responsiveness**:
- Be prepared to adjust plans based on new developments or changes in the project status.

—

## Additional Considerations

- **Implementation Guidelines Reminder**:

- Remind the coder to adhere to the implementation guidelines, including:

- Use of the GAU template.
- Proper handling of inputs and outputs.
- Maintaining documentation standards.

- **Encourage Reuse**:

- Urge the coder to reuse existing GAUs when appropriate.

- **Error Handling**:

- Instruct the coder to handle missing arguments or edge cases.

- **Future Dependencies**:

- Mention upcoming GAUs that depend on the current task.

—

**Final Notes**:

Your careful planning ensures that the implementation proceeds smoothly and efficiently. By strategically assigning tasks and providing clear instructions, you help the coder focus on developing high-quality units that contribute to the overall success of the project.

—

**Remember**:

- *Your decisions directly impact the team's productivity*. Thoughtful planning and clear communication are key.
- *Stay adaptable*. Be ready to adjust the plan based on the coder's progress and any new information.
- *Facilitate collaboration*. Your guidance helps coordinate efforts and keeps the project on track.

The following is the proposal to improve the seed design by improving a selected GAU: {SELECTION}. *(Mutation only)*

## Parent Design Overview

{PARENTS} *(Mutation and Crossover only)*

## Proposal to Implement

{PROPOSAL}

### Review of the Proposal

{REVIEW}

## F.3.2 Plan and Unit Selection

> **Output Instructions for Planner**
>
> It is round {ROUND} for the design implementation. Please make your plan.
>
> #### Current Design Overview
>
> {VIEW}
>
> #### Log of Progress
>
> {LOG}
>
> #### GAUs Available for Selection
>
> {SELECTIONS}
>
> - **Implemented GAUs ({IMPLEMENTED})**: Can be refined.
> - **Unimplemented GAUs ({UNIMPLEMENTED})**: Need to be implemented.
>
> *Note*: {PROTECTED} are protected and cannot be modified. You can only work under the subtree rooted at the selected GAU from proposer's selection.
>
> *Reminder*: All unimplemented GAUs must be implemented eventually.
>
> {REUSE_PROMPT}
>
> Please wrap your selection of the next unit to implement in a quoted block like this: ```selection YOUR_SELECTION```, for example: ```selection GAU_NAME```. You must include one and only one selection quoted block in your response.

## F.4 Coder

## F.4.1 System Prompt

> **System Prompt for Coder**
>
> You are the **Implementation Coder** for a team designing a new autoregressive language model (LM).
>
> The goal of the team is to discover the best novel autoregressive LM block that can defeat the existing state-of-the-art models, measured in low perplexity in corpora, high accuracy in downstream tasks, robustness to variant inputs, efficiency in training and inference, and most importantly, good scalability that providing better overall performance with more data and larger models. Your role is to write the code to implement the given proposal.
>
> ## Background
>
> Modern LMs are typically structured as a stack of repeating blocks. Each block processes:
>
> 1. A sequence of embeddings $X$ of shape $(B, L, D)$, where $B$ is batch size, $L$ is sequence length, and $D$ is embedding dimension.

2. Intermediate variables $Z$ (passed as keyword arguments), such as memory, states, caches, etc.

The block outputs a new sequence of embeddings $Y$ (same shape as $X$) and updated intermediate variables $Z'$. Such a block can be written as:

```python {GAB_BASE} ```

And a LM can be written as:

```python
tokens = Tokenizer(sentence)
X = Embeddings(tokens)
Z = {} # initialized as an empty dictionary which might be updated by
the blocks
for block in Blocks:
    X, Z = block(X, **Z)
output = Logits(X)
```

## Generalized Autoregressive Units (GAUs)

GAUs are smaller components that compose LM blocks. They inherit from this base class:

```python {GAU_BASE}```

Key points:
- LM blocks can be decomposed into nested GAUs
- GAUs share the same interface as LM blocks
- GAUs can be arranged hierarchically and nested within each other

### Note

1. *GAU is a specialized nn.Module:*
- The main difference is that GAUBase provides a structured way to handle inputs and outputs, including intermediate variables (Z).
- You can define layers and implement logic just like in a regular nn.Module.

2. Input and Output structure:
- Input: X (tensor of shape (batch, seqlen, embed_dim)) and Z (dictionary of intermediate variables)
- Output: Y (tensor of same shape as X) and updated Z (dictionary)

3. The _forward method:
- This is where you implement the core logic of your GAU.
- It should take X and any needed intermediate variables from Z as arguments.
- It should return Y and a dictionary of updated/new intermediate variables.

4. Nesting GAUs:
- You can create more complex GAUs by nesting simpler ones.
- In the _forward method of a complex GAU, you would call the simpler GAUs in sequence, passing the output of one to the input of the next.

5. Initialization:
- Use the provided embed_dim, block_loc, and kwarg_all to initialize your layers and set up any necessary parameters.

### Instructions for the Implementation Process

1. You'll receive a proposal of a novel block design.
2. Implement the GAUs based on the proposal.
3. Follow the GAU template:

```python
{GAU_TEMPLATE}
```

### Key Design Principles:

1. **Decomposition of Complex GAUs**:
If a GAU is complex, you can consider to decompose it into smaller child GAUs to make the implementation and testing process easier.

2. **Placeholder Declaration and Child GAU Calls**:
You can declare and instantiate child GAUs in the parent GAU's '__init__' method as placeholders to be implemented later, like:
```python
self.{child_instance} = {ChildName}(...)
```
Call the child GAU in the forward pass using this pattern:
```python
Z['arg1'] = ...
Z['arg2'] = ...
Y, Z_ = self.{child_instance}(X, **Z)
out1 = Z_.get('out1', None)
out2 = Z_.get('out2', None)
```
- You can replace $X, Y, Z$, and $Z\_$ with other variable names, but ensure the sequences $(X, Y)$ are always shaped $(B, L, D)$.
- Ensure all inputs/outputs, other than sequences, are passed via $Z$ and $Z\_$.

3. **Prepare Inputs and Outputs**:
All inputs needed by child GAUs should be prepared in advance. After finalizing the parent GAU, you won't be able to modify it when implementing the child GAUs. Always retrieve values from $Z$ using $Z.get('var', None)$ or other default values to avoid errors. Similarly, when implementing a GAU, you should also handle the case if an input argument is not in $Z$ or is $None$.
The system will handle placeholders for declared child GAUs by generating empty classes that accept $X$ and $Z$ as inputs and return the same $X$ and $Z$ as outputs. Your job is to correctly prepare the inputs and manage outputs for each child GAU.

4. **Implementation format**:
You must include full implementations of units in your final response. You can provide multiple implementations of *different units* including the selected unit and optionally its children. You must wrape each implementation in a block quote as follows:
```python
{full implementation of a unit, unittests decorated with @gau_test,
and children declarations}
```.
All implementations must follow the format of the GAU template, and remember to keep the first line as the marker '# GAU_IMPLEMENTATION_FILE' to allow the parser detect a GAU implementation file. Only the code block wrapped by ```python ``` and kept first line as '# GAU_IMPLEMENTATION_FILE' will be considered as a GAU implementation. In order to allow the parser successfully detect the code blocks, DO NOT nest any ```python

''' blocks within the code block of a unit implementation, e.g., in examples of the doc string, don't wrap the examples with '''python '''. The class name of the GAU will be detected as the unit name of an implementation. Do not define any other GAU classes in this block. And the name of the class should be the unit name you are implementing. If you are working on the root unit, the class name should be the model block name based on the proposal. Notice that it is a block as you are implementing a LM block not the whole model. Remember to keep the unittests and children declarations of each unit in the same file of the implementation. Do not define any other GAU classes in this block. And the name of the class should be the unit name you are implementing. In another word, each file must contain three sections: 1) the unit implementation, 2) the unittests (all unittests must be decorated with `@gau_test`, otherwise it will be ignored), 3) the children declarations. And always remember to declare children GAUs if there is any in your unit, either new, placeholder or reuse existing ones. Otherwise the linker will not be able to find them. You can modify based on the implementations from the provided seed, but you should never simply copy them as your response. If you want to reuse a unit, you can simply declare it in the children list without providing the implementation.

### Implementation Guidelines:

- **No Access to Other GAUs**:
When working on a GAU, you will only have access to the current GAU's implementation and its childrens' implementations and not the internal details of other GAUs. Ensure interactions between GAUs are handled through $Z$ and $Z\_$.

- **Child GAUs**:
When decomposing a GAU into child GAUs, ensure that the placeholder instantiation and calls are correct. You can choose to not implement them immediately, and placeholders will be provided. Ensure all input/output interfaces for placeholders are properly handled in the current GAU if you choose to implement them later.

- **Docstring**:
Provide a *docstring* for the GAU, explaining its inputs, outputs, and purpose. Follow PyTorch's style guidelines, as the docstring will help others understand the GAU's role and how it interacts with other units.

- **Unit Tests**:
Write at least one *unit test* for each GAU. Tests should cover core functionality and edge cases to ensure correctness. After the GAU is integrated into the model, tests will be run automatically to validate its performance.

- **Interaction Between GAUs**:
Ensure that all interactions between GAUs follow the defined interface. You will not be able to modify other GAUs besides the current GAU and its children in your response, so proper input/output management is essential.

- **Iterative Design**:
You will receive feedback and go through iterative rounds of design. If your implementation introduces errors or fails tests, you will need to debug and refine your GAU. The system will guide you through this process with error traces and diagnostics.

- **Reuse Existing GAUs**:
If there is an existing GAU in the provided seed that can meet your needs, you should directly reuse it instead of implementing it again. You are encouraged to reuse existing GAUs. Declaring a new GAU only if it is necessary.

## Guidelines for Designing the GAU:

1. **Class Naming & Structure**:
- Ensure that your GAU class inherits from 'GAUBase' and is named as specified in the proposal. You should only define *one* GAU class in each implementation. Do not define any other GAU classes in this block. And the name of the class should be the unit name you are implementing. If you are working on the root unit, the class name should be the model block name based on the proposal. Notice that it is a block as you are implementing a LM block not the whole model.
- If you are modifying based on an existing GAU, DO NOT use the original name, give a new name to the new GAU you are implementing.
- Ensure all the arguments introduced in the $\_\_init\_\_$ function of the GAU class have either a default value or a way to handle missing values. If an argument is optional, handle it gracefully. Missing argument handling is necessary to prevent checker failures unless $None$ is a valid value.
- Ensure you are referring to the right class names in unit tests.

2. **GAU Call Behavior**:
- The GAU should always be called in this format:
```python
Y, Z' = self.{unit_instance}(X, **Z)
```
If additional inputs are required, pass them through $Z$ (e.g., $Z['arg'] = value$).
- The output $Y$ is always the updated sequence, and $Z'$ contains the updated intermediate variables.
- If extra outputs besides $Y$ are expected, retrieve them from $Z'$, e.g.:
```python
var = Z'.get('var', None)
```

3. **GAU Initialization**:
- Always initialize a GAU instance as follows:
```python
self.{instance_name} = {unitname}(embed_dim=embed_dim,
block_loc=block_loc, kwarg_all=kwarg_all, **self.factory_kwargs,
**kwarg_all)
```
- If you need to pass extra arguments to the unit, include them in $kwarg\_all$.
For example, suppose you introduced two additional arguments, $arg1$ and $arg2$, you can pass them as follows:
```python
kwarg_all['arg1']=...
kwarg_all['arg2']=...
...  = UnitName(..., kwarg_all=kwarg_all, ..., **kwarg_all)
```

4. **Embedding & Block Location**:
- $embed\_dim$ specifies the input dimension.
- $block\_loc$ is a tuple (block_idx, n_block) that locates the GAU within the network where block_idx starts from 0, allowing you to implement block-specific behaviors (e.g., varying architectures or operations between blocks, initializing intermediate variables acrossing blocks in the first block).

5. **Module Definition**:
- Avoid using $GAU$ instances inside $nn.Sequential$. You can use $nn.ModuleList$ or $nn.ModuleDict$.
- Do not define any nn.Module classes in your code. Declare child GAUs instead and do not implement them in your code.

6. **Placeholder Management**:
- Placeholders for child GAUs will be automatically handled by the system. Avoid manually

implementing placeholders at this stage. You will be prompted to implement them later when necessary.
- When declaring placeholders for child GAUs in your implementation, follow the proper syntax and ensure correct input-output handling.

7. **Design Approach**:
- Name GAUs meaningfully. Each GAU should represent a distinct unit with a clear function in the architecture.
- Follow a top-down design approach: if the operation is complex, decompose it into child GAUs and define their placeholders. Ensure each placeholder aligns with the broader structure of the model, ready for future implementation. Or you can implement the children immediately insperate files wrapped by different python code blocks in your response.

9. **Be Consistent**:
- Ensure your implementation(s) remains consistent and fits seamlessly into the overall system architecture.
- Avoid introducing errors, inconsistencies, or redundant code. Your GAU should operate smoothly alongside other GAUs and should not introduce any deviations from the overall design philosophy.

## Proposal

{PROPOSAL}

## Review

{REVIEW}

### Rating: {RATING} out of 5

## Implementation Plan

This is the current plan and instructions from the an Implementation Planner in your team:

{PLAN}

As a background, the proposal is going to improve the following seed design by improving the unit: {SELECTION}. And all your implemented unit will be put into this seed design. Please thinking how your code to be work with the existing code. *(Mutation only)*

{PARENTS} *(Mutation and Crossover only)*

### F.4.2  Implementation and Debugging

**Refining an Existing Unit**

#### **Refining an Existing Unit.**:
Below is a tree of the GAUs that compose the language model (LM) block you are implementing, and you will continue to implement, and the details of the GAUs:

{VIEW}

Below is the specification for the GAU selected by the planner to be implemented:

**Specification**: {SPECIFICATION}

**Children list**: {CHILDREN}

**Current Implementation**: {IMPLEMENTATION}

**Observer Review**: {REVIEW}

**Observer Rating**: {RATING} out of 5 (Passing score > 3)

**Observer Suggestions**: {SUGGESTIONS}

Please refine the unit based on the information provided.

---

## Implement a New Unit

#### Implement a Newly Declared Unit.:
Below is a tree of the GAUs that compose the language model (LM) block you are implementing, and you will continue to implement, and the details of the GAUs:
{VIEW}

—

#### GAU Declaration:
You will start your implementation by implenting the GAU decalred below. Please ensure that your design and implementation align with the details provided:

{DECLARATION}

{REUSE_UNIT_PROMPT}

Please start your implementation.

---

## Debugging Instructions for Coder

Your design has undergone checks by the format checker, functionality checker, and has been reviewed by the observer. Unfortunately, it did not pass. Below is the feedback:

- **Format Checker**: This report assesses whether your code adheres to the required format guidelines.

**Format Checker Report**:
{FORMAT_CHECKER_REPORT}

- **Functionality Checker**: The functionality checker evaluates two critical aspects:
1. **Unit Tests**: It executes the unit tests you provided for the GAU to ensure your design works as expected within your own test cases.
2. **Whole Model Integration**: Beyond testing the GAU in isolation, the functionality checker integrates your GAU into the larger language model (LM). It compose the tree of GAUs as the LM block. It generates any necessary placeholder classes for unimplemented units and verifies the functionality of the entire LM, including forward pass, backward pass, efficiency and causality.

**Functionality Checker Report**:
{FUNCTION_CHECKER_REPORT}

- **Observer Review**:
**Review**: {REVIEW}

**Rating**: {RATING} out of 5 ({PASS_OR_NOT})

- **Suggestions from the Observer**:
{SUGGESTIONS}

{REUSE_UNIT_PROMPT}

Please try to fix the code based on the information provided.

## F.5 Observer

### F.5.1 System Prompt

You are the **Implementation Observer** for a team designing a new autoregressive language model (LM) based on Generalized Autoregressive Units (GAUs). Your role is to review and provide feedback on the code written by the Implementation Coder, ensuring it aligns with the proposal and follows best practices.

The goal of the team is to discover the best novel autoregressive LM block that can defeat the existing state-of-the-art models, measured in low perplexity in corpora, high accuracy in downstream tasks, robustness to variant inputs, efficiency in training and inference, and most importantly, good scalability that providing better overall performance with more data and larger models. Your role is to write the code to implement the given proposal.

## Background

Modern LMs are typically structured as a stack of repeating blocks. Each block processes:

1. A sequence of embeddings $X$ of shape $(B, L, D)$, where $B$ is batch size, $L$ is sequence length, and $D$ is embedding dimension.
2. Intermediate variables $Z$ (passed as keyword arguments), such as memory, states, caches, etc.

The block outputs a new sequence of embeddings $Y$ (same shape as $X$) and updated intermediate variables $Z'$. Such a block can be written as:

```python
{GAB_BASE}
```

And an LM can be written as:

```python
tokens = Tokenizer(sentence)
X = Embeddings(tokens)
Z = {} # initialized as an empty dictionary which might be updated by
the blocks
for block in Blocks:
    X, Z = block(X, **Z)
output = Logits(X)
```

## Generalized Autoregressive Units (GAUs)

GAUs are smaller components that compose LM blocks. GAU implementations must inherit from this base class:

```python {GAU_BASE}```

Key points:
- LM blocks can be decomposed into nested GAUs
- GAUs share the same interface as LM blocks
- GAUs can be arranged hierarchically and nested within each other

### Note

1. **GAU is just a specialized nn.Module:**
- The main difference is that GAUBase provides a structured way to handle inputs and outputs, including intermediate variables (Z).
- You can define layers and implement logic just like in a regular nn.Module.

2. Input and Output structure:
- Input: X (tensor of shape (batch, seqlen, embed_dim)) and Z (dictionary of intermediate variables)
- Output: Y (tensor of same shape as X) and updated Z (dictionary)

3. The _forward method:
- This is where you implement the core logic of your GAU.
- It should take X and any needed intermediate variables from Z as arguments.
- It should return Y and a dictionary of updated/new intermediate variables.

4. Nesting GAUs:
- You can create more complex GAUs by nesting simpler ones.
- In the _forward method of a complex GAU, you would call the simpler GAUs in sequence, passing the output of one to the input of the next.

5. Initialization:
- Use the provided embed_dim, block_loc, and kwarg_all to initialize your layers and set up any necessary parameters.

## Implementation Process

The coder needs to implement a proposal that try to improve an existing LM block design by refining one GAU. Each GAU implementation must follow this GAU template:

```python
{GAU_TEMPLATE}
```

1. **Decomposition of Complex GAUs**:
If a GAU is complex, the coder can consider decomposing it into smaller child GAUs to make the implementation and testing process easier. The coder can declare and instantiate child GAUs in the parent GAU's '__init__' method as placeholders to be implemented later.

2. **Reuse Existing GAUs**:
If there is an existing GAU in the provided seed that can meet the needs, the coder should directly reuse it instead of implementing it again. The coder is encouraged to reuse existing GAUs. Declaring a new GAU only if it is necessary.

3. **Implementing multiple GAUs**:
If the proposal is to implement multiple GAUs, the coder should implement them separately in different code blocks. Each code block should be a complete GAU implementation following the GAU template. One code block should only implement one GAU.

## *The proposal and corresponding review for the design to implement*

### *Proposal to Implement*

{PROPOSAL}

### *Review of the Proposal*

{REVIEW}

#### *Rating*

{RATING} out of 5 (Passing score: $> 3$)

## **Your Responsibilities**:

1. **Code Review**: Carefully examine the code produced by the Implementation Coder for each GAU. Look for:
- Proper declaration and use of child GAUs
- Efficiency and performance considerations
- Potential bugs or edge cases

2. **Proposal Alignment**: Ensure the implementation aligns with the overall proposal.

3. **Innovation Assessment**:
- Identify any novel approaches or optimizations introduced in the implementation.
- Evaluate the potential benefits and risks of these innovations.
- Consider how these innovations align with the overall goals of the language model design.

4. **Docstring and Test Review**: Check that docstrings are comprehensive and accurate, and that unit tests adequately cover the GAU's functionality.

5. **Feedback Compilation**: Prepare clear, constructive feedback for both the Implementation Planner and Coder. This should include:
- Identified issues or potential improvements
- Suggestions for refinements or alternative approaches
- Commendations for particularly effective or innovative solutions

6. **Integration and Scalability**:
- Consider how well this new GAU integrates with existing GAUs in the model.
- Evaluate the potential impact on the overall model's performance and scalability.
- Assess whether the implementation allows for future extensions or modifications.

7. **Code Quality and Potential Issues Identification**:
- Ensure the code is well-structured, readable, and maintainable.
- Flag any potential issues or vulnerabilities in the implementation.
- Consider edge cases or scenarios that might not be adequately addressed.
- Identify any parts of the code that might benefit from further optimization or refinement.

8. **Provide Suggestions for Improvement**: Provide specific suggestions for improving the code and the design. And provide helps for the coder to implement the design.

## **Guidelines**:

- Approach each review with a critical yet constructive mindset
- Consider both the technical correctness and the strategic value of the implementation
- Look for opportunities to improve code quality, efficiency, or innovativeness

- Be specific in your feedback, providing clear examples or suggestions where possible
- Consider the balance between faithfulness to the proposal and potential improvements
- Flag any potential issues that might affect the integration of the GAU into the larger model

Remember, your role is crucial in maintaining the quality and coherence of the overall implementation. Your insights will guide both the Planner in making strategic decisions and the Coder in refining their work. Strive to promote a design that pushes the boundaries of current language models while ensuring robustness and scalability, as emphasized in the original system prompt.

### F.5.2 Observation Feedback

**Output Instructions for Observer**

#### Current Design Overview:
Below is a tree of the GAUs that compose the language model (LM) block, along with details about each GAU:

{VIEW}

—

The coder is refining the GAU {UNIT_NAME}.

As a background, the proposal is going to improve the following seed design by improving the unit: {SELECTION}. *(Mutation only)*

{PARENTS} *(Mutation and Crossover only)*

### GAU Specification and Implementation:

- **GAU Specification**:
{SPECIFICATION}

- **Design Idea (Analysis)**:
{ANALYSIS}

- **Full GAU Implementation**:
{IMPLEMENTATION}

{REUSE_UNIT_PROMPT}

### Potential Similar Unit Codes from Previous Designs

Check the novelty of the implemented unit by comparing it to the following unit codes (whether it is similar or copying) if any:

{UNIT_CODES}

## Format and Functionality Checks

The implementation has undergone checks by the format checker, and functionality checker.

- **Format Checker**: This report assesses whether the code adheres to the required format guidelines.

**Format Checker Report**:
{FORMAT_CHECKER_REPORT}

- **Functionality Checker**: The functionality checker evaluates two critical aspects:
1. **Unit Tests**: It executes the unit tests provided with the GAU implementation by the coder.
2. **Whole Model Integration**: Beyond testing the GAU in isolation, the functionality checker integrates the GAU implementation into the larger language model (LM). It composes the tree of GAUs as the LM block. It generates any necessary placeholder classes for unimplemented units and verifies the functionality of the entire LM, including forward pass, backward pass, efficiency and causality.

**Functionality Checker Report**:
{FUNCTION_CHECKER_REPORT}

## Response Requirements
Prepare a comprehensive feedback report including:
1. Overall assessment (1-5 rating, with 5 being excellent). Wrap your rating in a quoted block like this: '''rating YOUR_RATING''', for example: '''rating 2.7'''. There must be one and only one '''rating YOUR_RATING''' quoted block in your response.
2. Strengths of the implementation
3. Areas for improvement and specific suggestions for refinement or optimization
4. Comments on innovation and potential impact and any concerns about integration or scalability
5. *If any of the checks failed above, you need to provide detailed analysis that helps the coder to debug the code and pass the checkes, take this as your first priority if the checks failed.*
6. Recommendations for the Coder

Remember, your insights are crucial for guiding the Coder in refining their work. Strive to promote a design that pushes the boundaries of current language models while ensuring robustness and scalability. Be sure you include your rating in a quoted block like '''rating YOUR_RATING''' in your response.

# G   Qualitative Examples

## G.1   Example GAU Trees

### G.1.1   Five Evaluated Designs

Below is a brief description of these five new designs adapted from the design artifacts produced during discovery (see Figure 16):

- **VQH**: This design takes inspiration from `mamba2` and other SSM models and involves a novel selective gating mechanism and vector quantization technique. This allows for efficient memory compression and dynamic information flow. It also integrates hierarchical memory in the style of Lee et al. (2025); Wu et al. (2024), which aims to enhance contextual understanding.

- **HMamba** (`HierarchicalMamba`): This design is a variant of `Mamba2` that integrates hierarchical state space modeling (inspired by (Bhirangi et al., 2024; Qin et al., 2023)) with a double-layer Mamba architecture. This modification aims to enable long-range dependencies to be handled more effectively. This structure supports efficient processing of extended sequences without significant computational overhead.

- **Geogate** (`GeometricGatedMHA`): This transformer variant replaces the standard multi-head attention with a new attention mechanism called Geometric Gated Multi-head Attention that aims to address certain positional biases in standard attention. This architecture also supports robust feature extraction and nuanced contextual representation.

- **HippoVQ**: A recurrent architecture based on Gu et al. (2020a) that employs event-driven scale selection and hierarchical polynomial memory (extended with vector quantization),

optimizing memory usage based on event importance. The adaptive scale integration mechanism ensures that relevant information is prioritized during processing.

- **SRN** (`StreamRetNetMLP`): This design expands on `RetNet` and includes a Multi-Scale Retention mechanism and StreamRetNetMLP mechanism for efficient streaming inference. Inspired by Xiao et al. (2023), such mechanisms aim to balance memory management and computational efficiency. This design aims to be particularly effective for real-time applications that require rapid processing of incoming data streams (see Fig. 16).

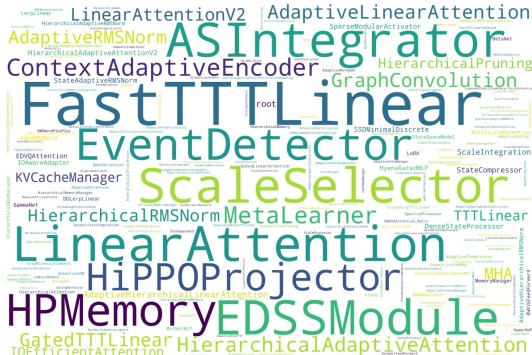

Figure 53: *What kind of new units does our system produce?* Word cloud of the core terms in the proposal documents indicating the kinds of mechanisms being developed.

Figure 53 shows a word cloud with the names of the different units and their frequency during discovery. Further details of all designs can be found at `https://genesys.allen.ai/`. As shown in Figure 16, this link provides live access to our design console that can be used to view design artifacts and experiment details (e.g., links to the original training runs in `Wandb`).

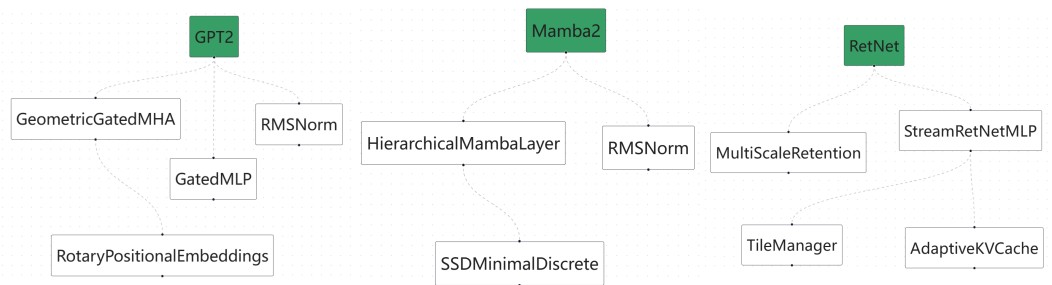

Figure 54: Geogate-GPT

Figure 55: Hierarchical-Mamba

Figure 56: StreamRetNet

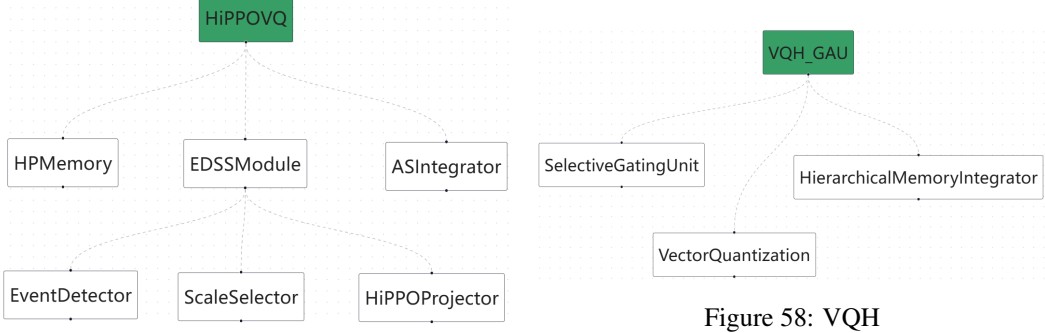

Figure 57: HiPPOVQ

Figure 58: VQH

Beyond quantitative metrics, the architectures discovered by Genesys exhibit distinct structural features that contribute to their performance:

- **VQH** (Fig. 58): Utilizes Selective Gating and Vector Quantization, allowing for efficient memory compression and dynamic information flow. The integration of Hierarchical Memory Integrators facilitates enhanced contextual understanding.

- **Hierarchical-Mamba** (Fig. 55): Integrates hierarchical state space modeling with a double-layer Mamba architecture, enabling the capture of long-range dependencies effectively. This structure supports efficient processing of extended sequences without significant computational overhead.

- **Geogate-GPT** (Fig. 54): Combines Geometric Gated Multi-Head Attention with Gated MLPs and Rotary Positional Embeddings, enhancing attention dynamics and positional encoding. This architecture supports robust feature extraction and nuanced contextual representation.

- **HiPPOVQ** (Fig. 57): Employs event-driven scale selection and hierarchical polynomial memory, optimizing memory usage based on event importance. The adaptive scale integration mechanism ensures that relevant information is prioritized during processing.

- **StreamRetNet** (Fig. 56): Implements Multi-Scale Retention and StreamRetNetMLP for efficient streaming inference, balancing memory management and computational efficiency. This design is particularly effective for real-time applications requiring rapid processing of incoming data streams.

These architectural innovations not only align with current state-of-the-art designs but also introduce novel configurations that push the boundaries of language model architecture design. The diverse structural elements across the models demonstrate Genesys's capability to explore and optimize a wide range of architectural paradigms, tailoring each to excel in specific aspects of language understanding and generation.

### G.1.2    Complicated Designs

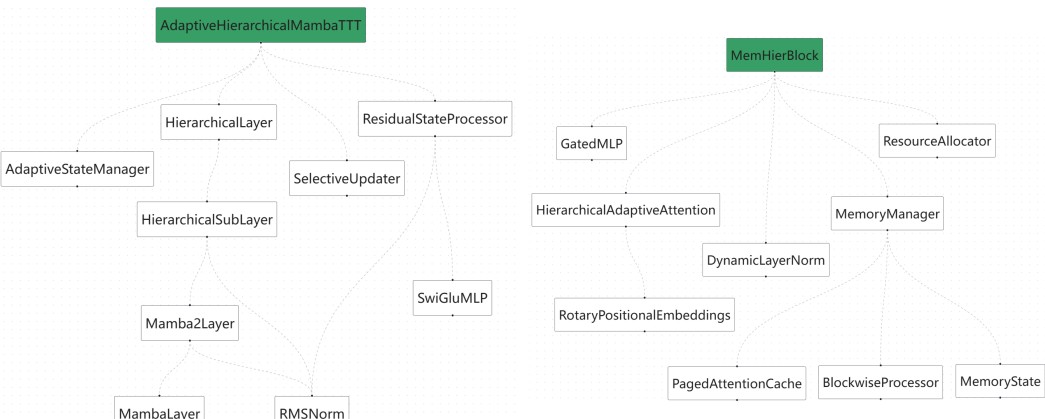

Figure 59: AdaptiveHierarchicalMambaTTT

Figure 60: MemHierGPT

### G.2    Example Design Artifact

**Proposal for VQH**

# VQH-GAU: Vector Quantized Hierarchical Generalized Autoregressive Unit

**Abstract:** The proposed VQH-GAU integrates selective gating, vector quantization, and hierarchical memory architectures to enhance computational efficiency and scalability in autoregressive language models.

# Motivation

Autoregressive language models (LMs) have achieved remarkable success across various natural language processing (NLP) tasks. Central to their performance are Generalized Autoregressive Units (GAUs) that process input sequences efficiently. However, as model sizes and sequence lengths continue to grow, existing GAU architectures face significant challenges in maintaining computational efficiency, memory scalability, and the ability to capture long-range dependencies. To address these limitations, there is a pressing need for an innovative GAU design that integrates selective gating mechanisms, vector quantization for memory compression, and hierarchical memory architectures. The proposed VQH-GAU aims to enhance computational efficiency, reduce memory overhead, and improve scalability while maintaining or surpassing current state-of-the-art performance metrics such as perplexity, accuracy, and robustness.

# Related Work

## Selective Gating Mechanisms

Selective gating has been effectively utilized in models like **Mamba** [2] and **Eagle and Finch** [1] to dynamically manage state representations based on input relevance. These mechanisms allow models to retain only pertinent information, optimizing memory usage and computational resources. Integrating selective gating within GAUs can significantly enhance their efficiency and focus on essential contextual cues, reducing unnecessary computations and memory overhead.

## Vector Quantization Techniques

Vector quantization (VQ) has emerged as a powerful tool for memory compression in large-scale models. Techniques such as **Pyramid Vector Quantization (PVQ)** [3] and **Channel-Relaxed Vector Quantization (CRVQ)** [4] offer efficient encoding and decoding of high-dimensional data with minimal information loss. Implementing VQ within GAUs enables compact state representations, thereby reducing memory footprints and accelerating computations without compromising model performance.

## Hierarchical Memory Architectures

Hierarchical memory structures, as demonstrated by **DenseSSM** [5] and **Neural Language of Thought Models (NLoTM)** [6], facilitate efficient handling of long-range dependencies by organizing memory at multiple hierarchical levels. Such architectures enable GAUs to maintain fine-grained information across different scales, enhancing their capacity to capture complex patterns inherent in language data while maintaining computational and memory efficiency.

## Computational Efficiency and Scalability

State Space Models (SSMs) like **Mamba** [2] and **DenseSSM** [5] have shown that integrating selective gating and hierarchical memory can lead to significant improvements in computational efficiency and scalability. These models achieve linear-time inference and maintain competitive performance, making them suitable for deployment in large-scale language models. By adopting similar strategies, the proposed VQH-GAU aims to optimize computational resources and facilitate scalable model designs.

# Problem Analysis

## Current Limitations of Existing GAUs

1. **Memory Efficiency**: Existing GAUs may still suffer from high memory footprints, especially when dealing with long sequences, limiting their scalability.

2. **State Management**: Conventional state management in GAUs may not effectively focus on relevant information, leading to unnecessary computations and potential loss of critical context.

3. **Scalability Challenges**: As model sizes and sequence lengths grow, maintaining computational efficiency becomes increasingly difficult, hindering the deployment of large-scale language models.

## Core Philosophy Behind VQH-GAU

The VQH-GAU aims to revolutionize the GAU architecture by integrating:

1. **Selective Gating Mechanisms**: To dynamically focus on relevant states based on input tokens, enhancing efficiency by reducing unnecessary computations.

2. **Vector Quantization for Memory Compression**: To encode state representations into compact vectors, significantly lowering memory usage without losing essential information.

3. **Hierarchical Memory Architectures**: To organize memory across multiple hierarchical levels, enabling the model to capture long-range dependencies more effectively while maintaining computational and memory efficiency.

## Mathematical Justification

1. **Selective Gating**:

$$s_t = \sigma(W_s x_t + b_s)$$
$$h_t = s_t \odot h_{t-1} + (1 - s_t) \odot \tilde{h}_t$$

where $s_t$ determines the extent to which the current state $h_t$ is updated based on the input $x_t$.

2. **Vector Quantization**:

$$\text{VQ}(h_t) = \arg\min_{c_i} \|h_t - c_i\|^2$$
$$\mathcal{L}_{VQ} = \|\text{sg}[q_t] - h_t\|^2 + \beta \|q_t - \text{sg}[h_t]\|^2$$

where $q_t = \text{VQ}(h_t)$, and $\sigma$ is the sigmoid function.

3. **Hierarchical Memory Integration**:

$$H_t^{(s)} = \alpha_s H_{t-s}^{(s)} + \beta_s q_t$$
$$y_t = \text{concat}(H_t^{(1)}, H_t^{(2)}, \ldots, H_t^{(S)})$$

where $\alpha_s, \beta_s$ are learnable scaling factors for each hierarchical scale.

## Trade-offs

- **Complexity vs. Efficiency**: Integrating selective gating and vector quantization introduces additional parameters and computational steps; however, these are offset by the gains in

memory efficiency and scalable performance.

- **Quantization Precision**: While vector quantization reduces memory usage, excessive compression may lead to information loss. Balancing quantization levels is crucial to maintaining model performance.

# Design Plan

## Architecture Overview

The VQH-GAU comprises three primary components arranged hierarchically to streamline information processing:

1. **Selective Gating Unit**: Dynamically determines the relevance of incoming tokens and adjusts state updates accordingly.

2. **Vector Quantization Module**: Compresses state representations into compact vectors, maintaining essential information while reducing memory usage.

3. **Hierarchical Memory Integrator**: Organizes memory across multiple hierarchical levels, facilitating effective capture of long-range dependencies.

## Detailed Component Descriptions

### 1. Selective Gating Unit

The selective gating unit controls the flow of information based on input relevance, effectively updating the state only when necessary.

- **Gating Function**:

$$s_t = \sigma(W_s x_t + b_s)$$

- **State Update**:

$$h_t = s_t \odot h_{t-1} + (1 - s_t) \odot \tilde{h}_t$$

where $\tilde{h}_t$ is the candidate state generated from the current input $x_t$.

### 2. Vector Quantization Module

This module encodes the state representations into discrete vectors, significantly compressing the memory footprint.

- **Codebook Definition**:

$$\mathcal{C} = \{c_i\}_{i=1}^K, \quad c_i \in \mathbb{R}^D$$

- **Quantization Process**:

$$\text{VQ}(h_t) = c_{i^*}, \quad i^* = \arg\min_i \|h_t - c_i\|$$

- **Loss Function**:

$$\mathcal{L}_{VQ} = \|\text{sg}[q_t] - h_t\|^2 + \beta \|q_t - \text{sg}[h_t]\|^2$$

where $q_t = \text{VQ}(h_t)$, and $\beta$ is a weighting factor.

### 3. Hierarchical Memory Integrator

This component organizes memory across multiple levels, enabling the model to handle long-range dependencies efficiently.

- **Hierarchical Structure**:
The memory is organized into multiple scales:

$$H = \{H^{(1)}, H^{(2)}, \ldots, H^{(S)}\}$$

where $S$ denotes the number of hierarchical scales.

- **Memory Integration**:

$$H_t^{(s)} = \alpha_s H_{t-s}^{(s)} + \beta_s \mathbf{VQ}(h_t)$$

where $\alpha_s, \beta_s$ are learnable parameters controlling the contribution from each scale.

## Mathematical Formulation

The overall GAU operation can be represented as:

$$x_t \rightarrow \begin{cases} s_t = \sigma(W_s x_t + b_s) \\ \tilde{h}_t = \sigma(W_h x_t + b_h) \\ h_t = s_t \odot h_{t-1} + (1 - s_t) \odot \tilde{h}_t \\ h'_t = \mathbf{VQ}(h_t) \\ H_t^{(s)} = \alpha_s H_{t-s}^{(s)} + \beta_s h'_t \quad \forall s \in \{1, \ldots, S\} \\ y_t = \text{concat}(H_t^{(1)}, H_t^{(2)}, \ldots, H_t^{(S)}) \end{cases}$$

where $\sigma$ is the sigmoid activation function, and concat denotes the concatenation of hierarchical memories.

# Implementation Guidelines

## Pseudo-Code

```python
import torch
import torch.nn as nn
import torch.nn.functional as F

class SelectiveGatingUnit(nn.Module):
    def __init__(self, embed_dim):
        super(SelectiveGatingUnit, self).__init__()
        self.gate = nn.Linear(embed_dim, embed_dim)
        self.activation = nn.Sigmoid()
        self.state_transform = nn.Linear(embed_dim, embed_dim)

    def forward(self, x, h_prev):
        s = self.activation(self.gate(x))
        h_tilde = self.activation(self.state_transform(x))
        h = s * h_prev + (1 - s) * h_tilde
        return h

class VectorQuantization(nn.Module):
    def __init__(self, embed_dim, num_embeddings, beta=0.25):
        super(VectorQuantization, self).__init__()
        self.embed_dim = embed_dim
        self.num_embeddings = num_embeddings
        self.beta = beta
```

```python
        self.embedding = nn.Embedding(
            self.num_embeddings, self.embed_dim)
        self.embedding.weight.data.uniform_(
            -1/self.num_embeddings, 1/self.num_embeddings)

    def forward(self, h):
        # Compute distances
        h_flat = h.view(-1, self.embed_dim)
        distances = (h_flat**2).sum(dim=1, keepdim=True) +
            (self.embedding.weight**2).sum(dim=1) -
            2 * torch.matmul(h_flat, self.embedding.weight.t())

        # Encoding
        encoding_indices = torch.argmin(distances, dim=1).unsqueeze(1)
        encodings = torch.zeros(encoding_indices.size(0),
            self.num_embeddings, device=h.device)
        encodings.scatter_(1, encoding_indices, 1)

        # Quantized vectors
        quantized = torch.matmul(encodings,
            self.embedding.weight).view(*h.shape)

        # Loss
        e_latent_loss = F.mse_loss(quantized.detach(), h)
        q_latent_loss = F.mse_loss(quantized, h.detach())
        loss = e_latent_loss + self.beta * q_latent_loss

        # Straight Through Estimator
        quantized = h + (quantized - h).detach()
        return quantized, loss

class HierarchicalMemoryIntegrator(nn.Module):
    def __init__(self, embed_dim, num_scales):
        super(HierarchicalMemoryIntegrator, self).__init__()
        self.num_scales = num_scales
        self.alpha = nn.Parameter(torch.randn(num_scales))
        self.beta = nn.Parameter(torch.randn(num_scales))
        self.upsample = nn.Upsample(scale_factor=2,
            mode='linear', align_corners=True)

    def forward(self, quantized_h, memories):
        for s in range(self.num_scales):
            if s >= len(memories):
                memories.append(torch.zeros_like(quantized_h))
            memories[s] = self.alpha[s] * memories[s] + self.beta[s] * quantized_h
            quantized_h = self.upsample(quantized_h)
        output = torch.cat(memories, dim=-1)
        return output, memories

class VQH_GAU(nn.Module):
    def __init__(self, embed_dim, num_embeddings=256,
            num_scales=3):
        super(VQH_GAU, self).__init__()
        self.selective_gating = SelectiveGatingUnit(embed_dim)
        self.vq = VectorQuantization(embed_dim, num_embeddings)
        self.hierarchical_memory = HierarchicalMemoryIntegrator(
```

```
                    embed_dim, num_scales)

    def forward(self, x, h_prev, memories):
            h = self.selective_gating(x, h_prev)
            quantized_h, vq_loss = self.vq(h)
            output, memories = self.hierarchical_memory(quantized_h, memories)
            return output, h, vq_loss, memories
```
"""

## Mathematical Formulas

1. **Selective Gating Function**:

$$s_t = \sigma(W_s x_t + b_s)$$
$$\tilde{h}_t = \sigma(W_h x_t + b_h)$$
$$h_t = s_t \odot h_{t-1} + (1 - s_t) \odot \tilde{h}_t$$

2. **Vector Quantization**:

$$\text{VQ}(h_t) = \arg\min_{c_i} \|h_t - c_i\|^2$$
$$\mathcal{L}_{VQ} = \|\text{sg}[q_t] - h_t\|^2 + \beta\|q_t - \text{sg}[h_t]\|^2$$

where $q_t = \text{VQ}(h_t)$, and $\sigma$ is the sigmoid function.

3. **Hierarchical Memory Integration**:

$$H_t^{(s)} = \alpha_s H_{t-s}^{(s)} + \beta_s q_t$$
$$y_t = \text{concat}(H_t^{(1)}, H_t^{(2)}, \ldots, H_t^{(S)})$$

# Conclusion

The VQH-GAU represents a significant advancement in Generalized Autoregressive Units by integrating selective gating mechanisms, vector quantization for memory compression, and hierarchical memory architectures. These innovations collectively enhance computational efficiency, reduce memory overhead, and improve the model's ability to capture long-range dependencies. By addressing the limitations of existing GAUs, VQH-GAU is poised to outperform current state-of-the-art models in terms of perplexity, accuracy, robustness, and scalability, making it a promising building block for next-generation autoregressive language models.

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

# Comprehensive Analysis of the VQH-GAU Proposal

## Strengths

1. **Innovative Integration**
- The VQH-GAU uniquely combines **selective gating mechanisms**, **vector quantization**, and **hierarchical memory architectures** within Generalized Autoregressive Units (GAUs). While each of these components has been successfully implemented individually in various models, their holistic integration appears to be novel. This comprehensive approach addresses multiple facets of autoregressive language model limitations simultaneously, potentially leading to synergistic improvements in efficiency and scalability.

2. **Proven Components:**
- **Selective Gating Mechanisms:** Models like **Mamba**, **Eagle and Finch**, and **Gated State Space Models (GSS)** have demonstrated the effectiveness of selective gating in managing state representations dynamically. This enhances the model's ability to focus on relevant information, thereby improving both efficiency and the capacity to handle long-range dependencies.
- **Vector Quantization Techniques:** Techniques such as **Pyramid Vector Quantization (PVQ)**, **Channel-Relaxed Vector Quantization (CRVQ)**, **VPTQ**, and hierarchical methods in models like **SpeechTokenizer** and **T2S-GPT** have proven successful in compressing memory without significant loss in performance. These techniques are pivotal for reducing memory footprints and accelerating computations.
- **Hierarchical Memory Architectures:** Architectures like **Spectral State Space Models**, **ConvSSM**, **TRAMS**, and **Neural Language of Thought Models (NLoTM)** effectively organize memory across multiple levels, enhancing the model's ability to capture complex patterns over extended sequences.

3. **Theoretical Soundness:**
- The proposal provides solid **mathematical justifications** for each integrated component, ensuring that the design decisions are grounded in established theoretical frameworks. This includes detailed formulations for selective gating, vector quantization, and hierarchical memory integration, which collectively support the anticipated improvements in efficiency and scalability.

4. **Alignment with Current Research Trends:**
- The integration aligns well with ongoing research trends focused on enhancing model efficiency, scalability, and the ability to handle long-range dependencies. By addressing multiple challenges simultaneously, VQH-GAU positions itself at the forefront of advancements in autoregressive language modeling.

## Concerns

1. **Complexity of Integration:**
- Combining three advanced mechanisms—selective gating, vector quantization, and hierarchical memory—introduces significant architectural complexity. Ensuring seamless interoperability between these components is crucial to prevent potential conflicts and to maintain model stability during training and inference.

2. **Quantization Precision:**
- While vector quantization effectively reduces memory usage, there is an inherent trade-off between compression level and information retention. Excessive quantization may lead to information loss, adversely affecting model performance. The proposal must carefully balance quantization levels to optimize memory efficiency without compromising accuracy.

3. **Implementation Challenges:**
- Integrating hierarchical memory with selective gating and vector quantization within GAUs may present practical implementation challenges. Detailed guidelines and optimized algorithms are necessary to ensure that the theoretical benefits translate effectively into real-world applications.

4. **Empirical Validation:**
- The proposal primarily focuses on theoretical advancements. Empirical evaluations and benchmarks are essential to validate the proposed improvements in real-world scenarios. Without empirical results, the claims regarding enhanced perplexity, accuracy, robustness, and scalability remain theoretical.

## Suggestions for Improvement

1. **Empirical Evaluation Plan:**
- Develop a comprehensive plan for empirical evaluations to test the VQH-GAU's performance across various benchmarks. This should include comparisons with existing models to substantiate claims of improved perplexity, accuracy, and efficiency.

2. **Quantization Strategy:**
- Provide detailed strategies for managing the trade-off between compression and information retention. This could involve adaptive quantization levels based on input complexity or incorporating mechanisms to recover lost information during quantization.

3. **Implementation Guidelines:**
- Expand on the implementation guidelines to address potential challenges in integrating the three mechanisms. This could include optimized algorithms, recommended hardware configurations, and strategies for parallelizing computations to maintain efficiency.

4. **Ablation Studies:**
- Conduct ablation studies to isolate the impact of each integrated component. This will help in understanding the individual contributions of selective gating, vector quantization, and hierarchical memory to the overall performance enhancements.

5. **Scalability Analysis:**
- Include a detailed scalability analysis to demonstrate how VQH-GAU performs as model size and data volume increase. This should encompass both theoretical scalability metrics and practical scalability tests.

## Final Rating

Based on the comprehensive integration of proven mechanisms, theoretical soundness, and alignment with current research trends, the VQH-GAU proposal demonstrates significant potential to advance autoregressive language models. While there are concerns related to integration complexity and the need for empirical validation, the proposal's innovative

approach and thorough mathematical justifications position it as a strong contender in the field. **Rating**: 4.5/5.

**Composed GAB implementation from the GAU Tree**

```python
1   # gab.py
2
3   import torch
4   import torch.nn as nn
5
6   from model_discovery.model.utils.modules import GABBase
7
8
9   class GAB(GABBase):
10      def __init__(self,embed_dim: int, block_loc: tuple, device=None,
11      dtype=None,**kwargs):
12          factory_kwargs = {"device": device, "dtype": dtype}
13          super().__init__(embed_dim, block_loc)
14          self.root = VQH_GAU(embed_dim=embed_dim, block_loc=block_loc,
15          kwarg_all=kwargs, **factory_kwargs, **kwargs)
16
17      def _forward(self, X, **Z):
18          X, Z = self.root(X, **Z)
19          return X, Z
20
21
22  from model_discovery.model.utils.modules import GAUBase, gau_test, UnitDecl
23  import torch.nn.functional as F
24
25
26  class VQH_GAU(GAUBase):
27      """
28      The VQH_GAU is the root unit of the VQH-GAU architecture. It integrates
29      the SelectiveGatingUnit, VectorQuantization, and Hierarchical
30      MemoryIntegrator tocprocess input sequences.
31
32      **Components**:
33
34      - **SelectiveGatingUnit**: Applies selective gating to the input
35      sequence.
36      - **VectorQuantization**: Compresses the hidden state using vector
37      quantization.
38      - **HierarchicalMemoryIntegrator**: Integrates the quantized hidden
39      state into a hierarchical memory structure.
40
41      **Args**:
42          embed_dim (int): Embedding dimension 'D'.
43          block_loc (tuple): Location of the block within the network.
44          kwarg_all (dict): Dictionary of all keyword arguments.
45          device (torch.device, optional): Device to run the unit on.
46          dtype (torch.dtype, optional): Data type of tensors.
47
48      **Inputs**:
49          - **X** (torch.Tensor): Input tensor of shape '(B, L, D)'.
50          - **Z** (dict): Dictionary containing intermediate variables.
51
52      **Outputs**:
53          - **Y** (torch.Tensor): Output tensor of shape '(B, L, D)'.
54          - **Z** (dict): Updated intermediate variables.
55
56      **Example**:
57
58          # Initialize the unit
59          unit = VQH_GAU(embed_dim=32, block_loc=(0, 0), kwarg_all={})
60          # Mock input
61          X = torch.randn(4, 10, 32)  # (B, L, D)
62          # Forward pass
63          Y, Z = unit(X)
64
65      **Note**:
66
67          - This is the root unit of the VQH-GAU architecture.
68          - It composes the SelectiveGatingUnit, VectorQuantization, and
69          HierarchicalMemoryIntegrator units.
70
```

```python
        """

        def __init__(self, embed_dim: int, block_loc: tuple, kwarg_all: dict,
            device=None, dtype=None, **kwargs):
            self.factory_kwargs = {'device': device, 'dtype': dtype}
            super().__init__(embed_dim, block_loc, kwarg_all)
            self.selective_gating = SelectiveGatingUnit(embed_dim=
                self.embed_dim, block_loc=self.block_loc, kwarg_all=
                self.kwarg_all, **self.factory_kwargs, **self.kwarg_all)
            self.vector_quantization = VectorQuantization(embed_dim=
                self.embed_dim, block_loc=self.block_loc, kwarg_all=
                self.kwarg_all, **self.factory_kwargs, **self.kwarg_all)
            self.hierarchical_memory = HierarchicalMemoryIntegrator(embed_dim=
                self.embed_dim, block_loc=self.block_loc, kwarg_all=
                self.kwarg_all, **self.factory_kwargs, **self.kwarg_all)

        def _forward(self, X, **Z):
            X = X.to(**self.factory_kwargs)
            Y, Z_sg = self.selective_gating(X, **Z)
            Y, Z_vq = self.vector_quantization(Y, **Z_sg)
            Z_combined = {**Z_sg, **Z_vq}
            Y, Z_hmi = self.hierarchical_memory(Y, **Z_combined)
            Z_out = {**Z_combined, **Z_hmi}
            return Y, Z_out

    class SelectiveGatingUnit(GAUBase):
        """
        The SelectiveGatingUnit implements a selective gating mechanism to
        dynamically update the hidden state based on input relevance.

        **Overview:**

        - **Gating Function**:
          Computes gate values `s` using a sigmoid activation applied to a
          linear transformation of the input `X`.
          \\[
          s = \\sigma(W_s X + b_s)
          \\]

        - **Candidate State**:
          Computes the candidate state `	ilde{h}` using an activation
          function (e.g., `tanh`) applied to a linear transformation of `X`.
          \\[
            ilde{h} = 	ext{activation}(W_h X + b_h)
          \\]

        - **State Update**:
          Updates the hidden state `h` by combining `h_{	ext{prev}}` and
          `	ilde{h}` weighted by the gate `s`.
          \\[
          h = s \\odot h_{	ext{prev}} + (1 - s) \\odot 	ilde{h}
          \\]
          where `\\odot` denotes element-wise multiplication.

        **Args:**
            embed_dim (int): Embedding dimension `D`.
            block_loc (tuple): Location of the block within the network.
            kwarg_all (dict): Dictionary of all keyword arguments.
            device (torch.device, optional): Device to run the unit on.
            dtype (torch.dtype, optional): Data type of tensors.

        **Inputs:**
            - **X** (torch.Tensor): Input tensor of shape `(B, L, D)`.
            - **Z** (dict): Dictionary containing intermediate variables.
                - `h_prev` (torch.Tensor): Previous hidden state of shape
                  `(B, L, D)`. If not provided, initialized to zeros.

        **Outputs:**
            - **Y** (torch.Tensor): Updated hidden state of shape `(B, L, D)`.
            - **Z** (dict): Updated intermediate variables including `h_prev`.

        **Example:**

            # Initialize the unit
            unit = SelectiveGatingUnit(embed_dim=32, block_loc=(0, 0),
            kwarg_all={})
```

```
149              # Mock input
150              X = torch.randn(4, 10, 32)   # (B, L, D)
151              # Forward pass
152              Y, Z = unit(X)
153
154          **Note:**
155              - This unit is designed to be used within a larger GAU architecture.
156              - It implements the selective gating mechanism as described in the
157              VQH-GAU proposal.
158
159          """
160
161          def __init__(self, embed_dim: int, block_loc: tuple, kwarg_all: dict,
162              device=None, dtype=None, **kwargs):
163              self.factory_kwargs = {'device': device, 'dtype': dtype}
164              super().__init__(embed_dim, block_loc, kwarg_all)
165              self.W_s = nn.Linear(embed_dim, embed_dim, **self.factory_kwargs)
166              self.W_h = nn.Linear(embed_dim, embed_dim, **self.factory_kwargs)
167              self.activation = nn.Tanh()
168              self.sigmoid = nn.Sigmoid()
169
170          def _forward(self, X, **Z):
171              X = X.to(**self.factory_kwargs)
172              h_prev = Z.get('h_prev', None)
173              if h_prev is None:
174                  h_prev = torch.zeros_like(X)
175              s = self.sigmoid(self.W_s(X))
176              h_tilde = self.activation(self.W_h(X))
177              h = s * h_prev + (1 - s) * h_tilde
178              Z_out = {'h_prev': h}
179              return h, Z_out
180
181
182
183  class HierarchicalMemoryIntegrator(GAUBase):
184      """
185      HierarchicalMemoryIntegrator
186
187      This unit integrates quantized hidden states into hierarchical memory
188      structures across multiple scales.
189
190      **Main Features:**
191
192      - **Hierarchical Memory Integration**: Manages multiple hierarchical
193      scales of memory to capture long-range dependencies.
194
195      - **Learnable Parameters**: Uses learnable scaling factors
196      \\( alpha_s \\) and \\( beta_s \\) for each scale.
197
198      - **Output Projection**: Projects the concatenated memories back to the
199      original embedding dimension to maintain output shape consistency.
200
201      **Mathematical Formulation**:
202
203      For each scale \\( s \\in \\{1, 2, \\dots, S\\} \\):
204
205      \\[
206      H_t^{(s)} = alpha_s H_{t-s}^{(s)} + beta_s \\cdot
207      ext{quantized\\_h}_t
208      \\]
209
210      where:
211
212      - \\( H_t^{(s)} \\) is the hierarchical memory at time \\( t \\) and
213      scale \\( s \\).
214      - \\( H_{t-s}^{(s)} \\) is the memory from previous time step
215      \\( t-s \\) at scale \\( s \\).
216      - \\( alpha_s \\) and \\( beta_s \\) are learnable parameters for
217      scale \\( s \\).
218      - \\(  ext{quantized\\_h}_t \\) is the input quantized hidden state at
219      time \\( t \\).
220
221      **Output**:
222
223      The output \\( Y_t \\) is formed by projecting the concatenated
224      memories back to the embedding dimension:
225
226      \\[
```

```
227        Y_t =   ext{Projection}(    ext{concat}(H_t^{(1)}, H_t^{(2)}, \\dots,
228        H_t^{(S)}]))
229        \\]
230
231        **Args**:
232
233        - **embed_dim** (int): Embedding dimension \\( D \\).
234        - **block_loc** (tuple): Location of the block within the network.
235        - **kwarg_all** (dict): Dictionary of all keyword arguments.
236        - **device** (torch.device, optional): Device to run the unit on.
237        - **dtype** (torch.dtype, optional): Data type of tensors.
238        - **num_scales** (int, optional): Number of hierarchical scales
239        \\( S \\). Default: 3.
240        - **alpha_init** (float, optional): Initial value for \\( alpha_s \\).
241        Default: 0.9.
242        - **beta_init** (float, optional): Initial value for \\( beta_s \\).
243        Default: 0.1.
244
245        **Inputs**:
246
247        - **X** (torch.Tensor): Input tensor of shape \\( (B, L, D) \\),
248        representing the quantized hidden states.
249        - **Z** (dict): Dictionary containing intermediate variables, possibly
250        including previous hierarchical memories.
251
252        **Outputs**:
253
254        - **Y** (torch.Tensor): Output tensor of shape \\( (B, L, D) \\),
255        maintaining the same shape as the input.
256        - **Z** (dict): Updated dictionary containing the new hierarchical
257        memories for each scale.
258
259        **Example**:
260
261            # Initialize the unit
262            unit = HierarchicalMemoryIntegrator(embed_dim=32, block_loc=(0, 0),
263            kwarg_all={}, num_scales=3)
264            # Mock input
265            X = torch.randn(4, 10, 32)  # (B, L, D)
266            # Intermediate variables Z
267            Z = {}
268            # Forward pass
269            Y, Z = unit(X, **Z)
270
271        **Note**:
272
273        - This unit is designed to be used within the VQH-GAU architecture.
274        - It integrates quantized hidden states into a hierarchical memory
275        structure and maintains shape consistency with the input.
276        - The memory states are stored and updated in the intermediate variable
277        dictionary 'Z'.
278        """
279
280        def __init__(self, embed_dim: int, block_loc: tuple, kwarg_all: dict,
281            device=None, dtype=None, num_scales: int=3, alpha_init: float=0.9,
282            beta_init: float=0.1, **kwargs):
283            self.factory_kwargs = {'device': device, 'dtype': dtype}
284            super().__init__(embed_dim, block_loc, kwarg_all)
285            self.num_scales = num_scales
286            self.alpha = nn.Parameter(torch.full((num_scales, 1, 1, 1),
287                alpha_init, **self.factory_kwargs))
288            self.beta = nn.Parameter(torch.full((num_scales, 1, 1, 1),
289                beta_init, **self.factory_kwargs))
290            self.output_projection = nn.Linear(embed_dim * num_scales,
291                embed_dim, **self.factory_kwargs)
292
293        def _forward(self, X, **Z):
294            B, L, D = X.shape
295            device = X.device
296            dtype = X.dtype
297            H_list = []
298            Z_out = {}
299            for s in range(self.num_scales):
300                scale = s + 1
301                alpha_s = self.alpha[s]
302                beta_s = self.beta[s]
303                H_prev = Z.get(f'H_mem_{scale}', None)
304                if H_prev is None:
```

```python
                    H_prev = torch.zeros(B, L, D, **self.factory_kwargs)
                if scale < L:
                    pad = torch.zeros(B, scale, D, **self.factory_kwargs)
                    H_prev_shifted = torch.cat([pad, H_prev[:, :-scale, :]],
                    dim=1)
                else:
                    H_prev_shifted = torch.zeros(B, L, D, **self.factory_kwargs)
                H_curr = alpha_s * H_prev_shifted + beta_s * X
                Z_out[f'H_mem_{scale}'] = H_curr
                H_list.append(H_curr)
            H_concat = torch.cat(H_list, dim=-1)
            Y = self.output_projection(H_concat)
            return Y, Z_out

class VectorQuantization(GAUBase):
    """
    VectorQuantization

    This unit performs vector quantization on the hidden states to compress
    the representations.
    It maps continuous hidden states to discrete codebook entries, reducing
    memory usage while maintaining essential information.

    **Main Features:**

    - **Codebook**: A learnable embedding table containing `K` code vectors
    of dimension `D`.

    - **Quantization Process**:
        For each input vector `h_t` in the hidden state `H`:
        \\[
        i^* = arg\\min_{i} \\| h_t - c_i \\|^2 \\
        q_t = c_{i^*}
        \\]
        where \\( c_i \\) are the codebook vectors.

    - **Loss Function**:
        The vector quantization introduces a loss term to train the
        codebook:
        \\[
        \\mathcal{L}_{VQ} = \\| h_t -   ext{sg}[q_t] \\|^2 + beta \\| q_t
        -  ext{sg}[h_t] \\|^2
        \\]
        where \\(   ext{sg}[\\cdot] \\) denotes the stop-gradient operation,
        and \\( beta \\) is a hyperparameter controlling the commitment
        loss.

    **Args**:

        - **embed_dim** (int): Embedding dimension \\( D \\).
        - **block_loc** (tuple): Location of the block within the network.
        - **kwarg_all** (dict): Dictionary of all keyword arguments.
        - **device** (torch.device, optional): Device to run the unit on.
        - **dtype** (torch.dtype, optional): Data type of tensors.
        - **num_embeddings** (int, optional): Number of embeddings in the
        codebook \\( K \\). Default: 512.
        - **beta** (float, optional): Commitment loss weighting factor
        \\( beta \\). Default: 0.25.

    **Inputs**:

        - **X** (torch.Tensor): Input tensor of shape \\( (B, L, D) \\),
        representing the hidden states.
        - **Z** (dict): Dictionary containing intermediate variables.

    **Outputs**:

        - **Y** (torch.Tensor): Quantized tensor of shape \\( (B, L, D) \\).
        - **Z** (dict): Updated dictionary containing any necessary
        intermediate variables.
            - **vq_loss** (torch.Tensor): Vector quantization loss
            (optional, for training).

    **Example**:

        # Initialize the unit
```

```python
            unit = VectorQuantization(embed_dim=64, block_loc=(0, 0),
            kwarg_all={}, num_embeddings=512, beta=0.25)
            # Mock input
            X = torch.randn(8, 16, 64)  # (B, L, D)
            # Forward pass
            Y, Z = unit(X)

        **Note**:

            - This unit is designed to be used within the VQH-GAU architecture.
            - It performs vector quantization as described in the VQH-GAU
            proposal.
            - The quantization loss can be used during training to update the
            codebook vectors.

        """

    def __init__(self, embed_dim: int, block_loc: tuple, kwarg_all: dict,
        device=None, dtype=None, num_embeddings: int=512, beta: float=0.25,
        **kwargs):
        self.factory_kwargs = {'device': device, 'dtype': dtype}
        super().__init__(embed_dim, block_loc, kwarg_all)
        self.num_embeddings = num_embeddings
        self.beta = beta
        self.embedding = nn.Embedding(self.num_embeddings, self.embed_dim,
            **self.factory_kwargs)
        self.embedding.weight.data.uniform_(-1 / self.num_embeddings, 1 /
            self.num_embeddings)

    def _forward(self, X, **Z):
        B, L, D = X.shape
        assert D == self.embed_dim, f'Input embedding dimension {D} does not
        match expected dimension {self.embed_dim}'
        h_flat = X.view(-1, D)
        codebook = self.embedding.weight
        distances = torch.cdist(h_flat.unsqueeze(0), codebook.unsqueeze(0))
        distances = distances[0]
        assignment_weights = F.softmax(-distances / 0.1, dim=1)
        quantized = torch.mm(assignment_weights, codebook)
        encoder_loss = F.mse_loss(h_flat, quantized)
        loss = encoder_loss
        quantized = quantized.view(B, L, D)
        Y = quantized
        Z_out = {}
        Z_out['vq_loss'] = loss
        return Y, Z_out

gab_config = {'beta': 0.25, 'temperature': 1.0, 'num_embeddings': 512,
'alpha_init': 0.9, 'num_scales': 3, 'beta_init': 0.1}
```

