# OpenReview forum: "Language Modeling by Language Models"
_NeurIPS.cc/2025/Conference — NeurIPS 2025 spotlight_

### Official Review · Reviewer_tRdq · 2025-06-19

**Clarity:** 1
**Significance:** 2
**Originality:** 3
**Rating:** 4
**Confidence:** 3

**Summary:**

The paper presents Genesys, a multi-agent, LLM-driven framework, for discovering language model architectures via evolutionary search and unit-based agent design. The core idea is to use LLMs themselves to design, propose, verify, and improve new LM architectures beyond standard transformers. The system discovers 1,062 novel architectures across different parameter scales, and find the best designs to be  competitive with known architectures like GPT2 and Mamba2.

**Questions:**

1. What tasks are you choose for evaluation and why?
2. Will you also do some search on training hyper-parameters?

**Ethical Concerns:**

["NO or VERY MINOR ethics concerns only"]

**Final Justification:**

My main concerns about the paper still focusing on the overall contributions and writing. The concerns about evaluation and training details are addressed by the authors' response. Therefore, I have raised the score accordingly.

**Limitations:**

yes

**Quality:**

2

**Strengths And Weaknesses:**

Strengths:
1. Clear motivation and important topic.
2. Extensive experiments and full ablation study for validating effects of different modules.

Weaknesses:
1. The whole paper is more like a system design report. While the whole system is a bit complicated, its presentation lacks clear organization, making it difficult to follow up.
2. The result transparency is limited. Not much details on how they compute the fitness scores (like what tasks? How tested? Model training and evaluation configurations?).
3. The scale is still limited, with max scale 350M.

---

> ### Author Rebuttal · Authors · 2025-07-30
>
> We thank Reviewer tRdq for their valuable feedback. We agree that the paper's organization can be improved to better highlight our scientific contributions, and we appreciate the chance to provide a clear summary of the extensive experimental details contained in the paper and its appendices.
>
> ---
>
> ## Weaknesses
>
> > **W1:** The whole paper is more like a system design report. While the whole system is a bit complicated, its presentation lacks clear organization, making it difficult to follow up.
>
>
> We thank the reviewer for their feedback on the paper's structure and contribution framing. We take the clear writing of our paper seriously and will improve the organization in the camera-ready version to make the scientific contributions more prominent.
>
> **Our Scientific Contributions**
>
> We acknowledge that the paper includes significant detail about the Genesys system. This is because the system itself serves as a novel scientific instrument for a complex task. However, our primary goal is to use this system to generate new insights. We believe our contributions go well beyond a system report and include:
>
> * **Methodological Innovations**: We introduce key research contributions to make ASD in this domain tractable including: (1) a novel, unit-based **Viterbi-style search** for code generation, which we prove formally to be exponentially more efficient than standard methods both empirically in Table 2 and formally in Appendix A; (2) a **"Ladder of Scales"** strategy for managing the prohibitive cost of evaluation; (3) and our system design of Genesys/LMADE which emulating human research workflow.
> * **Empirical Scientific Insights**: We use the system to conduct experiments that yield new knowledge. Our **ablation studies** in Table 1 empirically quantify the value of knowledge and experimental feedback in the discovery process. Furthermore, our analysis of the **evolutionary tree** reveals emergent discovery patterns in Appendix E.1.2, and we pioneer an analysis directly **linking architectural units to downstream performance** in Appendix E.3.3.
> * **Tangible Discoveries**: The system is not just a theoretical construct; it produced **1,062 verified novel architectures**. The top designs are competitive with strong human baselines, demonstrating the method's effectiveness.
>
> Beside, we will release all code, discovery artifacts, and experiment runs (via wandb), including our console for running new experiments, upon publication.
>
> **Improving Paper Organization and Readability**
>
> To improve clarity, we will make the following changes in the camera-ready version:
>
> * **Itemized Contributions**: We will add a dedicated, itemized list of our main contributions in the **Introduction**, providing a clear, upfront summary of our methodological and scientific takeaways.
> * **Improved Signposting**: We will enhance the "road map" at the beginning of major sections. For instance, before Sec. 4, we will add a sentence explaining that the section describes the instrument used to generate the scientific insights presented in Sec. 5.
>
> We are confident these improvements will better highlight our scientific contributions and make the paper easier to follow.
>
>
>
> > **W2:** The result transparency is limited. Not much details on how they compute the fitness scores (like what tasks? How tested? Model training and evaluation configurations?).
>
> We thank the reviewer for this question. The paper provides extensive details on fitness calculation, tasks, and configurations, primarily in Sections 3.1, 5.3, and Appendices B and C. Here is a summary of the key information.
>
> **1. Fitness Score Computation and Tasks**
>
> The method for calculating fitness is formally defined and consistently applied throughout the experiments.
>
> * **Fitness Formula**: The fitness score ($\mathcal{F}$) is the average empirical performance across **M** downstream tasks and **K** model scales (Sec. 3.1).
> * **Evaluation Tasks**: Models are evaluated on **29 selected LM-Eval benchmarks** (listed in Table 16). The nine most informative tasks were chosen for final comparison (Table 3) based on a statistical analysis (Appendix E.3.1) considering result variance and example count.
> * **Evaluation Metric**: The primary metric for comparison is **zero-shot accuracy (%)** (Table 3).
>
> **2. Model Training and Evaluation Configurations**
>
> We provide full details in Appendix C for reproducibility.
>
> * **Training Corpus**: All models were pretrained on a custom 34.78B token **SmolLM-1/8-Corpus**, which is a high-quality subset of the SmolLM corpus filtered for educational content to improve training efficiency (details in Appendix C.1.1, Table 8).
> * **Training Settings**: We detail the hardware environment in Appendix C.1.2, model experiment settings in C.1.3, and verification engine hyperparameters in Table 11.
> * **Evaluation Framework**: We use a customized **LM-Eval framework**. An **Auto-Tuner** ensures models fit the target parameter scale and tunes gradient accumulation to prevent OOM errors. A **Runtime Checker** terminates failed runs early (Details in Appendix B.1.4).
>
> In addition, we provide full details for the evolutionary system and agent setups in Appendix C.2 and Appendix B.1.5. Moreover, we will release all code, discovery artifacts, and experiment runs (via wandb), including our console for running new experiments, upon publication.
>
>
>
> > **W3:** The scale is still limited, with max scale 350M.
>
> We agree that 350M parameters is not a large language model by today's standards. Our choice of this scale was deliberate and is well-suited for the primary goals of this study.
>
> **Justification for the 350M Scale**
>
> * **Focus on Validating the Discovery Method**: The main contribution of this paper is not a single, large-scale model, but rather the validation of **Genesys**, a novel ASD system for discovering architectures. The 14M-350M parameter range is large enough to reveal meaningful differences between architectures and demonstrate stable performance trends, while remaining computationally tractable for an extensive search involving over 1,000 unique, trained designs.
>
> * **Relevance to Architectural Research**: This scale is typical and useful for comparative architectural studies. Many recent and impactful models included the scale of 125M and 350M as a part of their standard evaluation, such as **Mamba2** and **RWKV7** compared in our study.
>
> * **Demonstrating Scaling Trends**: Our "Ladder of Scales" strategy is predicated on the idea that performance trends at smaller scales correlate with those at larger scales. Our results support this, with discovered models showing consistent loss reduction as they scale up (as seen in Figure 43), suggesting the architectural merits we observe are not confined to a single scale.
>
> In summary, the 350M scale provides a meaningful and computationally feasible setting to validate our discovery methodology and demonstrate its ability to produce architectures that are competitive with the state-of-the-art in this domain.
>
>
> ## Questions
>
> > **Q1:** What tasks are you choose for evaluation and why?
>
> As discussed above, we evaluates models on **29 selected LM-Eval benchmarks**, those tasks are selected by performing a basic clearing from the LM-Eval task list available in https://github.com/EleutherAI/lm-evaluation-harness/blob/main/lm_eval/tasks/README.md: removing overly difficult tasks (such as MT Bench) for the considered model scales, or the ones that are too time consuming (such as requiring long-text generation), or the ones that cannot be directly applied with autoregressive LMs (such as the ones designed for BERT), or erroneous ones.
> These tasks are listed in Table 16. For the tasks in Table 3, we selected the nine most informative benchmarks based on a statistical analysis detailed in Appendix E.3.1, considering factors like a high standard deviation of results and a sufficient number of examples.
>
> > **Q2:** Will you also do some search on training hyper-parameters?
>
> No, the current system does not perform an automated search for optimal training hyperparameters (e.g., learning rate, optimizer settings) for both evolution and model evaluations in Sec. 5.3 Table 3. Instead, we adopted a standardized protocol to ensure fair and reproducible comparisons, as described in Sec. 3.2.
>
> **Our Hyperparameter Protocol**
>
> * **Standardized Training for Discovered Models**: To ensure consistency, all discovered architectures were trained using a fixed set of standard pre-training hyperparameters. These settings, including the AdamW optimizer and a cosine learning rate scheduler, are detailed in Appendix B.1.4 and Table 11.
>
> * **Role of the Auto-Tuner**: We do use an **Auto-Tuner**, but its role is not to optimize training hyperparameters for performance. As described in Appendix B.1.4, its purpose is to:
>     1.  Adjust architectural parameters (number of layers and embedding dimension) to ensure a model correctly fits the target parameter count (e.g., 350M).
>     2.  Tune the gradient accumulation steps to prevent out-of-memory errors, ensuring the model can train on the available hardware.
>
> * **Official Settings for Baselines**: For the final model evaluation in Sec. 5.3, we trained the human-designed baselines (like Mamba2 and RWKV7) using the **official hyperparameters published in their original papers or code repositories**. This process, detailed in Appendix C.1.3, was chosen to ensure that the baseline models were evaluated under their own optimal, specialized conditions.
>
> ---
>
> We are confident that the planned organizational changes and the clarifications provided here will resolve the concerns about clarity and transparency. Please do not hesitate to contact us if any aspect of our response requires further explanation.

---

> > ### Author Response · Authors · 2025-08-04
> >
> > Dear Reviewer tRdq,
> >
> > Thank you again for your time and for your review.
> >
> > With the discussion period ending soon, we just wanted to gently follow up. We hope our rebuttal helped address your concerns about clarity and transparency by pointing to the extensive experimental and fitness computation details located in the appendices.
> >
> > We would be happy to provide any further clarification you might need.
> >
> > Best regards,
> > The Authors

---

> > ### Comment · Reviewer_tRdq · 2025-08-06
> >
> > Thanks for the response. It addresses some of my concerns. I will raise my score accordingly.

---

> > > ### Author Response · Authors · 2025-08-06
> > >
> > > Dear Reviewer tRdq,
> > >
> > > Thank you for your response and for taking the time to consider our rebuttal. We sincerely appreciate you raising your score.
> > >
> > > We are glad that our clarifications addressed some of your concerns. As the discussion period is still open, we would be grateful for the opportunity to address any remaining points. If you have a moment, could you please let us know which aspects of the paper still lack clarity or transparency? Your feedback is invaluable as we work to improve the final version.
> > >
> > > Thank you again for your constructive engagement with our work.
> > >
> > > Best regards,
> > >
> > > The Authors

---

### Official Review · Reviewer_iHdi · 2025-06-29

**Clarity:** 3
**Significance:** 3
**Originality:** 4
**Rating:** 5
**Confidence:** 4

**Summary:**

This paper presents Genesys, a large-scale, LLM-driven system for automated neural architecture discovery. The system combines a modular architecture representation (the GAU tree), a multi-agent proposal-implementation-verification pipeline, and a distributed evolutionary framework to generate, evaluate, and refine novel model designs. The authors conduct extensive experiments and ablation studies to demonstrate the effectiveness, stability, and practicality of their system.

**Questions:**

- Is the current framework capable of supporting more flexible units designs? For example, can it accommodate using different block types at different layers within a Transformer, or design parallel block layers that merge information after repeating several blocks?

- Do you think Genesys is capable of producing effective results at larger parameter scales? A brief explanation of your understanding would be appreciated.

**Ethical Concerns:**

["NO or VERY MINOR ethics concerns only"]

**Final Justification:**

After reviewing the authors’ response and other feedback, I have raised my initial score to accept the paper. The paper explores an interesting topic by using multi-agent systems to automatically design language models, thereby broadening traditional NAS and opening a new research direction. Both its motivation and technical design are well-suited for NeurIPS.

**Limitations:**

yes

**Quality:**

3

**Strengths And Weaknesses:**

## Strengths:

- The introduction of GAU trees to factorize neural architecture code into interpretable, discrete units provides a meaningful structure for evolutionary optimization and design comparison.

- The paper is clearly structured with well-defined terminology, making it easy for readers to follow the overall system workflow and module design.

- The work achieves a comprehensive and complex system design through a substantial engineering effort.

- The experiments and evaluation metrics are well-designed and appropriate for demonstrating the system's effectiveness.

## Weakness:

- How does Genesys ensure diversity during the design process? What methods are used or could be used to evaluate the diversity and innovation of the generated designs? Additionally, while the paper demonstrates the ability to design new units from scratch, I am curious whether Genesys can leverage the knowledge engine (KE) to incorporate mathematical or physical principles to create entirely novel module components.

- Ablation experiments are an important part of the scientific process. Does Genesys support such ablation experimentation? Specifically, can it perform joint analyses of multiple designs’ structures and performances to better understand the contribution of certain modules?

---

> ### Author Rebuttal · Authors · 2025-07-30
>
> We thank Reviewer iHdi for their positive, thorough, and insightful review. We are grateful for the excellent questions regarding diversity, ablation support, and architectural flexibility, as they touch upon the core strengths and advanced capabilities of the Genesys framework.
>
> ---
>
> ## Weaknesses
>
> > **W1:** How does Genesys ensure diversity during the design process? What methods are used or could be used to evaluate the diversity and innovation of the generated designs? Additionally, while the paper demonstrates the ability to design new units from scratch, I am curious whether Genesys can leverage the knowledge engine (KE) to incorporate mathematical or physical principles to create entirely novel module components.*
>
> > *“How does Genesys ensure diversity during the design process?”*
>
> Genesys employs several mechanisms to encourage diversity and prevent premature convergence:
>
> * **Novelty Checks**: The system actively checks for novelty to avoid self-replication.
>     * **Proposal stage**: the Reviewer agent compares the new proposal against previous proposals from the same parents ("siblings") and similar past proposals (**Appendix B.1.5**).
>     * **Implementation stage**: the Observer agent assesses the novelty of the generated code against prior and sibling implementations (**Sec. 4.2**).
> * **Exploration-Exploitation Strategy**: The design selection process explicitly balances exploiting promising designs with exploring diverse or less successful ones (**Sec. 4.3**). The quadrant-based selection strategy dedicates a portion of the search to "Poor & Confident" and "Poor & Unconfident" designs to ensure the search does not get stuck in local optima.
> * **Probabilistic Selection and Restarts**: Probabilistic selection and a random restart mechanism also foster new evolutionary paths (**Appendix C.2**).
>
> > *“What methods are used or could be used to evaluate the diversity and innovation of the generated designs?”*
>
> We evaluate diversity both quantitatively and qualitatively:
>
> * **Quantitative Analysis**: We analyze the **Evolutionary Tree** structure (**Appendix E.1.2**) to understand discovery patterns. As seen in the comparison between our "Full" (**Fig. 8, Right**) and "w/o Exp." (**Fig. 35**) systems, fitness-guided selection creates hubs (exploitation). We also analyze the reuse and adaptation of GAU components (**Appendix E.3.2**) to show how existing ideas are built upon to create new designs.
> * **Qualitative Agent-Based Evaluation**: LLM agents directly assess innovation (**Sec. 4.2**). The **Reviewer agent** scores each proposal's novelty, and the **Observer agent** rates the code's quality and innovation, providing a peer-review-like assessment for each design.
>
> > *“...whether Genesys can leverage the knowledge engine (KE) to incorporate mathematical or physical principles to create entirely novel module components.”*
>
> Yes, the system is designed with the flexibility to incorporate deeper scientific principles. The **Knowledge Engine (KE)** already provides academic literature to the Proposer agent, informing its design choices (**Sections 3.2, 4.2**).
>
> This capability can be extended straightforwardly:
>
> * **Expanding the Knowledge Base**: Augmenting the KE library with papers on mathematical principles (e.g., control theory) or physics, thus hinting the Proposer to draw inspiration from these domains.
> * **Integrating External Tools**: Equipping agents with tools like a symbolic math engine (e.g., Mathematica) to allow them to experiment with mathematical principles while drafting proposals.
>
> > **W2:** Ablation experiments are an important part of the scientific process. Does Genesys support such ablation experimentation? Specifically, can it perform joint analyses of multiple designs’ structures and performances to better understand the contribution of certain modules?
>
> We thank the reviewer for this excellent question. The ability to perform ablations and analyze component contributions is a key capability of our system.
>
> Yes, Genesys is explicitly designed to support such experimentation, and we have already conducted a preliminary analysis demonstrating its ability to understand the contributions of specific modules.
>
> **1. Support for Ablation via GAU Factorization**
>
> The core design of Genesys makes architectural ablation straightforward. Our system is built on the principle of factorizing every architecture into a tree of **Generalized Autoregressive Units (GAUs)**.
>
> * **Ablation as Mutation**: An ablation is equivalent to a targeted **mutation**. To ablate a module (e.g., `VectorQuantization` from VQH), the Proposer agent replaces that GAU with a simpler one, like an identity function.
> * **Unit-Based Control**: Because the entire search process operates on these swappable, high-level units, the system provides the necessary control to systematically modify, remove, or replace any component of a discovered architecture for targeted analysis.
>
> **2. Joint Analysis of Module Contributions**
>
> Analyzing multiple designs to understand module contributions is a key scientific goal of our work. We present a direct proof-of-concept for this capability in **Appendix E.3.3 ("Unit-Performance Relation")**.
>
> * **"Bag-of-Units" Representation**: We represented each of our 1,062 designs as a **"Bag-of-Units"** (BoU), where the "words" are the names of the GAUs used in its architecture.
> * **Predicting Performance from Structure**: We then trained a Naive Bayes classifier to predict a design's performance on various downstream tasks based on its BoU representation.
> * **Key Finding**:We found a clear predictive link between GAU components and task performance, achieving an overall fitness level with an F1 score of 0.658 on held-out data (Table 17) and showed predictive power on 19 different downstream tasks (Table 18).
>
> This result demonstrates that Genesys can perform a joint analysis across its entire discovery history to learn which architectural modules are correlated with success. As we conclude in the paper, this capability opens the door for more sophisticated, insight-driven discovery, where the system can learn design principles that guide future exploration.
>
> ## Questions
>
> > **Q1:** Is the current framework capable of supporting more flexible units designs? For example, can it accommodate using different block types at different layers within a Transformer, or design parallel block layers that merge information after repeating several blocks?
>
> Yes, the framework's design is highly flexible and supports both sophisticated patterns. This is a core capability, enabled by the **`block_loc` parameter** and the intermediate variable dictionary **`Z`**.
>
> **1. Different Block Types Across Layers**
>
> The `block_loc` parameter `(layer_idx, n_block)` informs each GAB/GAU of its position in the network stack, allowing for layer-specific logic (**Appendix F.4.1**). A Proposer agent can use conditional logic based on `layer_idx` to instantiate different child units at different layers (e.g., varying architectures or operations between blocks).
>
>   * **Example Usage: GPT-Mamba hybrid model**:
>     ```python
>     # Pseudo-code for a block's __init__ method
>     if self.block_loc[0] % 2 == 0:
>         # Even layers are GPT-style
>         self.layer = GPT_Attention_Unit(...)
>     else:
>         # Odd layers are Mamba-style
>         self.layer = Mamba_SSM_Unit(...)
>     ```
>
> **2. Parallel Blocks with Merging**
>
> Parallel blocks with merging are supported via the flexible `Z` dictionary. As defined in our base classes (**Fig. 11, Appendix F.1.1**), `Z` is a key-value carrier for information, like tensors from parallel branches, passed between blocks. One block can add outputs to `Z`, and a subsequent block can retrieve and merge them, using `layer_idx` to trigger merging periodically.
>
>   * **Example Usage: Multi-branch model**:
>     ```python
>     # Pseudo-code for a block's _forward method
>
>     # --- In blocks where merging happens (e.g., every 4th layer) ---
>     if self.block_loc[0] % 4 == 0 and self.block_loc[0] > 0:
>         # Retrieve outputs from parallel branches stored in Z
>         branch1_out = Z.get('branch1_output', 0)
>         branch2_out = Z.get('branch2_output', 0)
>
>         # Merge the main path with the side channels
>         X = X + branch1_out + branch2_out
>
>     # --- In all blocks, run parallel branches and store outputs in Z ---
>     z_out['branch1_output'] = self.branch1_unit(X, **Z)
>     z_out['branch2_output'] = self.branch2_unit(X, **Z)
>
>     return X, z_out
>     ```
>
> These core constructs provide a powerful and extensible framework for exploring a vast and complex architectural design space.
>
> > **Q2:** Do you think Genesys is capable of producing effective results at larger parameter scales? A brief explanation of your understanding would be appreciated.
>
> Yes, we are confident Genesys can be effective at larger scales, based on its design and experimental results.
>
> * **Stable Evolutionary Progress**: The "Full" system shows a stable, positive improvement trajectory (**Sec. 5.1, Table 1, Fig. 8**). A process that consistently improves upon strong baselines is likely to continue this progress at larger scales.
> * **Efficient Multi-Scale Methodology**: The **"Ladder of Scales"** strategy (**Sec. 4.3**) is designed for efficient discovery, using smaller scales to filter for promising architectures before committing to expensive, large-scale training. This design is fundamental to making a large-scale search tractable and effective.
> * **Architectures Follow Scaling Laws**: As shown in our analysis (**Appendix E.3.1, Fig. 43**), discovered architectures follow expected scaling laws, with training loss decreasing as model size increases. This suggests the architectural merits we found will persist at larger scales.
>
> ---
>
> Thank you again for the engaging and thoughtful review. Please do not hesitate to reach out if you have any follow-up questions; we would be happy to discuss these aspects of our system further.

---

> > ### Author Response · Authors · 2025-08-04
> >
> > Dear Reviewer iHdi,
> >
> > Thank you again for your positive and insightful review.
> >
> > With the discussion period ending soon, we just wanted to gently follow up. We hope our rebuttal fully addressed your excellent questions regarding the system's diversity mechanisms, support for ablation experiments, and architectural flexibility.
> >
> > We would be happy to answer any further questions you may have.
> >
> > Best regards,
> > The Authors

---

> > ### Comment · Reviewer_iHdi · 2025-08-05
> > **Response**
> >
> > Thank you for the detailed response. All my concerns have been addressed, and I appreciate the paper's innovative use of multi-agent systems for language modeling. I will accordingly raise my initial score.

---

> > > ### Author Response · Authors · 2025-08-06
> > >
> > > Dear Reviewer iHdi,
> > >
> > > Thank you for your detailed and positive feedback. We are delighted to hear that our response fully addressed your concerns. We truly appreciate your insightful questions and your recognition of the paper's contributions to the field.
> > >
> > > Thank you again for your support.
> > >
> > > Best regards,
> > >
> > > The Authors

---

### Official Review · Reviewer_cxPy · 2025-07-01

**Clarity:** 4
**Significance:** 4
**Originality:** 4
**Rating:** 6
**Confidence:** 4

**Summary:**

This paper proposes an automated method for discovering novel language model architectures based on multi-agent large language models, simulating the full research process of design, implementation, pretraining, and evaluation as in real scientific work. The Genesys system employs a "ladder of scales" strategy, progressively expanding model sizes (from 14M to 350M parameters) while tightening training budgets, to propose, adversarially review, and implement architecture designs. To improve design generation efficiency, Genesys introduces a genetic programming-based backbone structure, achieving about an 86% higher success rate compared to traditional direct prompting methods. The experiments discovered a total of 1,162 new architecture designs, with 1,062 fully validated through the entire process; the best designs outperformed existing architectures like GPT-2 and Mamba-2 on 6 out of 9 common benchmarks. Additionally, through systematic ablation studies and formal analysis, the paper deeply explores the impact of design components on automated discovery efficiency, providing important insights for efficient automated architecture discovery.

**Questions:**

1. The current Genesys system allows three types of modification operations: mutation, crossover, and design from scratch. Could you please explain the proportion and effectiveness of these three different modification operations in the final proposals generated by the system?
2. The system presets five initial seed architectures as initialization. Is it possible to remove this initialization so that the system can perform literature review and design entirely from scratch, thereby generating novel architectures different from known ones?
3. Genesys achieves overall language model architecture design and review through interactions among different agents. In this process, have you observed any phenomena like reward hacking by agents? If so, what solutions have been considered or implemented?

**Ethical Concerns:**

["NO or VERY MINOR ethics concerns only"]

**Final Justification:**

After carefully considering the authors’ response, I have decided to raise my score. All of the concerns I previously raised have been fully addressed.
I believe the motivation, foundation, and the amount of work presented in this paper are sufficiently novel and substantial. Therefore, I consider the paper worthy of acceptance to NeurIPS.

**Limitations:**

Yes

**Paper Formatting Concerns:**

No major formatting issues found.

**Quality:**

4

**Strengths And Weaknesses:**

Strengths：

1. The paper addresses an important problem with strong motivation: using large language models to model the discovery of novel language model architectures.
2. It innovatively introduces a genetic programming backbone structure, combining multiple components such as a knowledge engine, to implement a ladder-of-scales approach for discovering new language model architectures.
3. Comprehensive ablation studies are conducted, demonstrating the overall system’s effectiveness and the rationality of each component’s design.

Weaknesses：

1. The system requires high computational resources, limiting its reproducibility locally and restricting exploration of larger-scale model architectures.

---

> ### Author Rebuttal · Authors · 2025-07-30
>
> We thank Reviewer cxPy for their exceptionally positive and encouraging review. We are delighted that they found our work to be excellent in its motivation, originality, and execution. We appreciate the opportunity to address the valid point on computational resources and answer the insightful questions raised.
>
> ---
>
> ## Weaknesses
>
> > **W1:** The system requires high computational resources, limiting its reproducibility locally and restricting exploration of larger-scale model architectures.
>
> We agree that discovering language model architectures is a computationally intensive task. This high cost is inherent to the problem, as any meaningful evaluation requires pre-training models, often across multiple scales. Our work directly confronts this challenge with a novel strategy designed to dramatically improve efficiency.
>
> Our **"Ladder of Scales"** strategy, described in **Sec. 4.3**, is our primary method for making this high-cost search tractable. Instead of wastefully training all candidate designs at the largest scale, this method uses smaller, cheaper scales to progressively filter for the most promising architectures.
>
> As shown in **Figure 9**, to find the best designs from an initial pool of 1,000 candidates, our system uses a pyramidal approach which starts by training all **1,000** candidates at a small **14M** parameter scale, and the selected **400** to the **31M** scale, then **150** trials at **70M** and **40** trials at **125M**, Finally, only the top **5** candidates are trained at the target **350M** scale.
>
> This selective, multi-scale evaluation strategy significantly reduces the computational burden compared to a naive brute-force search. Based on our training time estimations for different hardware scenarios in Appendix E.4.1 and Table 19, it reduced the H100 GPU hours required for training all 1000 candidates in the target 350M scale directly from around 38,000 hours to 1100 hours.
>
>
> ## Questions
>
> > **Q1:** The current Genesys system allows three types of modification operations: mutation, crossover, and design from scratch. Could you please explain the proportion and effectiveness of these three different modification operations in the final proposals generated by the system?
>
> Yes, the paper provides these details in the main text and appendices.
>
> **Proportion of Modification Operations**
>
> The proportions of the three modification operations were set as a probabilistic choice for the Designer agents. As detailed in **Sec. 4.3**, the probabilities used in our experiments were **75% for mutation, 20% for crossover, and 5% for design-from-scratch**. This distribution was chosen to prioritize the refinement and improvement of existing successful designs (exploitation) while still allowing for the combination of ideas from different lineages and the introduction of radical novelty (exploration).
>
>
> **Effectiveness of Modification Operations**
>
> The paper does not provide a direct quantitative comparison of which operation leads to the highest fitness gain, as a successful design in a later generation is often the result of a long lineage of different operations, making attribution difficult.
>
> However, we can infer the effectiveness from their roles and the resulting evolutionary patterns analyzed in **Appendix E.1.2** :
> * **Mutation (75%)**: This is the primary driver for local, iterative improvement. The high probability reflects a strategy focused on refining successful ideas. This aligns with the "hubness" pattern we observed in the "Full" system's evolutionary tree (**Figure 8 Right**), where successful designs become parents to many incremental refinements.
> * **Crossover (20%)**: This operation is effective at combining successful "genes" from different evolutionary branches. An example that led to a successful final model is shown in **Figure 16**.
> * **Design from Scratch (5%)**: This is the main source of radical novelty, but it's used sparingly due to its higher risk of producing less effective designs. However, one of our top designs VQH (also presented in Appendix G.2) is generated from this mode, despite it can still include other previously discovered designs as references (e.g., one reference is a mutated version of another top design HippoVQ).
>
> In summary, the high proportion of **mutation** is effective for the steady, exploitative progress that dominates the search, while **crossover** and **design-from-scratch** are crucial for exploration and introducing valuable diversity into the gene pool.
>
>
>
> > **Q2:** The system presets five initial seed architectures as initialization. Is it possible to remove this initialization so that the system can perform literature review and design entirely from scratch, thereby generating novel architectures different from known ones?
>
> Yes, the Genesys framework is fully capable of designing architectures without being initialized with seed models. This capability is a core part of its design. "Design from scratch" is one of the three fundamental genetic programming operations available to the Designer agents, alongside mutation and crossover. As detailed in the system prompts in **Appendix F**, when this operation is selected, the Proposer agent is **not given a parent architecture to modify**. Instead, its sole task is to perform a literature review using the Knowledge Engine and then propose a completely novel GAU tree from first principles, based on the insights it gathers. As mentioned above, one example is VQH, one of our top designs.
>
> While the system can design from a blank slate, we intentionally initialized the primary experiment with five state-of-the-art seeds for a specific methodological reason: our goal was to model the **real-world scientific process**. Researchers rarely start in a vacuum; they build upon, challenge, and improve the current state-of-the-art.
>
> By providing seeds, we tasked Genesys with the more realistic and arguably more difficult challenge of advancing beyond today's highly optimized architectures, rather than spending significant computational resources rediscovering foundational concepts (like basic attention or normalization layers) that are already well-established.
>
>
>
> > **Q3:** Genesys achieves overall language model architecture design and review through interactions among different agents. In this process, have you observed any phenomena like reward hacking by agents? If so, what solutions have been considered or implemented?
>
> That's a great question about a crucial aspect of multi-agent systems. Based on the system's design, reward hacking is unlikely due to the nature of the feedback signals.
>
> The Genesys system has two primary feedback mechanisms, neither of which is easily exploitable in a "reward hacking" sense.
>
> **1. Final Fitness Score**
>
> The ultimate "reward" for any discovered architecture is its **empirical performance on held-out downstream benchmarks**. This final fitness score is calculated by the **Verification Engine** based on real-world training and evaluation. Because this score is tied to objective, external tasks, it is not possible for an agent to "hack" it in the traditional sense. The architecture either performs well or it doesn't.
>
> **2. Intermediate Design-Phase Signals**
>
> During the design process, there are two intermediate signals that guide the agents:
>
> * **Symbolic Checker (SC)**: This is a non-learnable, rule-based system. As detailed in **Appendix B.1.2**, it uses a series of static and runtime checks (e.g., AST parsing, checking for differentiability, causality, and numerical stability) to validate a design. Since these are hard-coded criteria for a valid program, an agent cannot "hack" the checker; it must produce code that passes these objective tests.
> * **Reviewer Agent Score**: The Proposer agent's goal is to get a high score from the adversarial Reviewer agent. While this is an agent-agent interaction, we did not observe reward hacking. The Reviewer is tasked with critically assessing **novelty, feasibility, and theoretical soundness** by comparing the proposal against the knowledge base, sibling designs, and past proposals. The diversity of foundation models used for the agents also helps prevent stable, exploitable behaviors from emerging.
>
> In summary, the final reward is objective, and the intermediate signals are either based on hard-coded rules or a critical, adversarial review process, making the system robust against reward hacking.
>
>
> ---
>
> We are grateful for such a supportive and thorough assessment of our work. Please do not hesitate to let us know if any of our answers can be clarified further. Thank you once again.

---

> > ### Comment · Reviewer_cxPy · 2025-08-04
> >
> > I sincerely appreciate the author's detailed and thoughtful response. The reply is not only comprehensive and well-articulated, but also highly insightful—particularly the clarification regarding the issue of reward hacking. All of my concerns have been thoroughly addressed. In light of this, I have decided to revise my score in favour of the paper.

---

> ### Author Response · Authors · 2025-08-04
>
> Dear Reviewer cxPy,
>
> We are writing to express our sincere gratitude for your exceptionally positive and insightful review. We were very encouraged by your thoughtful assessment and are delighted that our rebuttal fully addressed your questions.
>
> Thank you for your strong support of our work and for revising your score in its favor. Your feedback has been invaluable.
>
> Best regards,
>
> The Authors

---

### Official Review · Reviewer_3FLT · 2025-07-03

**Clarity:** 2
**Significance:** 2
**Originality:** 2
**Rating:** 3
**Confidence:** 4

**Summary:**

This work proposes Genesys, a multi-agent system that autonomously discovers new language model architectures using LLMs. The pipeline generates, verifies, and evolves models across scales using a unit-based code representation. Genesys is demonstrated to produce over 1000 verified designs, with some architectures outperforming GPT2 and Mamba2 on standard benchmarks.

**Questions:**

1. Whether the GPT-style architecture is in the knowledge base and serves as a seed architecture of the evolution process? If so, the improvement of the finally searched architecture is relatively minor.

2. Will the proposed system achieve better results than traditional NAS methods that use mutation/cross-over instead of LLM proposers?

3. Could you show the single-best architecture that the proposed system searched? How difference it is compared to current LM architectures? Any new architecture design insights we can learn from it?

**Ethical Concerns:**

["NO or VERY MINOR ethics concerns only"]

**Final Justification:**

While the authors have clarified some aspects of their work, key concerns remain unresolved. The proposed contributions, e.g., factorizing architectures into Generalized Autoregressive Units and introducing a "ladder of scale", appear tailored to this specific architecture search framework, rather than offering broadly applicable insights into ASD (Automated System Design).

Additionally, the absence of a fair and thorough comparison with traditional NAS methods makes it difficult to evaluate the effectiveness and generality of the proposed approach. Given that architecture search performance is highly sensitive to search space design, stronger empirical evidence is necessary to support the claimed benefits. The lack of a clear improvement over GPT-2 further weakens the case for novelty and practical impact.

For these reasons, I lean toward a rejection.

**Limitations:**

yes

**Quality:**

2

**Strengths And Weaknesses:**

**Strength**

1. Generally, the work is clearly written and easy to follow.

2. This work demonstrates good engineering efforts in building an ASD system for a specific purpose.

**Weakness**

1. The major concern with this work is the limited technical novelty and limited research insights offered to the community. This work is more like a demonstration of a successful engineering effort to build an ASD system, while the scientific aspects of LM agents and new insights are insufficient, limiting its value to the research community. In case I missed something, the authors are expected to summarize the insights and major contributions of this work.

2. Another major concern is the lack of an apple-to-apple comparison to demonstrate the effectiveness of LLMs. Simple mutation/crossover, as done in traditional NAS methods, can replace the role of LLMs. A comparison between LLM-based ASD systems and rule-based mutation/crossover in traditional NAS should be provided to show the benefits of LLMs. The effectiveness of LLMs remains unclear to me.

3. The searched architectures are highly sensitive to the search space. The authors are expected to (1) improve the clarity of the search space definition, (2) show the architectural details of the best searched architectures, and (3) ablate the search space choices to validate the robustness of the ASD system.

4. Table 3 is unclear to me, and I do not understand where the better performance on 6 out of 9 benchmarks comes from. It seems the single best model searched in this work is very comparable to GPT.

---

> ### Author Rebuttal · Authors · 2025-07-31
>
> We thank Reviewer 3FLT for the critical and detailed review. We appreciate the opportunity to clarify our work's technical novelty, justify our methods against alternatives like traditional NAS, and elaborate on the scientific insights generated by our system.
>
> ---
>
> ## Weaknesses
>
> >  **W1:** The major concern with this work is the limited technical novelty and limited research insights offered to the community. This work is more like a demonstration of a successful engineering effort to build an ASD system, while the scientific aspects of LM agents and new insights are insufficient, limiting its value to the research community. In case I missed something, the authors are expected to summarize the insights and major contributions of this work.
>
> > *“The major concern with this work is the limited technical novelty and limited research insights...”*
>
> We respectfully disagree regarding the limited technical novelty and insights. Our work introduces key innovations to tackle ASD in a complex, high-cost domain like LM:
>
> * **Novel Genetic Programming (GP) for Code Generation**: Our novel GP framework factorizes architectures into **Generalized Autoregressive Units (GAUs)**, enabling a Viterbi-style search. This addresses a key bottleneck, improving valid LM code generation from 6% (direct prompting) to 92% (**Table 2, proof in Appendix A**).
> * **Budget-Aware "Ladder of Scales" Strategy**: Our "Ladder of Scales" strategy (**Fig. 9**) makes high-cost ASD tractable by pyramidally allocating the verification budget, using smaller scales to efficiently filter promising architectures for promotion to larger, more expensive scales.
> * **Scalable Multi-Agent Research System**: Genesys & LMADE is the first system for this task to model the complete research lifecycle with highly-scalable distributed designs.
>
> > *“This work is more like a demonstration of a successful engineering effort... while the scientific... insights are insufficient...”*
>
> While Genesys is a significant engineering effort, our primary goal was to use it as an instrument to generate scientific insights into both ASD and LM architectures:
>
> * **Systematic Ablation of ASD Components**: Our ablations (**Sec. 5.1, Table 1**) empirically quantify the value of discovery components. Removing experimental feedback (`w/o Exp.`) or literature access (`w/o Lit.`) hurts performance, validating our experiment-guided and knowledge-driven evolution.
> * **Analysis of AI-Driven vs. Human Discovery**: We analyze and contrast the evolutionary trees of AI vs. human research, finding that our system develops a "hubness" pattern of exploitation, distinct from the more diffuse human research network (**Appendix E.1.2, Figs 34-36**).
> * **Linking Architectural Units to Performance**: We pioneer an analysis linking specific GAUs to downstream performance. A classifier using a "Bag-of-Units" representation can predict a design's fitness from its components (0.658 F1 score on held-out data) (**Appendix E.3.3, Tables 17-18**).
>
> > *“In case I missed something, the authors are expected to summarize the insights and major contributions...”*
>
> We thank the reviewer for this suggestion and agree that a concise summary would improve the clarity. We will add an itemized list of contributions to the introduction, summarizing our **System design**, **New Methodology**, **Discovered Architectures**, and **Scientific Insights** as detailed above.
>
> >  **W2:** Another major concern is the lack of an apple-to-apple comparison to demonstrate the effectiveness of LLMs. Simple mutation/crossover, as done in traditional NAS methods, can replace the role of LLMs. A comparison between LLM-based ASD systems and rule-based mutation/crossover in traditional NAS should be provided to show the benefits of LLMs. The effectiveness of LLMs remains unclear to me.
>
> We thank the reviewer for raising this important point. We argue that traditional NAS methods are ill-suited for this domain due to its unique challenges. We will clarify this in **Sec. 2**.
>
> 1.  **Search Space is Too Broad for Fixed Operators**: Traditional NAS requires a concise set of operators, which is infeasible to express diverse paradigms like GPT, SSMs, and TTT in a compact way. *It remains an open problem how to do NAS effectively to the unbounded LM designs.*
>
> 2.  **Prohibitive Evaluation Costs Invalidate Brute-Force Search**: Traditional NAS requires evaluating thousands of candidates, each involves expensive pre-training. LLMs improve **sample efficiency** by proposing knowledge-driven, targeted architectural changes rather than simple syntactic swaps. The importance of this knowledge is confirmed by our `w/o Lit.` ablation (**Table 1**).
>
> Furthermore, LLMs uniquely support the broader scientific goals of ASD by providing **interpretable, human-readable research proposals**, operating on high-level, conceptual units (GAUs), enabling analyses like our unit-performance correlation study (**Appendix E.3.3**).
>
> > **W3:** The searched architectures are highly sensitive to the search space. The authors are expected to (1) improve the clarity of the search space definition, (2) show the architectural details of the best searched architectures, and (3) ablate the search space choices to validate the robustness of the ASD system.
>
> > *“(1) improve the clarity of the search space definition”*
>
> We thank the reviewer for their feedback and agree that the clarity of search space is crucial. The search space is defined by the **Generalized Autoregressive Block (GAB)** code construct (**Sec. 3.1**), constrained by a **Symbolic Checker** that enforces differentiability, causality, stability, and efficiency (**Sec 3.2, Appendix B.1.2**). To improve clarity, we will move key details of the GAB, GAU, and symbolic checks from the appendix into the main paper.
>
> > *“(2) show the architectural details of the best searched architectures”*
>
> We detail the GAU trees for five top architectures in **Appendix G.1.1** and provide a full design artifact for VQH in **Appendix G.2**. We will also release all discovered architectures, including their training results and design histories, with the final paper.
>
> > *“(3) ablate the search space choices to validate the robustness of the ASD system”*
>
> We conducted a series of system-level ablations to validate the robustness and effectiveness of the Genesys system in **Sec. 5.1 (Table 1)**. Removing fitness-based selection (`w/o Exp.`) or literature access (`w/o Lit.`) degrades performance, confirming that our method of navigating the vast architectural search space is effective and robust.
>
> > **W4:** Table 3 is unclear to me, and I do not understand where the better performance on 6 out of 9 benchmarks comes from. It seems the single best model searched in this work is very comparable to GPT.
>
> Your observation is correct. The claim compares the *group* of top 5 discovered models against the *group* of 5 baselines. The statement "outperformed on 6/9 benchmarks" means that for 6 of the 9 tasks, one of our discovered models achieved the highest score among all. It shows Genesys's ability to generate a diverse portfolio of specialized, competitive architectures, which is a key strength, as it's common in architecture comparisons that no single model dominates all tasks (**Sec. 5.3**).
>
> ## Questions
>
> > **Q1:** Whether the GPT-style architecture is in the knowledge base and serves as a seed architecture of the evolution process? If so, the improvement of the finally searched architecture is relatively minor.
>
> **1. Yes, GPT was a seed.** We intentionally included five diverse, state-of-the-art seeds (GPT, Mamba2, etc.) to maximally model the real-world research, where scientists improve upon existing work, other than start from a blank slate and rediscover known architectures.
>
> **2. The contribution is the stable evolutionary progress and potential.** While individual improvements may seem incremental, our key result is a system demonstrating **stable, consistent fitness improvements** over generations (**Table 1, Fig. 8**). It shows the potential for more significant advances with further scaling supported by our scalable designs. Furthermore, the system produced a *portfolio* of models that collectively won on 6/9 benchmarks (**Table 3**), showing its ability to discover specialized architectures.
>
> While this is the largest automated LM discovery of its kind, it's not surprising to see incremental improvements rather than a major leap, given the task's complexity, our relatively low cost for the task scale, and our status as the first work of this kind.
>
> > **Q2:** Will the proposed system achieve better results than traditional NAS methods that use mutation/cross-over instead of LLM proposers?
>
> As detailed in our response to W2, we argue traditional NAS is not viable in this domain due to the huge search space, high costs, and no existing effective NAS solution, making a direct comparison infeasible.
>
> > **Q3:** Could you show the single-best architecture that the proposed system searched? How difference it is compared to current LM architectures? Any new architecture design insights we can learn from it?
>
> We present **VQH** in **Appendix G.2**. It sequentially processes information via three stages: 1) a **selective gating** unit filters relevant information to update memory; 2) a **vector quantization** module compresses this memory by matching it to a learned codebook; and 3) a **hierarchical integrator** organizes these compressed summaries into multi-scale memory banks (short- and long-term). The key insight is that language understanding can be effectively decomposed into these specialized, sequential jobs of filtering, summarizing, and organizing information over time.
>
> ---
>
> We hope these clarifications have adequately addressed the concerns raised. Please do not hesitate to let us know if further details on any of these points would be helpful. We appreciate the opportunity to improve the paper based on this feedback.

---

> > ### Comment · Reviewer_3FLT · 2025-08-05
> >
> > Thank you to the authors for providing a detailed response. Although some of my questions have been addressed, I still have the following concerns:
> >
> > (1) The three novelties mentioned by the authors, such as factorizing architectures into Generalized Autoregressive Units to simplify the search and using a "ladder of scale" to better allocate resources, are specific to the architecture search problem rather than being generally applicable to ASD systems. More generalizable insights and discoveries are expected.
> >
> > (2) I recommended a comparison with traditional NAS methods because architecture search is a tricky problem that heavily depends on the design of the search space. With a well-designed space, even random search can yield increasingly better architectures through more iterations. Without a fair comparison against existing search algorithms in an apple-to-apple setting, it is difficult to assess the true quality of the ASD system. For instance, even if the proposed system cannot assess the quality of architectures, as long as it can generate correct code, better architectures might emerge simply through more search iterations. Moreover, the searched architecture does not appear significantly better than GPT-2, which was used as the initial seed.
> >
> > To acknowledge the authors' efforts, I will raise my score to 3.

---

> ### Author Response · Authors · 2025-08-06
>
> Dear Reviewer 3FLT,
>
>
> We sincerely thank you for your continued engagement, for raising your score, and for providing these additional points for clarification. We appreciate the opportunity to address your remaining concerns regarding the generalizability of our contributions and the comparison to traditional NAS.
>
>
> ---
>
>
> > (1) ...the novelties mentioned... are specific to the architecture search problem rather than being generally applicable to ASD systems.
>
>
> We appreciate this perspective and agree our methods' broader applicability should be more explicit. While developed for LM architecture discovery, our core contributions are generalizable principles for high-cost ASD problems.
>
>
> * **Unit-Based Program Factorization:** The core innovation is factorizing a complex program into modular, evolvable units. Many scientific discovery tasks can be framed as program searches (e.g., finding simulation models or chemical synthesis pathways). Our approach is generalizable for evolving any complex program by relaxing the function signature from a tensor-specific `(X, Z) -> (X', Z')` to a generic `Z -> Z'`, which maps a dictionary of arbitrary typed arguments to an updated one.
>
>
> * **Ladder of Scales:** This "start cheap, then escalate" principle is a cornerstone of efficient, high-cost research. Our method is applicable to fields like **Engineering Design** (ladder of FEM simulation fidelities), **Materials Science** (ladder of DFT calculation precisions), and **Drug Discovery** (ladder of high-throughput screening depths).
>
>
> Furthermore, our work makes two broader contributions to ASD systems, an area still developing standard benchmarks and principled design patterns: We propose **LMADE (Sec. 3.2)** as a concrete, high-cost benchmark for real-world ASD. We will release all code, data, artifacts, and our system console; Unlike ad-hoc ASD workflows, our system design is centered on the *formal principle of program factorization*, which we prove in Appendix A is exponentially more efficient and robust. This offers a generalizable template for future systems.
>
>
> > (2) I recommended a comparison with traditional NAS methods... Without a fair comparison... it is difficult to assess the true quality of the ASD system.
>
> We agree that traditional methods like NAS perform well when the search space is well-defined, local area. However, a vast search space and high evaluation costs are core assumptions of our problem setting. We believe this framing is key to modeling realworld ASD for LM designs and we aim to realistically model the challenges human researchers face, which drove us to design a system that directly confronts this complexity.
>
> These assumptions make it difficult to apply NAS and led to our choice of LLM operators. A key issue is **step size**: NAS typically makes small, syntactic modifications, which is inefficient for exploring a vast space. In contrast, an LLM can propose large, semantically-aware changes. While NAS excels at fine-grained local search, it would struggle to discover a new architectural family, such as transitioning from GPT to Mamba. As the probability of a multi-step random walk locating novel families in such a large space is extremely low, making it an unsuitable baseline.
>
> A promising future direction is a **hybrid approach**: LLMs perform fast, large-step explorations to discover new design families, while NAS conducts fine-grained searches within those promising local areas.
>
> Besides, LLMs provide additional features crucial for ASD, such as better interpretability through unit-based mutations and grounding in scientific literature.
>
>
> > Moreover, the searched architecture does not appear significantly better than GPT-2…
>
>
> We agree the improvements over a strong seed like GPT are incremental. However, in a mature field like LM architecture, demonstrating that an automated system can reliably produce **stable, consistent evolutionary improvements** (Table 1, Fig. 8) upon state-of-the-art designs is a significant achievement.
>
>
> Our results, along with our scalable design, provide confidence that this progress would continue with additional resources, potentially leading to major breakthroughs. Furthermore, our system's budget is modest for a task of this complexity—far less than many large-scale AI discovery systems. Our work is a crucial first step: creating a principled, efficient process that shows a viable path toward more significant, automated breakthroughs.
>
>
>
>
> ---
>
>
> We thank you again for your thoughtful and highly valuable feedback. We will incorporate these important discussions in our revision and are confident they will significantly strengthen the paper.
>
>
> Please let us know if these clarifications have adequately addressed your concerns.
>
>
> Best regards,
>
>
> The Authors

---

> > ### Comment · Reviewer_3FLT · 2025-08-07
> >
> > Thank you to the authors for the further clarification. I am confused about the following claim:
> >
> > > A key issue is step size: NAS typically makes small, syntactic modifications, which is inefficient for exploring a vast space. In contrast, an LLM can propose large, semantically-aware changes. While NAS excels at fine-grained local search, it would struggle to discover a new architectural family, such as transitioning from GPT to Mamba.
> >
> > In my understanding, the step sizes taken by NAS methods are still based on predefined design choices, e.g., varying the number of heads in GPT architectures, switching to linear attention mechanisms like Mamba, or hybridizing them. For LLMs, they cannot invent entirely new architectures from scratch; they still rely on predefined rules. The key difference is that LLMs already possess (or can use tools to retrieve) knowledge of existing architectures and can more freely propose potential modifications. Therefore, whether it is better to guide the search process using expert-designed NAS priors or to let LLMs take over this process remains an open question that requires further validation.

---

> > > ### Author Response · Authors · 2025-08-08
> > >
> > > Dear Reviewer 3FLT,
> > >
> > > Thank you for your thoughtful follow-up and for this crucial point of clarification. We are deeply grateful for the opportunity to elaborate on the distinction between traditional NAS and our LLM-driven approach, as it is central to our work's contribution.
> > >
> > >
> > > ---
> > > > In my understanding, the step sizes taken by NAS methods are still based on predefined design choices, e.g., varying the number of heads in GPT architectures, switching to linear attention mechanisms like Mamba, or hybridizing them.
> > >
> > > This characterization of NAS is accurate and highlights a key assumption: **the efficiency of NAS relies on a deliberately constrained search space**. As we argued in our paper (Sec. 2, lines 76-78), our problem framing intentionally assumes the opposite: we posit that **a vast, open-ended search space is *necessary* for true Automated Scientific Discovery (ASD)**, as it better models real-world scientific discovery settings. Constraining the search space a priori, while computationally convenient, sacrifices the potential for discovering genuinely new insights or novel architectural families—the very goal of scientific discovery. The risk of missing a global optimum by over-constraining the search is, in our view, a critical limitation for this task
> > >
> > >
> > > > For LLMs, they cannot invent entirely new architectures from scratch; they still rely on predefined rules.
> > >
> > >
> > > We would like to clarify the nature of the "rules" in our system. Unlike NAS, there are no predefined *architectural* rules. The only constraints are minimal correctness and validity checks (e.g., differentiability, causality) imposed by our **Symbolic Checker** (Section 3.2, Appendix B.1.2). Within this framework, the LLM can generate free-form code to create conceptually novel architectures by unboundedly synthesizing diverse concepts. For instance, using its "design from scratch" GP operation, our system produced **VQH**—a coherent architecture that uniquely integrates selective gating, vector quantization, and a hierarchical integrator.
> > >
> > >
> > > > The key difference is that LLMs already possess (or can use tools to retrieve) knowledge of existing architectures and can more freely propose potential modifications.
> > >
> > > We agree that the LLM's knowledge is a key advantage. However, its crucial distinction for this task is the ability to take large, semantically-aware "steps"—it can leverage this knowledge to propose novel designs and then implement them via free-form code generation. This capability is what makes our approach so effective in traversing the vast search space. For this reason, **the use of LLMs becomes *necessary* to efficiently explore the vast and high-cost search space of LM design**. Furthermore, LLMs provide other advantages essential for ASD, such as interpretability and literature grounding.
> > >
> > >
> > > > Therefore, whether it is better to guide the search process using expert-designed NAS priors or to let LLMs take over this process remains an open question that requires further validation.
> > >
> > >
> > > We agree this remains an open question. While we argue that LLMs are a necessary tool for this task, **we do not claim they are a universally sufficient or optimal solution.** In fact, we highly value traditional methods like NAS and believe a hybrid approach is a promising future direction—one that uses LLMs for broad, large-step exploration while capitalizing on the **fine-grained search advantage** of NAS for local optimization.
> > >
> > > However, **solely using current NAS methods is not viable for the problem** as we have framed it. Given a vast search space and a realistic verification budget (e.g., 1,062 trained models in our case), the small, syntactic steps of traditional NAS would amount to an inefficient random walk. Such a process would likely only explore minor variations close to the initial seed designs, failing to discover entirely new architectural families.
> > >
> > > Therefore, **a direct, apple-to-apple comparison with current NAS methods is infeasible.** These methods are fundamentally ill-equipped to tackle the assumed scope of our problem. Indeed, **no practical NAS methodology has been established** for this type of unbounded, high-cost discovery task to the best of our knowledge, which makes a direct comparison unavailable.
> > >
> > >
> > > ---
> > >
> > >
> > > Thank you once again for your insightful feedback, which has been invaluable in helping us sharpen the core message of our paper. We will integrate this crucial discussion into our revision to better contextualize our contributions.
> > >
> > > We hope our response addresses your concerns, and if any points remain unclear, please let us know, as we would be more than happy to discuss them further.
> > >
> > >
> > > Best regards,
> > >
> > > The Authors

---

> > > > ### Comment · Reviewer_3FLT · 2025-08-08
> > > >
> > > > Thank you to the authors for providing another round of detailed clarification. Although I agree with the authors that LLMs can handle a vast and open search space more flexibly, I still believe a comparison is needed to understand whether we truly need LLMs for this task, or if NAS remains the better choice at present. Otherwise, it is unclear whether LLMs genuinely possess the ability to find better architectures, or if they are simply proposing architectures in a more random way and occasionally hitting good designs through extensive exploration. Including the discovery process and the rationale behind LLM decisions could be meaningful in the next version.
> > > >
> > > > I will take the above clarification into consideration during the discussion phase with the other reviewers.

---

> ### Author Response · Authors · 2025-08-08
>
> Dear Reviewer 3FLT,
>
> We sincerely appreciate your continued engagement and thoughtful follow-up. Your points are crucial, and we are grateful for the opportunity to clarify our position.
>
>
> > Thank you to the authors for providing another round of detailed clarification. Although I agree with the authors that LLMs can handle a vast and open search space more flexibly, I still believe a comparison is needed to understand whether we truly need LLMs for this task, or if NAS remains the better choice at present.
>
>
> We agree that understanding the trade-offs between different search paradigms is vital. However, we believe this perspective presents a false dilemma. Our work is not predicated on the idea that LLMs must be proven "better" than NAS to be valuable. Rather, we argue that ***LLMs enable a fundamentally different paradigm***—one better aligned with the goals of open-ended ASD through its unique advantages that traditional NAS cannot, such as generating interpretable design rationales and modeling the human research process from literature review to verification.
>
> The value, therefore, lies not just in the final architecture but in the scientific modeling of the discovery process itself. For these reasons, we argue that **exploring LLM-based methods is scientifically valuable, regardless of whether NAS remains the better choice at present**, particularly as **no suitable NAS baseline** for this unbounded, high-cost discovery task currently exists to compare with.
>
>
>
> > Otherwise, it is unclear whether LLMs genuinely possess the ability to find better architectures, or if they are simply proposing architectures in a more random way and occasionally hitting good designs through extensive exploration.
>
> We respectfully disagree with the premise that our results stem from random or "extensive" exploration, as the evidence points to a demonstrably guided and efficient search. **While training 1,062 novel architectures is a major effort, it represents a tiny fraction of the unbounded design space, making random success highly improbable.** In contrast, our "Full" system shows consistent fitness improvement over generations, a positive trend vanishes in our ablation study when experimental feedback is removed (`w/o Exp.`), proving the search is guided by evidence, not chance.
>
>
> > Including the discovery process and the rationale behind LLM decisions could be meaningful in the next version.
>
> We agree completely, and illustrating the discovery process is a central feature of our work. We provide insights at two levels: a macro-level analysis of discovery patterns via the evolutionary tree (Appendix E.1.2), and a micro-level case study with a detailed, multi-page design artifact for a top model, VQH (Appendix G.2). This appendix showcases the entire LLM-generated rationale—from the initial literature review and motivation to the mathematical justification, design plan, and final adversarial review. We will expand this analysis in the final paper, adding further discussion to explicitly detail the rationale behind these LLM-driven decisions.
>
>
> To further this commitment to transparency, we will release our complete design artifacts with training records for all 1,162 architectures, via an **interactive online console** that will allow the community to deeply analyze the automated discovery process.
>
> ---
>
> Thank you once again for your critical and highly valuable feedback. We hope these clarifications further solidify the contributions and rationale of our work.
>
> Best regards,
>
> The Authors

---

### Note · Authors · 2025-08-12

Dear Area Chair and Reviewers,

We thank all reviewers for their insightful feedback, which has helped sharpen our paper's message. We conclude by highlighting our work's three core contributions:

---

First, we introduce and validate **a vital new paradigm for high-cost Automated Scientific Discovery (ASD)**. We argue this approach is necessary for problems with vast, unbounded search spaces like LM design, where traditional methods are ill-suited. Our full discussion with **Reviewer 3FLT** details this crucial framing.

Second, our system is proven to be **scaling-ready and extensible**. The "Ladder of Scales" strategy is central to this, reducing required GPU-hours by over 95%, as clarified for **Reviewer cxPy**. Our core principles of **unit-based program factorization** and multi-fidelity evaluation are extensible to broader ASD domains like materials science or drug discovery, as detailed in our rebuttal to **Reviewer 3FLT**. Moreover, the system's stable evolutionary improvement upon state-of-the-art designs provides strong evidence of its potential.

Finally, we provide **valuable resources and insights for the community**. This includes:
1.  **1,162 novel LM architectures**, with a portfolio of top designs that collectively outperform strong baselines on 6/9 benchmarks. To the best of our knowledge, this is the largest ASD experiment of its kind.
2.  **Novel scientific analyses** that link architectural components to performance (a capability confirmed for **Rev iHdi**; Appx E.3.3), alongside extensive empirical analyses for practitioners (Appx E).
3.  The full **open-source release** of our Genesys/LMADE system, all code, data, discovery artifacts, and an interactive console to ensure reproducibility, addressing the transparency questions from **Reviewer tRdq**.

We believe this work establishes a viable path toward more significant, automated breakthroughs in this new paradigm.

---

Thank you once again for your constructive engagement and valuable feedback.

Sincerely,
The Authors of Submission 8319

---

### Decision · Program_Chairs · 2025-09-17

**Decision:**

Accept (spotlight)

**Comment:**

This paper proposes solving the problem of LLM architecture search by using LLMs in agentic setups. Essentially, at each stage and scale, the LLM agents propose, generate code, train and evaluate LMs and then select promising architectures for subsequent evolution. Although this appears similar to NAS, the concept is slightly different, since everything from ideation to development to iteration is done via LLMs. Overall the reviewers have indicated that this is an interesting work as well with the major complaints centered around the scale of models to be tested and the maximum model sizes that can be tested. However, given the agentic approach for architecture discovery, I believe that this work will be of interest to, at the very least, the agentic community.